# Glycan chip based on structure-switchable DNA linker for on-chip biosynthesis of cancer-associated complex glycans

Hye Ryoung Heo [1,2], Kye Il Joo [2], Jeong Hyun Seo [3], Chang Sup Kim [4✉] & Hyung Joon Cha [2✉]

On-chip glycan biosynthesis is an effective strategy for preparing useful complex glycan sources and for preparing glycan-involved applications simultaneously. However, current methods have some limitations when analyzing biosynthesized glycans and optimizing enzymatic reactions, which could result in undefined glycan structures on a surface, leading to unequal and unreliable results. In this work, a glycan chip is developed by introducing a pH-responsive i-motif DNA linker to control the immobilization and isolation of glycans on chip surfaces in a pH-dependent manner. On-chip enzymatic glycosylations are optimized for uniform biosynthesis of cancer-associated Globo H hexasaccharide and its related complex glycans through stepwise quantitative analyses of isolated products from the surface. Successful interaction analyses of the anti-Globo H antibody and MCF-7 breast cancer cells with on-chip biosynthesized Globo H-related glycans demonstrate the feasibility of the structure-switchable DNA linker-based glycan chip platform for on-chip complex glycan biosynthesis and glycan-involved applications.

[1] School of Interdisciplinary Bioscience and Bioengineering, Pohang University of Science and Technology, Pohang, Republic of Korea. [2] Department of Chemical Engineering, Pohang University of Science and Technology, Pohang, Republic of Korea. [3] School of Chemical Engineering, Yeungnam University, Gyeongsan, Republic of Korea. [4] School of Chemistry and Biochemistry, Yeungnam University, Gyeongsan, Republic of Korea. ✉email: cskim1409@ynu.ac.kr; hjcha@postech.ac.kr

Glycan–protein interactions play an important role in a variety of biological processes, including angiogenesis, stem cell development, immune responses, and neuronal development[1,2]. Alterations in glycosylation patterns have also been shown to regulate the development and progression of cancer[3]. Therefore, it is important to understand these interactions to analyze the mechanisms of glycan-mediated biological processes and to develop therapeutic agents to treat glycan-related diseases.

Glycan chips have been developed in response to the essential need for high-throughput analysis of glycan-involved interactions. Currently, glycan chips have been employed for screening therapeutic agents and profiling glycan-involved interactions, including glycan–lectin, glycan–cytokine, glycan–antibody, and glycan–virus/bacteria interactions[4–8]. However, studies on glycan-related biological processes using glycan chips are still at an early stage relative to our knowledge of the biological functions of proteins and genes. To construct glycan chips, most homogeneous glycans are commonly provided by either chemical synthesis or natural purification using multiphase chromatography. These methods have several limitations, including being labor-intensive, costly, and time-consuming processes, the requirement for protection and deprotection steps, and the difficulty in controlling the stereochemistry of glycosidic linkages, resulting in poor purity and low stepwise yield[9–11]. Limited access to glycan libraries with diverse structures restricts extensive studies on their roles in vivo using glycan chips. In addition, current glycan chip platforms have used a method of conjugating chemoenzymatically synthesized glycans with linkers to immobilize them on the surface[4–8]. This method would be unsuitable for complex glycans because these compounds have a significantly low synthesis yield compared to simple glycans, and conjugation with linkers has the limitations of being capable of eliminating labile sialic acid and resulting in significant loss[12–14].

Considering that on-chip syntheses of oligonucleotides and peptides have been successfully utilized for genomics and proteomics[15–17], on-chip enzymatic glycan synthesis is an attractive tool for glycomics. This method has several merits over conventional methods, including a low-cost and simple process without any additional protection/deprotection, purification, and immobilization steps, the use of small amounts of expensive glycan-processing enzymes and nucleotide sugar donors, the synthesis of many glycosidic linkages in a straightforward manner, and the direct application of synthesized glycans for glycomics. However, only a few glycan chips have been generated by on-chip enzymatic synthesis[18–20]. Previously, we demonstrated the on-chip enzymatic synthesis of GM1 pentasaccharide-related complex glycans[19]. Although the feasibility of on-chip complex glycan biosynthesis was confirmed, the current platforms have a key technical barrier. Quantitative information on biosynthesized complex glycans on the surface is unavailable because it is impossible to isolate and analyze immobilized glycans on the current platforms. In addition, these studies analyzed on-chip enzymatic glycosylation reactions by only interacting with fluorescence dye-labeled lectins. It is impossible to calculate the enzymatic glycosylation efficiency from the fluorescence intensity value of lectin bound on the chip. These drawbacks make it difficult to control and optimize stepwise glycosylation reactions using the current platforms, resulting in structurally undefined glycans on the chip, which leads to unequal and unreliable results. Therefore, an advanced strategy is required for on-chip enzymatic glycosylation-based glycan chips.

A DNA hybridization-based immobilization method has been used to immobilize glycans for fabrication of glycan chips[21–26]. Double-stranded DNA (dsDNA) linkers act as rigid arms, allowing better presentation of glycans and glycoclusters on the surface[21–23,27]. They also permit tailoring of spatial arrangements[21,22,27]. This performance leads to the glycoside cluster effect[21–23,25,27,28], which is a key factor for glycan–protein interactions[24,26,29,30]. In addition, DNA linkers were used as identifiers to code individual glycans binding to glycan-binding proteins and cells by conjugating individual oligonucleotides according to glycans[31]. With these merits, a DNA-based glycan chip coupled with mass spectrometry showed the possibility of on-chip glycan biosynthesis by analyzing the activities of glycan-processing enzymes on the chip[24].

In the present work, we develop a glycan chip platform for the on-chip enzymatic synthesis of complex glycans based on pH-responsive i-motif DNA as a linker material for immobilizing glycans on solid supports. The i-motif DNA containing stretches of cytosine residues forms a stable four-stranded helical secondary structure (quadruplex) at acidic pH[32–35]. The pH-responsive structural switch makes it possible to reversibly immobilize and isolate glycans on a solid surface. As shown in Fig. 1a, conjugates of glycans and complementary single-stranded oligonucleotides can hybridize to i-motif DNAs that are immobilized on the chip surface under slightly basic conditions. As the pH is lowered, i-motif DNA tends to strongly form a quadruplex structure, resulting in the denaturation of the DNA double helix and thereby enabling the isolation of glycan-oligonucleotide conjugates from the surface. This property would make it possible to optimize on-chip enzymatic glycosylation by analyzing isolated glycan-oligonucleotide conjugates using liquid chromatography, resulting in the synthesis of structurally defined complex glycans on chip. Therefore, our proposed glycan chip platform can improve the limitations of current platforms by immobilizing structurally simple disaccharides with high synthesis yields and then synthesizing complex glycans on the chip using glycosyltransferases under optimized conditions.

To determine the feasibility of on-chip enzymatic glycosylation using the structure-switchable DNA-based glycan chip platform, Globo H series (Table 1) are selected as target complex glycans, which are aberrantly overexpressed on human tumor cells and known to be involved in tumor progression[36,37]. Five Globo H-related complex glycans (from trisaccharide to hexasaccharide) are successfully synthesized from chip surface-immobilized lactose (disaccharide) using the pH-responsive i-motif DNA linker under optimized conditions of stepwise enzymatic glycosylation reactions. We hypothesize that specific binding proteins for glycans can be present on breast cancer cells for tumor progression and metastasis when considering that globoseries glycosphingolipids Gb5, SSEA-4, and Globo H are specifically overexpressed on cancer cells[38–40]. We analyze interactions between breast cancer cells and on-chip-biosynthesized Globo H-related complex glycans to examine the glycan-binding specificity. This analysis can show the potential of our DNA-based glycan chip platform combined with on-chip enzymatic glycosylation to prepare complex glycan sources and to simultaneously analyze glycan-related biological interactions.

## Results

**Preparation of DNA linker for glycan chip.** i-motif DNAs with four stretches of a series of cytosines (CCCs) in sequence were modified with thiolated T10 at 5′ to immobilize them onto gold-coated glass slides (Supplementary Table 1). The T10 sequence was introduced to increase the structural stability of i-motif DNA and to prevent fluorescence quenching by maintaining a constant distance from the gold surface. Complementary single-stranded DNA (ssDNA) with three mismatches was designed for glycan-oligonucleotide conjugates (Supplementary Table 1). The

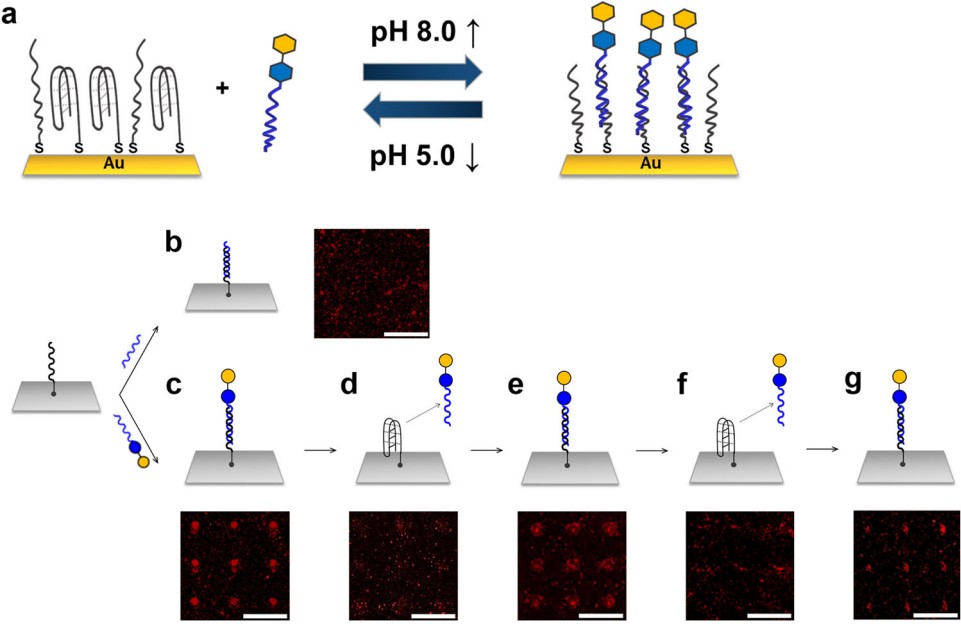

**Fig. 1 Structure-switchable DNA-based glycan chip platform. a** Schematic illustration of a structure-switchable DNA-based glycan chip platform using a pH-responsive i-motif DNA linker. **b** Schematic illustration and scanned raw image for hybridization of complementary single-stranded oligonucleotides with surface-immobilized i-motif DNAs under basic conditions (pH 9.0). **c–g** Schematic illustrations and scanned raw images for hybridization and denaturation of lactose-oligonucleotide conjugates by pH-responsive structural change in surface-immobilized i-motif DNAs. **c** The first round hybridization of lactose-oligonucleotide conjugates with surface-immobilized i-motif DNAs under basic conditions (pH 9.0). **d** The first round of denaturation of lactose-oligonucleotide conjugates from surface-immobilized i-motif DNAs under acidic conditions (pH 4.5). **e** The second round of hybridization of isolated lactose-oligonucleotide conjugates with surface-immobilized i-motif DNAs under basic conditions. **f** The second round of denaturation of lactose-oligonucleotide conjugates from surface-immobilized i-motif DNAs under acidic conditions. **g** The third round of hybridization of isolated lactose-oligonucleotide conjugates with surface-immobilized i-motif DNAs under basic conditions. The hybridized lactose-oligonucleotide conjugates were detected by using biotinylated RCA$_{120}$ lectin and Alexa Fluor® 647-conjugated streptavidin. Scale bar is 800 μm. Symbols: blue circle, Glc; yellow circle, Gal.

**Table 1 Glycans used in this work and their sequences.**

| Glycan | Saccharide | Sequence | Symbol[a] |
|--------|-----------|----------|--------|
| Globo H | Hexasaccharide (globohexaose) | Fucα1-2Galβ1-3GalNAcβ1-3Galα1-4Galβ1-4Glc | |
| SSEA 4 | Hexasaccharide (globohexaose) | Neu5Acα2-3Galβ1-3GalNAcβ1-3Galα1-4Galβ1-4Glc | |
| Gb5 | Pentasaccharide (globopentaose) | Galβ1-3GalNAcβ1-3Galα1-4Galβ1-4Glc | |
| Gb4 | Tetrasaccharide (globotetraose) | GalNAcβ1-3Galα1-4Galβ1-4Glc | |
| Gb3 | Trisaccharide (globotriaose) | Galα1-4Galβ1-4Glc | |
| Lactose | Disaccharide | Galβ1-4Glc | |

[a] ●, glucose (Glc); ●, galactose (Gal); ■, N-acetylgalactosamine (GalNAc); ◆, N-acetylneuraminic acid (Neu5Ac); ▶, fucose (Fuc)

three mismatches enable prevention of the formation of the G-quadruplex structure of the ssDNA itself and facilitate the denaturation of hybridized dsDNA[41]. Lactose-oligonucleotide conjugates were synthesized using a thiol-ene photochemical reaction in which 5′ thiol-modified complementary ssDNA was covalently linked to allyl lactose under UV light (Supplementary Fig. 1). The synthesized lactose-oligonucleotide conjugates were purified using high-pressure liquid chromatography (Supplementary Fig. 2) and confirmed by [1]H nuclear magnetic resonance (NMR) and mass spectrometry (MS) analyses (Supplementary Figs. 3–9).

**Development of a glycan chip platform based on a pH-responsive DNA linker.** Under acidic conditions, i-motif DNA exists as a four-stranded quadruplex structure via intramolecular base pairing between cytosine (C) and protonated cytosine (C+) in its sequence[42]. The folded structure can prevent the dense immobilization of i-motif DNAs on the surface, providing sufficient surface space for reversible structural changes in i-motif DNA. First, 70 µM i-motif DNAs dissolved in phosphate-buffered saline (PBS) solution (pH 4.5) were robotically spotted onto a gold chip and incubated overnight in a humidified chamber. i-motif DNAs were attached onto a gold surface via gold–thiol interactions[43]. After immobilization of i-motif DNAs, the chip was blocked with poly(ethylene glycol) methyl ether thiol in PBS (pH 4.5). The lactose-oligonucleotide conjugates were incubated onto an i-motif DNA-immobilized surface that was pretreated with PBS buffer (pH 9.0). DNA hybridization-based lactose immobilization was confirmed through interaction analysis of *Ricinus communis* agglutinin I (RCA$_{120}$) lectin, which specifically binds to terminal galactose (Gal). While the complementary ssDNA-treated surface did not show strong fluorescence intensity (Fig. 1b), the lactose-oligonucleotide conjugate-treated surface did (Fig. 1c). To check whether immobilization and isolation of lactose-oligonucleotide conjugates could be controlled by reversible structural changes of i-motif DNAs on the surface, the fabricated chip was sequentially treated with pH 4.5 and pH 9.0 PBS buffers. When the chip was incubated with RCA$_{120}$ after treatment with the pH 4.5 solution, no fluorescence was observed (Fig. 1d, f). To further validate the separation of glycan-oligosaccharide conjugates at acidic pH, single-stranded oligonucleotides conjugated with Alexa Fluor® 647 at the 5′ end were used as a model. There was barely any fluorescence even at high concentrations of i-motif DNA when treated with acidic solution (pH 4.5) after incubating Alexa Fluor® 647-conjugated complementary oligonucleotides on the chip (Supplementary Fig. 10). Because Alexa Fluor® 647 dye is pH-resistant from pH 4 to pH 10[44], the fluorescence change seemed to be due to the separation of the dye-conjugated oligonucleotides, not to the inactivation of the dye in acidic conditions. These results indicated that lactose-oligonucleotide conjugates were completely separated from the chip surface by forming a quadruplex shape of i-motif DNA under acidic conditions. When the isolated lactose-oligonucleotide conjugates dissolved in pH 9.0 buffer were added again on the chip immobilized with i-motif DNAs, the lactose-oligonucleotide conjugates were successfully rehybridized (Fig. 1e, g). These results showed that the pH-dependent structural change of i-motif DNAs caused the glycan-oligonucleotide conjugates to be separated from the surface and reimmobilized on the surface. Consequently, a glycan chip platform based on the reversible structural change of i-motif DNAs in a pH-dependent manner was successfully developed to quantitatively analyze complex glycans biosynthesized on the surface and to directly provide complex glycans for glycan-related applications.

**On-chip enzymatic synthesis of cancer-associated complex glycans.** The hybridized form should be structurally stable under neutral pH conditions in which on-chip enzymatic glycosylation reactions are performed. Thus, prior to on-chip glycan biosynthesis, we checked the stability of the hybridized form in pH 7.0 solution according to incubation time. After treatment with complementary ssDNA on the i-motif DNA-immobilized chip, the chip was incubated with a pH 7.0 solution for 24–72 h, and then doxorubicin, which has intrinsic fluorescence, was added to the chip. There was no change in fluorescence intensity until 72 h of incubation time (Supplementary Fig. 11), indicating that hybridization is structurally stable under on-chip enzymatic glycosylation conditions, which is supported by a previous study showing that i-motif DNA has a linear structure in solution above pH 6.4[32].

To substantiate the feasibility of the DNA-based glycan chip platform for on-chip biosynthesis of complex glycans, enzymatic glycosylations were performed on a chip surface using several glycosyltransferases. Each glycosylation product was analyzed by using fluorescent dye-labeled lectins. All enzymatic reactions were performed for 48 h at 37 °C. First, a solution of Tris-HCl (pH 7.0) containing α-1,4-galactosyltransferase (LgtC), UDP-Gal, and MgCl$_2$ was applied to the lactose disaccharide-immobilized surface. The α-Gal-specific *Griffonia simplicifolia* isolectin B$_4$ (GS-IB$_4$) bound to the glycosylated products on the LgtC enzyme-treated surface, indicating that Gb3 trisaccharides were successfully biosynthesized (Fig. 2a). Next, the synthesized Gb3 trisaccharide-immobilized surface was treated with a Tris-HCl (pH 7.0) solution containing β-1,3-*N*-acetylgalactosaminyltransferase (LgtD), UDP-*N*-acetylgalactosamine (UDP-GalNAc), and MgCl$_2$. The soybean agglutinin (SBA) lectin interacted with the products synthesized by the LgtD enzyme on the chip surface, while the LgtD-untreated surface showed no fluorescence (Fig. 2b). This result indicated that GalNAc residues were successfully transferred onto immobilized Gb3 trisaccharide by LgtD. A Tris-HCl (pH 7.0) solution containing UDP-Gal, LgtD, and MgCl$_2$ was incubated on a Gb4 tetrasaccharide-immobilized surface. RCA$_{120}$ lectin bound to the microspots where Gb4 tetrasaccharides reacted with LgtD on the surface, indicating that Gb5 pentasaccharide was enzymatically synthesized from Gb4 tetrasaccharide (Fig. 2c). Then, a solution of α-2,3-sialyltransferase (α2,3-SialT) was added to the Gb5 pentasaccharide-immobilized surface to synthesize SSEA-4 hexasaccharide. The α2,3-linked *N*-acetylneuraminic acid (Neu5Ac)-specific *Maackia amurensis* lectin II (MAL II) interacted with the synthesized SSEA-4 hexasaccharide on the surface, while the α2,3-SialT-untreated surface did not (Fig. 2d). This result indicated that Neu5Ac was transferred to immobilized Gb5 pentasaccharide by α2,3-SialT. Finally, Globo H hexasaccharide was synthesized by incubating a solution of Tris-HCl (pH 7.0) containing α-1,2-fucosyltransferase (α1,2-FucT), GDP-fucose (GDP-Fuc), and MgCl$_2$ onto a Gb5 pentasaccharide-immobilized surface. However, the α-L-Fuc-specific *Lotus tetragonolobus* (LTL) lectin poorly interacted with the synthesized Globo H hexasaccharide on the surface (Fig. 2e). This result might be due to the intrinsically poor binding affinity of LTL to large glycans containing Fucα1-2Galβ1-3(4)GlcNAc(GalNAc)[45].

**Optimization of on-chip glycan biosynthesis reactions.** To optimize the on-chip enzymatic glycosylations, each reaction was performed by changing the reaction times in the presence of sufficient amounts of glycosyltransferases and glycan donors. The biosynthesized products were recovered from the chip surface by DNA denaturation through pH-dependent structural change of

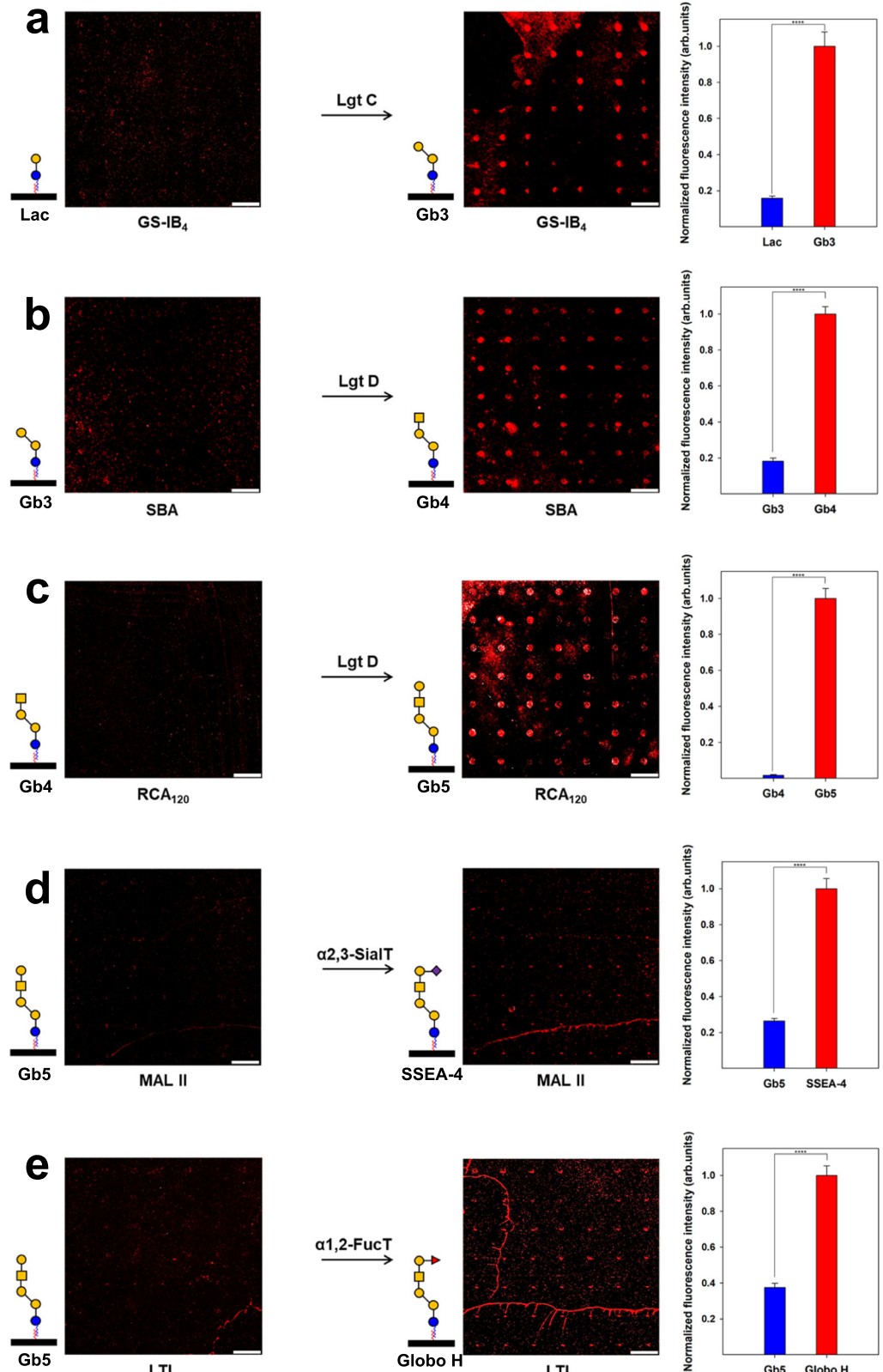

the i-motif DNA and quantitatively analyzed using liquid chromatography to determine the conversion efficiency for the biosynthesized glycans (Fig. 3a). The LgtC enzyme was treated on a lactose disaccharide-immobilized surface for 12, 24, and 48 h at 37 °C. Initially, Gb3 trisaccharides were enzymatically synthesized slowly at a conversion efficiency of ~14%, but after 48 h, all lactose disaccharides were converted to Gb3 trisaccharides on the surface (Fig. 3b and Supplementary Fig. 12). For Gb4 tetrasaccharide synthesis, Gb3 trisaccharide-immobilized surfaces (LgtC reaction for 48 h at 37 °C) were reacted with the LgtD enzyme. Approximately 80% of Gb3 trisaccharide was rapidly converted to Gb4 tetrasaccharide by the LgtD reaction within 12 h. Moreover, the conversion was almost complete within 24 h (Fig. 3b and Supplementary Fig. 13). Compared with Gb3

**Fig. 2 Enzymatic glycosylation on structure-switchable DNA-based glycan chip.** Scanned raw images and quantitative intensity plots for on-chip biosynthesized (**a**) Gb3 trisaccharide, (**b**) Gb4 tetrasaccharide, (**c**) Gb5 pentasaccharide, (**d**) SSEA-4 hexasaccharide, and (**e**) Globo H hexasaccharide. Synthesized complex glycans were detected by using biotinylated lectins and Alexa Fluor® 647-conjugated streptavidin. Each value presents the mean ± SEM from forty-nine independent spots excluding the highest and lowest signals. Statistical significance was assessed using Student's unpaired $t$ test (****$p < 0.0001$). GS-IB$_4$ *Griffonia simplicifolia* isolectin B$_4$, SBA soybean agglutinin lectin, RCA$_{120}$ *Ricinus communis agglutinin* I lectin, MAL II *Maackia amurensis* lectin II, LTL *Lotus tetragonolobus* lectin, LgtC α-1,4-galactosyltransferase, LgtD β-1,3-*N*-acetylgalactosaminyltransferase/β-1,3-galactosyltransferase, α2,3-SialT α-2,3-sialyltransferase, α1,2-FucT α-1,2-fucosyltransferase. Symbols: blue circle, Glc; yellow circle, Gal; yellow square, GalNAc; red triangle, Fuc; purple square, Neu5Ac. Scale bar is 800 μm. Source data are provided as a Source Data file.

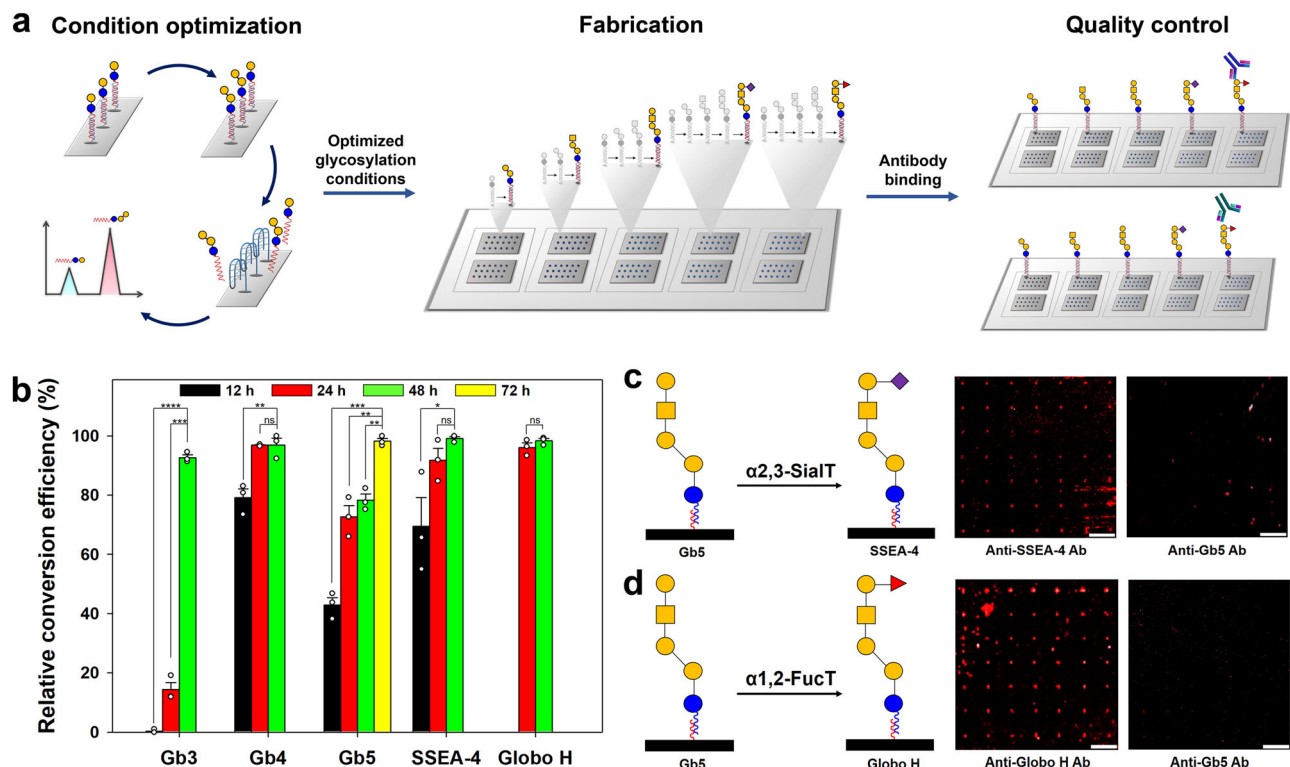

**Fig. 3 Structure-switchable DNA-based glycan chip combined with on-chip glycan biosynthesis. a** The workflow for the construction of a structure-switchable DNA-based glycan chip via on-chip complex glycan biosynthesis under optimized conditions. On-chip enzymatic glycosylation conditions were optimized by analyzing biosynthesized glycans isolated from the chip in acidic pH (4.5) through Bio-LC. To fabricate a DNA-based glycan chip composed of Globo H and its related structures, a lactose-oligonucleotide conjugate-immobilized surface was divided into five blocks, followed by treatment with glycosyltransferases under optimized conditions. For quality control, the DNA-based glycan chips were incubated with antibodies against starting materials and products. **b** Quantitative analyses of relative conversion efficiencies for on-chip enzymatic syntheses of Globo H hexasaccharide and related structures in Supplementary Figs. 6–10. Each value presents the mean ± SEM from forty-nine independent spots excluding the highest and lowest signals. Statistical significance was assessed using Student's unpaired $t$ test (*$p < 0.05$; **$p < 0.01$; ***$p < 0.001$; ****$p < 0.0001$). Schematic illustrations and scanned raw images for on-chip enzymatic glycosylation of (**c**) SSEA-4 hexasaccharide and (**d**) Globo H hexasaccharide under optimized conditions (Scale bar: 800 μm). Synthesized complex glycans were detected by using DyLight 650-conjugated monoclonal antibodies or monoclonal antibodies with Alexa Fluor® 647-conjugated polyclonal secondary antibodies. Anti-Gb5 Ab DyLight 650-conjugated anti-Gb5 monoclonal antibody, Anti-SSEA-4 Ab DyLight 650-conjugated anti-SSEA-4 monoclonal antibody, Anti-Globo H Ab Anti-Globo H monoclonal antibody (VK9), α2,3-SialT α-2,3-sialyltransferase, α1,2-FucT α-1,2-fucosyltransferase. Symbols: blue circle, Glc; yellow circle, Gal; yellow square, GalNAc; red triangle, Fuc; purple square, Neu5Ac. Source data are provided as a Source Data file.

trisaccharide, Gb4 tetrasaccharide was biosynthesized on the surface more quickly due to the higher catalytic activity of LgtD than of LgtC[46]. In the same way, a solution of LgtD containing UDP-Gal was added on the prepared Gb4 tetrasaccharide-immobilized surface to synthesize Gb5 pentasaccharide. The LgtD enzyme exhibits β-1,3-*N*-acetylgalactosaminyltransferase activity when Gb3 trisaccharide and UDP-GalNAc are used as an acceptor and a donor, respectively[44]. The enzyme also shows β-1,3-galactosyltransferase activity when we use Gb4 tetra-saccharide as an acceptor and UDP-Gal as a donor[47]. Unlike the biosynthesis of Gb4 tetrasaccharide using the LgtD enzyme, Gb5 pentasaccharide was slowly synthesized on the surface at a

conversion efficiency of ~43% for 12 h (Fig. 3b and Supplementary Fig. 14). Even after 48 h, the conversion efficiency of Gb4 tetrasaccharide to Gb5 pentasaccharide was ~80%. Thus, an enzymatic reaction time of 72 h was required for complete bio-synthesis of Gb5 pentasaccharide from Gb4 tetrasaccharide using the same amount of LgtD. This result might be explained because LgtD has much lower activity with Gb4 tetrasaccharide than with Gb3 trisaccharide as an acceptor[43]. Finally, biosynthesis of Globo H hexasaccharide and SSEA-4 hexasaccharide was conducted on a Gb5 pentasaccharide-immobilized surface using α1,2-FucT and α2,3-SialT, respectively. The Gb5 pentasaccharide-immobilized surface was prepared by serial enzymatic reactions under the

optimized conditions. The results showed that Gb5 pentasaccharides were almost completely converted to Globo H and SSEA-4 hexasaccharides on the chip surface within 24 h, which might be due to the high catalytic activities of both enzymes (Fig. 3b and Supplementary Figs. 15, 16)[48–50]. To check whether there are minor unreacted starting glycans on the chip after enzymatic reactions under optimized conditions, we analyzed antibody binding to spots where immobilized glycans reacted with glycosyltransferases (Fig. 3a). Due to the absence of commercially available anti-lactose and anti-Gb4 antibodies, it was impossible to confirm that all glycosylation processes were completed. However, using anti-Gb5, anti-SSEA-4, and anti-Globo H antibodies, we confirmed that there was no fluorescence when the antibodies against starting glycans were treated on the chip after on-chip enzymatic syntheses of Gb4 tetrasaccharide, SSEA-4 hexasaccharide, and Globo H hexasaccharide (Fig. 3c, d and Supplementary Fig. 17). These results clearly indicated that the on-chip enzymatic glycosylations completely proceeded under the conditions used, along with high reproducibility (relative standard deviation of 0.9–3.4%).

**Interaction analysis of anti-Globo H antibody with on-chip biosynthesized Globo H-related complex glycans.** To prepare a Globo H-related glycan chip and apply it for interaction analysis with antibody, we performed sequential syntheses of Gb3 trisaccharide, Gb4 tetrasaccharide, Gb5 pentasaccharide, Globo H hexasaccharide, and SSEA-4 hexasaccharide on a lactose disaccharide-immobilized chip using glycosyltransferases under their optimized conditions. Briefly, the lactose disaccharide-immobilized surface was divided into five blocks, and these blocks were treated with glycosyltransferase(s) and sugar nucleotide(s) to individually biosynthesize Globo H hexasaccharide and its related glycans (Fig. 3a). The relative glycan-binding specificity of a mouse IgG anti-Globo H monoclonal antibody (VK9) was investigated for on-chip biosynthesized Globo H-related complex glycans. It has been established that the VK9 antibody has a high binding affinity to Globo H hexasaccharide without any cross-reactivity to other Globo H analogs[51,52]. Scanned fluorescence imaging showed that VK9 had a much higher binding affinity to biosynthesized Globo H hexasaccharide than to the other synthesized glycans (Fig. 4a, b), consistent with a previously reported binding specificity of VK9[52]. The difference in fluorescence intensities of Globo H and SSEA-4 hexasaccharides showed that the Fuc moiety plays a significant role in VK9 binding, which is supported by previous reports[51,52]. These results confirmed that the Globo H hexasaccharide series was successfully biosynthesized on the lactose disaccharide-immobilized surface.

**Interaction analysis of cholera toxin B subunit with on-chip biosynthesized GM1-related complex glycans.** To further validate the feasibility of the structure-switchable DNA-based glycan chip for on-chip complex glycan biosynthesis, GM1 pentasaccharide and its related glycans were also synthesized on the lactose disaccharide-immobilized surface using glycosyltransferases under their optimized conditions (Supplementary Table 2 and Supplementary Figs. 18–20). Unlike GM3 trisaccharide and GM2 tetrasaccharide, GM1 pentasaccharide was biosynthesized at low efficiency. This result was due to the low catalytic activity of the CgtB enzyme (17 U/L), which was consistent with a previous study[53]. We analyzed the interaction of the cholera toxin B subunit and GM1 pentasaccharide-related complex glycans biosynthesized on the chip surface (Supplementary Fig. 21). This result showed that the intrinsic selectivity of the cholera toxin B subunit to the on-chip biosynthesized glycans was consistent with previous studies[5,19].

**Interaction analysis of MCF-7 breast cancer cells with on-chip biosynthesized Globo H-related complex glycans.** Screening the glycan-binding specificities of cancer cells enables the discovery of glycan-based target probes and suggests a direction for cancer cell targeting for effective diagnostics and therapy. To examine the feasibility of a DNA-based glycan chip combined with on-chip glycan biosynthesis for this screening application, we analyzed glycan binding of MCF-7 breast cancer cells on the developed Globo H series glycan chip composed of Globo H hexasaccharide and its related complex glycans. The Globo H series glycan chip was constructed as described above. All living cells ($4 \times 10^5$ cells/mL) stained with calcein-AM dye were applied on the glycan chip. MCF-7 breast cancer cells strongly recognized Globo H hexasaccharide on the chip, while MCF-10A normal breast cells did not (Fig. 4c, d). This was also supported by flow cytometry analyses of MCF-7 cancer and MCF-10A normal cells for Globo H hexasaccharide binding (Fig. 4e–g). Except for that of Globo H hexasaccharide, the binding affinities of both cells for other biosynthesized complex glycans were similar. These results indicated that Globo H hexasaccharide-binding proteins can be present on MCF-7 breast cancer cells, which supports a previous study that identified the major binding protein for Globo H hexasaccharide on the cancer cell membrane[54]. In addition, quantitative binding analysis was performed using a cell mixture with different ratios of MCF-7 cancer cells to MCF-10A normal cells. The glycan chip exhibited stronger fluorescence intensity according with an increased ratio of MCF-7 cells to MCF-10A cells (Supplementary Fig. 23).

In this work, we clearly demonstrated that the structure-switchable DNA-based glycan chip platform simultaneously enables the biosynthesis of complex glycans and the analysis of glycan-involved applications. For its practical application, it is necessary to prepare a large-scale glycan chip with a large number of on-chip biosynthesized complex glycans. Thus, we are working on further studies to devise a solution for synthesizing a large number of glycans on the chip via combination with a microfluidics system containing multichannels. The introduction of a microfluidics system has the advantages of providing a large number of glycans on the chip by biosynthesizing different glycans for each channel, efficiently reusing glycan-processing enzymes, and minimally using expensive reagents (e.g., nucleotide sugars). The structure-switchable DNA-based glycan chip combined with a microfluidics system makes it possible to synthesize a variety of complex glycans from simple glycans immobilized on the chip by adjusting the combination of glycan-processing enzymes according to channels. Given the optimized reaction conditions for various enzymes, users can easily fabricate glycan chips composed of diverse complex glycans without further purification and conjugation with a specific linker. Our proposed platform would be advantageous in that it allows users to fabricate customized glycan chips (microarrays) by directly synthesizing various glycans on the chip through a combination of glycan-processing enzymes. However, this technology is still in a relatively early stage of development, and there may be questions about whether the final products are in structurally defined forms. Thus, further work will be performed to make our proposed platform a (semi)preparative scale capable of carrying out full structural characterization of the final glycan products. We anticipate that the improved platform would enable practical application of glycan chips with on-chip biosynthesized complex glycans.

In particular, the structure-switchable DNA-based glycan chip platform might be used to analyze the activities of various glycan-processing enzymes in a label-free manner on a single chip because immobilized glycans can be individually separated by structural changes in pH-responsive i-motif DNA. We anticipate

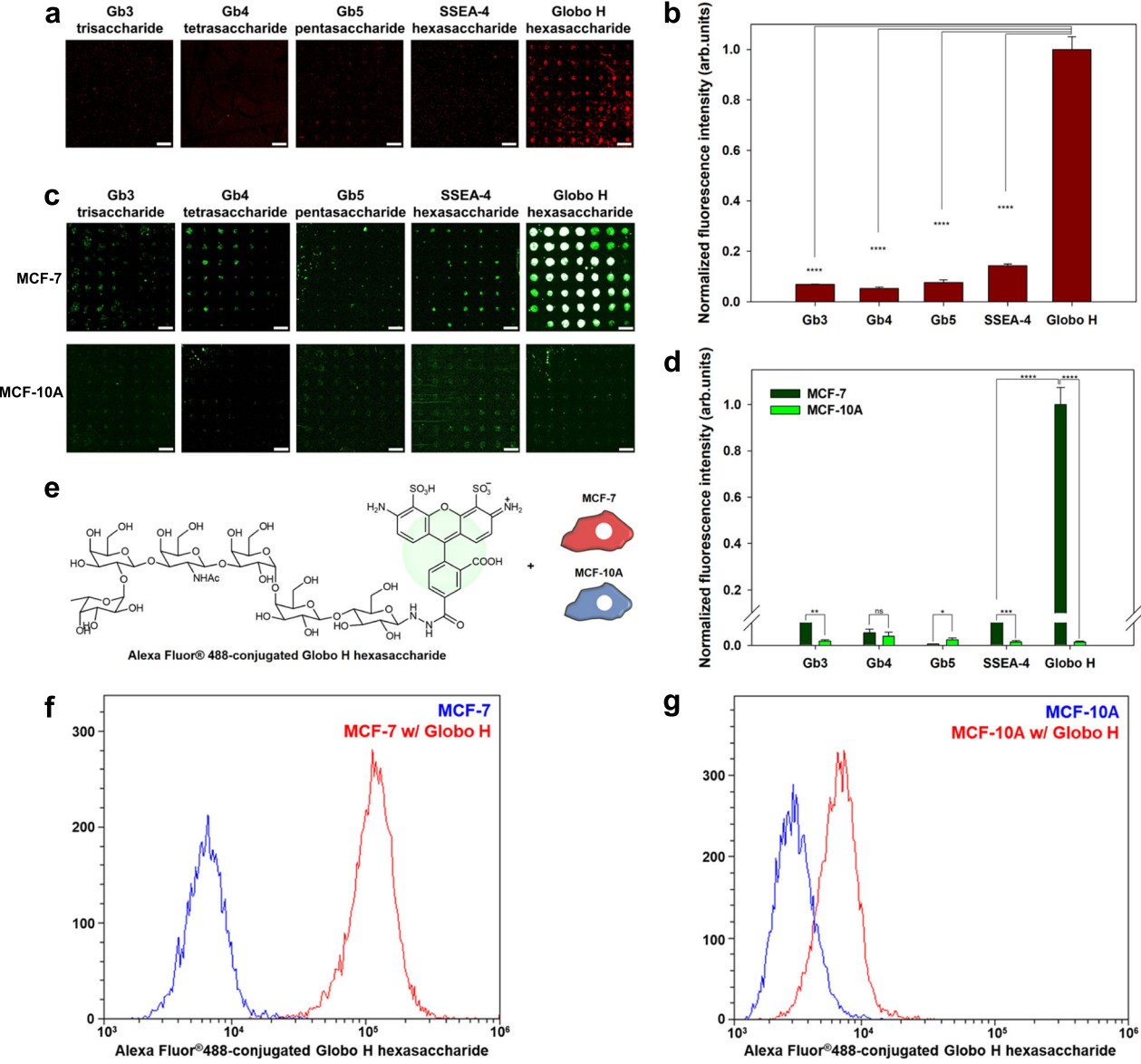

**Fig. 4 Applications of structure-switchable DNA-based glycan chips. a**, **b** Analysis of the glycan-binding specificity of the VK9 antibody on the glycan chip.
**a** Scanned raw images and (**b**) quantitative fluorescence intensity plot for the binding of VK9 with biosynthesized Gb3 trisaccharide, Gb4 tetrasaccharide,
Gb5 pentasaccharide, SSEA-4 hexasaccharide, and Globo H hexasaccharide (Scale bar: 800 μm). **c**, **d** Analysis of glycan-binding specificity of MCF-7 breast
cancer cells on the glycan chip. **c** Scanned raw images and (**d**) quantitative fluorescence intensity plot for the binding of MCF-7 breast cancer and MCF-10A
normal breast cells with on-chip biosynthesized Gb3 trisaccharide, Gb4 tetrasaccharide, Gb5 pentasaccharide, SSEA-4 hexasaccharide, and Globo H
hexasaccharide (Scale bar: 800 μm). Each value presents the mean ± SEM from forty-nine independent spots excluding the highest and lowest signals.
Statistical significance was assessed using Student's unpaired *t* test (*$p < 0.05$; **$p < 0.01$; ***$p < 0.001$; ****$p < 0.0001$). **e** Schematic presentation for
analyzing the binding of MCF-7 breast cancer and MCF-10A normal breast cells to Globo H hexasaccharide using flow cytometry. FACS analyses for the
binding of (**f**) MCF-7 breast cancer and (**g**) MCF-10A normal breast cells with Globo H hexasaccharide. Source data are provided as a Source Data file.

that the number of available glycan-processing enzymes is
increased by efficiently screening their activities using our
developed glycan chip platform.

In summary, we developed a glycan chip platform based on an
i-motif DNA linker with a pH-responsive structural change for
effective on-chip enzymatic glycosylation of complex glycans. The
structural change enabled us to reversibly control the immobi-
lization and separation of the biosynthesized complex glycans on
a surface. This approach also enabled optimization of the on-chip
enzymatic glycosylation reaction conditions through quantitative
analyses of the biosynthesized glycans, which can overcome the
limitations of previous on-chip glycan synthesis methods. Globo

H hexasaccharide and its related complex glycans were success-
fully biosynthesized from lactose disaccharide on the surface
using several glycosyltransferases under the optimized conditions.
This platform was strongly confirmed by additional on-chip
biosynthesized GM1 pentasaccharide and its related complex
glycans. The constructed glycan chip containing Globo H
hexasaccharide and its related complex glycans was applied to
analyze the glycan-binding specificities of antibodies and breast
cancer cells, clearly demonstrating the feasibility of a DNA-based
glycan chip with on-chip glycan biosynthesis for glycan-
biomolecule and glycan-cell interaction analyses. Therefore, we
anticipate that the developed structural switchable DNA-based

glycan chip platform will present directions for the efficient on-chip biosynthesis of complex glycans and the realization of diverse glycan-related applications.

## Methods

**Materials.** Poly(ethylene glycol) methyl ether thiol (average Mn 800), 1,4-dithiothreitol, 4-dimethylaminopyridine, 2,2-dimethoxy-2-phenylacetophenone (DMPA), uridine 5′-diphospho-$N$-acetylgalactosamine (UDP-GalNAc) disodium salt, uridine 5′-diphosphogalactose (UDP-Gal) disodium salt, biotinylated *Griffonia simplicifolia* isolectin B₄ (GS-IB₄), and Envi-Carb SPE columns were purchased from Sigma-Aldrich (St. Louis, MO, USA). Cytidine-5′-monophospho-$N$-acetylneuraminic acid (CMP-Neu5Ac) sodium salt and *Pasteurella multocida* α-2,3-sialyltransferase (α2,3-SialT) were obtained from GeneChem Inc. (Daejeon, Korea). Guanosine 5′-diphospho-β-L-fucose disodium salt (GDP-Fuc), *Neisseria meningitides* β-1,3-$N$-acetylgalactosaminyltransferase (LgtD), *Neisseria meningitides* α-1,4-galactosyltransferase (LgtC), and *Helicobacter mustelae* α-1,2-fucosyltransferase (α1,2-FucT) were purchased from Chemily Glycoscience (Atlanta, GA, USA). 1-$O$-Allyl-$D$-lactose was purchased from Carbosynth (Newbury, Berkshire, UK). PBS buffers (100 mM sodium phosphate and 1 M NaCl; pH 4.5 and pH 9.0) were prepared with ultrapure Milli-Q water (resistance > 18 MΩ cm). All thiol-modified oligonucleotides were purchased from Integrated DNA Technologies, Inc. (Coralville, IA, USA). Biotinylated *Ricinus communis* agglutinin I (RCA₁₂₀), biotinylated soybean agglutinin (SBA), biotinylated *Lotus tetragonolobus* lectin (LTL), and biotinylated *Maackia amurensis* lectin II (MAL II) were purchased from Vector Laboratories (Burlingame, CA, USA). Alexa Fluor® 647-conjugated streptavidin, Geneframe (25 μL, 1.0 × 1.0 cm), and Calcein-AM were purchased from Thermo Fisher Scientific (Waltham, MA, USA). DyLight 650-conjugated anti-Gb5 monoclonal antibody, DyLight 650-conjugated anti-SSEA-4 monoclonal antibody, and anti-Globo H monoclonal antibody (VK9) were purchased from Invitrogen (Carlsbad, CA, USA). Anti-Gb3 monoclonal antibody was purchased from Tokyo Chemical Industry Co., Ltd. (Tokyo, Japan). Alexa Fluor® 647-conjugated goat anti-mouse IgG H&L was purchased from Abcam (Cambridge, MA, USA). MCF-7 cells (ATCC® HTB-22™) and MCF-10A cells (ATCC® CRL-10317) were purchased from ATCC (Manassas, VA, USA). HyClone™ Dulbecco's modified Eagle's medium (DMEM), Illustra™ NAP-10 Column, HyClone™ Dulbecco's phosphate-buffered saline (DPBS) solution, HyClone™ fetal bovine serum (FBS), and HyClone™ penicillin–streptomycin 100X solution were purchased from GE Healthcare Life Sciences (Chicago, IL, USA). The Mammary Epithelial Cell Growth Medium Bulletkit was purchased from Lonza (Basel, Switzerland).

**Synthesis of glycan-oligonucleotide conjugates.** Lactose-oligonucleotide conjugates were synthesized by conjugating allyl lactose with thiol-modified oligonucleotides via photochemical reaction using a photoinitiator[55]. Single-stranded oligonucleotides (Supplementary Table 1) pretreated with 1,4-dithiothreitol (100 nmol, 1 equiv.), 1-$O$-allyl-$D$-lactose (100 nmol, 1 equiv.), and photoinitiator (10 nmol, DMPA) were mixed in 1 mL of deionized water. The mixture was stirred under UV light (365 nm) using a UVP Blak-Ray® XX-15 L UV bench lamp (15 W; Analytik Jena, Upland, CA, USA) for 2 h. After the reaction, the mixture was analyzed by normal-phase high-performance liquid chromatography (HPLC; Gilson, Middleton, WI, USA) using an LC-321 and OD-300 column (4.6 mm × 250 mm; PerkinElmer, Waltham, MA, USA). The sample was eluted using a linear gradient of acetonitrile (25–100% (v/v)) in 0.1 M triethylammonium acetate (pH 7.0) and detected at 260 nm with a diode array detector (UV/Vis-151 detector; Gilson). Data were acquired using the TRILUTION® LC software v 2 .1.

**Nuclear magnetic resonance (NMR) analysis of glycan-oligonucleotide conjugates.** ¹H NMR spectra (solvent D₂O) of glycan-oligonucleotide conjugates were acquired using an NMR spectrometer (500 MHz; Bruker, Karlsruhe, Germany) and TopSpin software v 3.6.2. Data are reported as follows: chemical shifts (δ ppm), multiplicity (s = singlet, d = doublet, q = quartet, m = multiplet), and coupling constants (Hz).

**Matrix-assisted laser desorption/ionization time-of-flight (MALDI-TOF) mass spectrometry (MS) analyses of glycan-oligonucleotide conjugates.** The MALDI-TOF spectrum was measured on an AXIMA LNR MALDI-TOF MS (Shimadzu, Kyoto, Japan) using 3-hydroxypyridine-2-carboxylic acid (3-HPA) as a matrix (50 mg/mL in deionized water). The dried sample was mixed with 10 μL of matrix solution directly on the MALDI target, followed by vacuum drying.

**Fabrication of the DNA-based glycan chip platform.** Glass slides (76 × 26 × 1 mm; Marienfeld GmbH & Co. KG, Lauda-Königshofen, Germany) were coated with ~100 nm thick gold and an ~10 nm titanium adhesive layer[56,57]. Glass slides were cleaned by ultrasonication in trichloroethylene, acetone, isopropyl alcohol, and pure water and dried by centrifugation. Titanium and gold films were formed on glass slides with a deposition of 1 Å/s and a chamber pressure of 3 × 10⁻⁶ mbar by an E-beam evaporator (KVE-4000; Korea Vacuum Tech, Gimpo, Korea). Prepared slide substrates were placed on the sample holder disc, and a titanium adhesion layer of 10 nm thickness was first deposited, followed by a gold layer with

a thickness of 100 nm. The film thickness was monitored using a quartz crystal microbalance (QCM; Inficon, Bad Ragaz, Switzerland). After the completion of deposition, gold-coated glass slides were rinsed with acetone and methanol.

A step-by-step protocol describing the fabrication and application of glycan chip is available via Protocol Exchange[58]. Single-stranded i-motif DNA (Supplementary Table 1) was dissolved in printing buffer (100 mM sodium phosphate, 10% (v/v) $N,N$-dimethylformamide, and 1 M NaCl; pH 4.5) to a final concentration of 70 μM. The solution was spotted on each gold slide (100 nm Au and 10 nm Ti adhesion layer) using a Microsys 5100 microarrayer (Cartesian Technologies, Ann Arbor, MI, USA) with a Chip Maker 2 pin (Telecom International, Sunnyvale, CA, USA). Array was designed using AxSys software v 1. 79. 4. 0. After incubating for 12 h under 75% humidity, the slides were treated with blocking solution (1 mM poly(ethylene glycol) methyl ether thiol in PBS buffer; pH 4.5) for 1 h to block nonspecific interactions. Next, the slides were removed from the blocking solution and rinsed with washing buffer (100 mM sodium phosphate and 1 M NaCl; pH 4.5) and deionized water. The slides were dried through centrifugation at 213 × $g$ for 3 min. Glycan immobilization was conducted by reacting the i-motif DNA-immobilized chip with 40 μL of hybridization solution (100 mM sodium phosphate and 1 M NaCl; pH 9.0) containing 1 nmol lactose-oligonucleotide conjugates for 3 h. Next, the DNA-based glycan chip was washed with 1X saline-sodium citrate (SSC) solution (150 mM NaCl and 15 mM sodium citrate; pH 7.0) with 0.2% (w/v) sodium dodecyl sulfate (SDS), 0.1X SSC solution with 0.2% (w/v) SDS, 0.1X SSC solution, and deionized water for 1 min each. After drying, the slides were stored at room temperature under vacuum until further use.

**Condition optimization for on-chip enzymatic synthesis of Globo H series.** Commercially available glycosyltransferases were used to biosynthesize Globo H hexasaccharide and its related glycans[46]. Each enzymatic glycosylation reaction was optimized by adjusting reaction times under certain concentrations of enzymes (4 mU) and nucleotide donors (10 mM) to synthesize Gb3 trisaccharide, Gb4 tetrasaccharide, Gb5 pentasaccharide, SSEA-4 hexasaccharide, and Globo H hexasaccharide from surface-immobilized lactose disaccharide in consecutive order. The slide was incubated with the enzyme solution at 37 °C for 12, 24, 48, and 72 h in a humid chamber. After the reaction, the slide was washed once with washing buffer I (137 mM NaCl, 2.7 mM KCl, 4.3 mM Na₂HPO₄, 1.4 mM KH₂PO₄, and 0.5% (v/v) Tween 20; pH 7.5) and two times with washing buffer II (137 mM NaCl, 2.7 mM KCl, 4.3 mM Na₂HPO₄, and 1.4 mM KH₂PO₄; pH 7.5) and dried by centrifugation at 213 × $g$ for 3 min. PBS buffer (pH 4.5) was dropped onto the chip where enzymatic glycosylation was carried out. After incubating for 2 h, the solutions were collected, desalted using a NAP-10 column, and evaporated. The products were analyzed by liquid chromatography (ICS-5000; Thermo Fisher Scientific) using a CarboPac PA100 column (4 mm × 250 mm; Dionex, Sunnyvale, CA, USA), isocratic elution mode with 100 mM sodium hydroxide, and an Ag/AgCl reference electrode for electrochemical detection. Data were acquired using the Chromeleon software v 7. 2. SR4. Because the Bio-LC used did not have a thermostat, an air conditioner was used to maintain a constant column temperature. The eluted sample was purified by solid-phase extraction chromatography[59]. The Envi-Carb SPE column was equilibrated in a 15 mL conical tube using 80% (v/v) acetonitrile in 0.1% (v/v) trifluoroacetic acid and ultrapure water and then spun at 60 × $g$ for 50 s. A 1 mL sample was added to the Envi-Carb column. The column was washed with 2 mL of ultrapure water, 2 mL of 25% (v/v) acetonitrile, 1 mL of ultrapure water, and 2 mL of 10 mM triethylammonium acetate (pH 7.0) sequentially. The final product was eluted with 2 mL of 25% (v/v) acetonitrile in 50 mM triethylammonium acetate (pH 7.0) and dried to remove the solvent. In addition, the slides were incubated with complexes of biotinylated GS-IB₄, RCA₁₂₀, SBA, LTL, and MAL II labeled by streptavidin-Alexa Fluor® 647 to assess the products of enzymatic reactions.

**On-chip enzymatic synthesis of Globo H hexasaccharide series from surface-immobilized lactose.** Commercially available glycosyltransferases were used for on-chip glycosylation of complex glycans[46]. The lactose-immobilized surface was divided into five different blocks using Geneframe®. For synthesis of Gb3 trisaccharide, a 25 μL solution of LgtC (4 mU), UDP-Gal (10 mM), Tris-HCl (100 mM; pH 7.0), and MgCl₂ (10 mM) was dropped into all five blocks, and the slide was incubated at 37 °C for 48 h in a humidified chamber. For synthesis of Gb4 tetrasaccharide from Gb3 trisaccharide, a 25 μL solution of LgtD (4 mU), UDP-GalNAc (10 mM), Tris-HCl (100 mM; pH 7.0), and MgCl₂ (10 mM) was dropped into four blocks of Gb3 trisaccharide-synthesized five blocks, and the slide was incubated at 37 °C for 48 h in a humidified chamber. For galactosylation (synthesis of Gb5 pentasaccharide) of Gb4 tetrasaccharide, a 25 μL solution of LgtD (4 mU), UDP-Gal (10 mM), Tris-HCl (100 mM; pH 7.0), and MgCl₂ (10 mM) was dropped into three of four Gb4 tetrasaccharide-synthesized blocks, and the slide was incubated at 37 °C for 72 h in a humidified chamber. For fucosylation (synthesis of Globo H hexasaccharide) of Gb5 pentasaccharide, a 25 μL solution of α1,2-FucT (4 mU), GDP-Fucose (10 mM), Tris-HCl (100 mM; pH 7.0), and MgCl₂ (10 mM) was dropped into one of three Gb5 pentasaccharide-synthesized blocks, and the slide was incubated at 37 °C for 48 h in a humidified chamber. For the synthesis of SSEA-4 hexasaccharide, a 25 μL solution of α2,3-SialT (4 mU), CMP-Neu5Ac (10 mM), Tris-HCl (100 mM; pH 7.5), and MgCl₂ (20 mM) was dropped into one of three Gb5 pentasaccharide-synthesized blocks, and the slide was incubated at 37 °C for 48 h in a humidified chamber. After each reaction, the slides were washed

once with washing buffer I and twice with washing buffer II. The slides were then dried by centrifugation at $213 \times g$ for 3 min. To determine whether there were unreacted starting glycans on the chip after each enzymatic reaction, antibody-binding analyses were performed using anti-ganglioside antibodies. Monoclonal antibodies and polyclonal secondary antibodies used in antibody-binding analyses were prepared by diluting to a concentration of 25 and 50 μg/mL in PBS buffer, respectively. Finally, the prepared glycan chips were applied to analyze the glycan-binding specificity of breast cancer cells as well as the interaction between the anti-Globo H antibody and Globo H glycan series. A laser scanner (GenePix® 4100 A; Molecular devices, Sunnyvale, CA, USA) was used for image acquisition and data were acquired using the GenePix Pro 7 Software.

**Fluorescence-activated cell sorting (FACS) analysis**. MCF-7 breast cancer cells were cultured in DMEM (high glucose) supplemented with 10% (v/v) heat-inactivated FBS, 100 U/mL penicillin, and 100 μg/mL streptomycin. MCF-10A normal breast cells were cultured with Mammary Epithelial Basal Medium, which contains bovine pituitary extract, hydrocortisone, human epidermal growth factor, insulin, gentamicin, and amphotericin-B. Both cell lines were incubated at 37 °C in a humidified atmosphere of 5% $CO_2$ and 95% air, and they were subcultured every 3 days. After incubation, the cells were detached and centrifuged.

To prepare dye-conjugated Globo H hexasaccharide, Alexa Fluor® 488 hydrazide (1.75 μmol, 1 equiv.) and Globo H hexasaccharide (0.88 μmol, 0.5 equiv.) were mixed in 1 mL of 100 mM PBS buffer (pH 7.0)[60]. The mixture was incubated at 37 °C for 6 h in a humidity chamber. After the reaction, the mixture was analyzed by liquid chromatography–mass spectrometry (LC–MS; Waters, Milford, MA, USA) (Supplementary Fig. 22).

To analyze cell-Globo H hexasaccharide interactions on the cell surface, both cell lines were treated with Alexa Fluor® 488-conjugated Globo H hexasaccharide in culture medium at 37 °C for 1 h in a humidified atmosphere of 5% $CO_2$ and 95% air. Each solution was centrifuged to remove the remaining dye-conjugated Globo H hexasaccharide. After washing with culture medium and DPBS, glycan-treated and nontreated cells resuspended in DPBS were placed into the wells of a noncoated 96-well plate. These cells were sorted by FACS (Beckman Coulter, Brea, CA, USA). Data were acquired and analyzed by using CytExpert software v 2.3.0 (Beckman Coulter).

**Interaction analysis of MCF-7 breast cancer and MCF-10A normal breast cells on the chip**. MCF-7 breast cancer and MCF-10A normal breast cells were cultured as described above. To analyze cell–glycan interactions on the chip, cells were stained by referring to a previously reported method[61]. Both cell lines ($4 \times 10^5$ cell/mL, the number of cells was counted by using a C-chip™) were treated with 4 nM calcein-AM in DPBS for 15 min. The solution was centrifuged to remove the remaining dye. After washing with DPBS and culture medium, dye-treated cells resuspended in cell culture medium were applied onto the glycan chip at 37 °C for 1 h in a humidified atmosphere of 5% $CO_2$ and 95% air. To remove unbound cells, the chip was washed once with cell culture medium and twice with DPBS.

To quantitatively analyze the binding of MCF-7 cancer cells to Globo H hexasaccharide on the glycan chip, the number ratio of MCF-7 cancer cells was adjusted by mixing with MCF-10A normal cells. Dye-treated cells resuspended in cell culture medium were mixed in three ratios (MCF-7 cells accounted for 100%, 50%, and 10%) and applied onto the glycan chip.

**Statistical analysis and reproducibility**. All values of quantitative data are presented as mean ± standard error of the means (SEM) in each legend. Experiments were performed with at least independent times yielding similar results and statistical significance was analyzed using Student's unpaired $t$ test. Statistical analyses and quantitative plots were performed using Microsoft Excel 2016 and SigmaPlot v 10.0. *$p < 0.05$; **$p < 0.01$; ***$p < 0.001$; ****$p < 0.0001$ were considered significant.

**Reporting summary**. Further information on research design is available in the Nature Research Reporting Summary linked to this article.

## Data availability

All data that support the findings of this study are available within the paper and Supplementary Information Files and uploaded to Figshare (https://doi.org/10.6084/m9.figshare.13634771). Source data are provided with this paper.

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

## Acknowledgements

Financial support was provided by the Basic Core Technology Development Program for the Oceans and the Polar Regions (NRF-2015M1A5A1037055) of the National Research Foundation funded by the Ministry of Science and ICT, Korea (to H.J.C. and C.S.K.) and the Basic Science Research Program (NRF-2019R1C1C1007379) through the National Research Foundation of Korea (NRF) funded by the Ministry of Education, Science and Technology, Korea (to C.S.K.).

## Author contributions

H.R.H., C.S.K., and H.J.C. designed the experiments. H.R.H., C.S.K., K.I.J., and J.H.S. performed the experiments and analyzed the data. H.R.H., C.S.K., and H.J.C. wrote the paper. H.J.C. is the principal investigator.

## Competing interests

The authors declare no competing interests.
