## [Peer Review File · Nature Communications]

Reviewers' Comments:

Reviewer #1:

Remarks to the Author:

This is an interesting approach to the development of glycan chip which allows the enzymatic glycan synthesis, immobilization and release of glycan-oligosaccharide conjugate for analysis. The immobilization of glycan-oligosaccharide is through hybridization to a structure changeable i-motif DNA in a pH dependent manner where at acidic pH the glycan-oligosaccharide is released for quantitative analysis. Compared to the traditional glycan chips or arrays, this method allows the synthesis, characterization and biological analysis to be performed together, while the traditional array requires the use of final synthetic glycan to spot on the surface. Though the traditional method also allows quantification of the glycan on the array based on the concentration of glycan used in arraying and the area of spot, this is an alternative strategy offering another way of quality control and is useful for small number of glycans as demonstrated in the work. However, a few questions should be addressed before the paper can be published. 1. What is the evidence that the release of glycan-oligosaccharide at acidic pH is complete? 2. In the enzymatic glycosylation, the product is checked with liquid chromatography. It is assumed that the byproducts and unreacted substrate will be separated at this stage to ensure purity of the product. How is this procedure applied to a large-scale chip preparation with large numbers of glycans on the chip? With appropriate description of the strengths and shortcomings of the method, the work is worth of publication in Nat. Commun.

Reviewer #2:

Remarks to the Author:

The authors describe a glycan array system involving on-chip enzymatic synthesis of glycans and subsequent binding analyses. The concept and practice of on-chip glycan synthesis has been previously described in multiple publications, including the authors' own in 2019 on the subject of on-chip synthesis of GM1. The novelty of the present method comes from the use of a pH regulated i-motif DNA linker which allows reversible immobilization of glycans on the array surface. Thus, the glycans could be eluted from the array surface (at low PH) after each step of enzymatic synthesis for analyses by HPLC to determine the yields (efficiency) of the enzymatic reactions. By this means, one can monitor the yields of the synthesized glycans on the arrays.

To demonstrate the application of the new method, the authors generated an array of five glycans of the globo-series (Gb3, Gb4, Gb5, SSEA4, Globo H hexasaccharide) together with the starting glycan lactose. Lectin binding was used to study the change of glycan structures after each on-chip glycosylation steps initially. This was followed by optimization of the reaction conditions through HPLC analyses of the DNA-linked glycans recovered from the array surface. The arrays that followed were used for binding studies of anti-Globo H antibody VK9, also human breast normal and cancer cells.

Main comments

The method of using the i-motif reversible DNA linker in glycan microarrays is of considerable interest to the field. However, there are several major issues for the authors to address

- 1 The first is the sustainability of such arrays. It seems the synthesis and the monitoring of the yields and purities of the desired glycans would need to be repeated for arrays of the same glycans to be generated in the future. Large scale (macro) rather than micro scale syntheses would seem desirable to have the same products to use in the long term.

2. It is rather surprising that the authors did not give an overview of the glycan microarray systems closely related to the method presented here, i.e. other DNA-based glycan array platforms (e.g. those of Chevlot et al., 2014; Yan et al., 2019) and other arrays of glycans synthesized 'on-chip' (e.g. the system coupled with MS described by Sabine Flitsch group)

3. The purity of the glycans on the arrays is a crucial aspect of any glycan array system and this is intended to be the main emphasis for this paper. It is however not clear to the reviewer whether the DNA-tagged glycans recovered from the array surface were subjected only to analytical HPLC for checking the glycosylation conversion efficiency or were actually purified by HPLC and quantified in some way before re-loading to the arrays for the next steps. It is noted that not all of the enzymatic reactions could be optimized to near full conversion, (e.g. the yield of Gb5 pentasaccharide is only about 80%). It would be important for the authors to clarify these points.

4. Related to the points above, is the lack of quality control data for the arrays generated. Only limited lectin binding results are shown to indicate the changes in the glycans after individual steps of glycosylation. The authors should include full data on lectin binding to all of the arrayed glycans. In addition, it would be important to include antibody binding data to show whether there are minor unreacted starting glycans on the arrays after enzyme reactions. There is a good range of anti-globo-glycan and anti-ganglioside antibodies that the authors could use for these analyses.

5. Further to the points above, there is some variability in the scanned spots presented in Figs 3 and 6. Authors should comment on the consistency or otherwise of all spots scanned. Bar charts for binding data with standard deviation would be good to present

6. There is a general lack of key references and experimental details in the manuscript, in particular for array construction and the protocols used for binding assays (see specific points below). The authors are strongly recommended to refer to and comply with the MIRAGE Glycan Array Guidelines This can be done using as supplementary material the template provided in the web page below

<https://www.beilstein-institut.de/en/projects/mirage/examples#block-4727>

7. The authors claim that this new array platform developed will present new directions for diverse application. It is however not clear what the key advantages are of this new platform compared with the existing advanced array systems of sequence-defined glycans, e.g. of the NCFG (previously CFG) covalent glycan arrays, the neoglycolipid microarray system of Imperial College Glycosciences Laboratory, and also the microarrays of the synthetic glycans of Seeberger group and chemo-enzymatically synthesized by the groups of Boons and George Peng Wang. Many of these array systems have hundreds of well characterized sequence-defined glycans in their glycan library, and some have the whole range of the globo- and ganglioside glycans. Considering the currently available array resources and the recent advances in automated solid phase enzymatic glycan synthesis, the authors should comment on the special features and future applications of the new array platform in the wider scene.

Other specific points

8. Line 112, reference should be given for the thiol-ene photochemical reaction. The lactose-DNA conjugate should be fully characterized and assigned by NMR, and also by MS.

9. Line 122, the chemistry used for covalent immobilization of the i-motif DNAs should be described here.

10. Lines 227-233, the GM1 related work doesn't fit under the heading.

11. Line 291, reaction scale should be described.

12. Line 310, more details (references) should be given on how the glass slides were coated with gold.

13. Line 324, the volume of 1nmol lactose-oligonucleotide conjugate should be given for the glycan immobilization step.

14. Line 353, the volume of enzyme solution should be given.

15. Table 1, Symbolic structures should be drawn from non-reducing end to reducing end (the same direction of the text sequence)

16. Figures S5-S8, conditions for HPLC analyses should be given in the footnote. How were the products characterised? Any MS data available?

17. In the Material and Methods sections, there is a heading Nuclear magnetic resonance (NMR) analysis of glycan-oligonucleotide conjugates. However only the lactose conjugate is investigated

by NMR.

18. The previous study cited by the authors identified a major binding protein for Globo H hexasaccharide on the cancer cell membrane; it is mentioned but not followed up.

19. L90, L95, L241; it is stated (L90) that glycans of Globo H series are overexpressed on human tumour cells, therefore, on the face of it, it is not immediately apparent why there should be interactions between breast cells (expressing Globo H glycans) and on-chip-biosynthesized Globo H glycans (L95). Some reconciliation by the authors is required at this stage and before L241 where the hypothesis that there are present on breast cancer cells specific binding proteins for Globo-series glycans. The issue of self-neutralization between glycans and glycan-binding proteins expressed on the same cells should be considered.

20. Bio-LC analyses, Figure S5-S11; the authors should comment as to the slight shifts in the retention times, especially of the major peaks, between 12h, 24h and 48h (a,b,c, respectively).

21. A Schematic presentation (flow chart of the sequential steps, including the recovery of the glycans from the chips, LC analysis, rehybridization of the glycans (purified?) on array, binding etc would be helpful

Reviewer #3:

Remarks to the Author:

In this work, Cha and co-workers developed a glycan chip for biosynthesizing glycans based on structure changeable DNA linker. Compared with reported glycan chips, the authors introduced a pH-regulated i-motif to control the immobilization and isolation of glycans synthesized. However, compared with the reported literature (Chem. Commun., 2019, 55, 71-74), I don't clearly know the novelty of this study. Moreover, the authors' conclusions were not clearly supported by convincing data. This work does not warrant the publication in Nature Communications. Here are my concerns:

1. The proposed on-chip glycans biosynthesizing strategy is not improved so much when compared with reported strategies (Chem. Commun., 2019, 55, 71-74). Just tethering an i-motif DNA linker on chips lacks novelty. Isolation and quantification of glycans can be achieved by other strategies, such as polypeptide. Except for glycan isolation, I strongly suggest the authors make a detailed introduction of the improvement in this work.

2. In Figure 2, it could be easily noticed that after rehybridization of isolated lactose-oligonucleotide conjugates, the hybridization efficiency is greatly reduced (Figure 2b vs. 2d). Therefore, more pH cycles should be tested to evaluate the efficiency of i-motif for controlling the DNA-based glycan chip platform.

3. In Figure 5, the authors tested the specific binding process of synthesized Globo H hexasaccharide and VK9 on the chip. But more experimental data should be provided. The author should measure the affinity coefficient between the synthesized Globo H hexasaccharide and VK9. In addition, the entire measurement process should be operated in human serum at 37 °C to prove its stability. (For example : Proc. Natl. Acad. Sci., 109(9), 3317-3322: Figure 2A, interaction between dimer peptide and PSD-95 protein).

4. In Figure 6, the author should conduct more experiments about the specific binding process of Globo H hexasaccharide to proteins and the strategy lacks generality. Except for MCF-7 cells, the author should utilize the proposed strategy to test more cancer cell lines, including high expression, medium expression, low expression, non-expression cell lines to verify the feasibility of the experiment.

5. The authors mentioned, "the conventional strategies are difficult to control and optimize stepwise glycosylation reactions using the current platforms, resulting in structurally undefined glycans on chip, which leads to unequable and unreliable results." Therefore, I suggest the authors should make a table to compare the synthesizing efficiency of current strategies and this work.

Other points:

1) Please provide some data to prove that the catalytic activity of the enzymatic glycosylation is independent of pH change.

- 2) More data about the comparison between cancer cells and normal cells should be provided to test the generality of the authors' methodology.
- 3) How about the cytotoxicity of the test?

Responses to Reviewers' Comments

We thank the reviewers for their efforts. We feel we've significantly improved the content of the manuscript based on the reviewers' comments.

Reviewer #1

This is an interesting approach to the development of glycan chip which allows the enzymatic glycan synthesis, immobilization and release of glycan-oligosaccharide conjugate for analysis. The immobilization of glycan-oligosaccharide is through hybridization to a structure changeable i-motif DNA in a pH dependent manner where at acidic pH the glycan-oligosaccharide is released for quantitative analysis. Compared to the traditional glycan chips or arrays, this method allows the synthesis, characterization and biological analysis to be performed together, while the traditional array requires the use of final synthetic glycan to spot on the surface. Though the traditional method also allows quantification of the glycan on the array based on the concentration of glycan used in arraying and the area of spot, this is an alternative strategy offering another way of quality control and is useful for small number of glycans as demonstrated in the work. However, a few questions should be addressed before the paper can be published.

1. *What is the evidence that the release of glycan-oligosaccharide at acidic pH is complete?*

Author response:

We thank the reviewer for the comment. Hybridization and denaturation of glycan-oligonucleotide conjugates were already analyzed by binding with glycan-specific lectins. After lactose-oligonucleotide conjugates were incubated on i-motif-immobilized surface that was pretreated with basic solution (pH 9.0), the surface showed strong fluorescence intensity when treated with fluorescent dye-labelled RCA₁₂₀ lectin (Figure 2b). However, RCA₁₂₀ did not bind to the surface after lactose-oligonucleotide-immobilized chip was treatment with acidic solution (pH 4.5). This result indicates that lactose-oligonucleotide can be separated from the chip surface by pH change. However, this result is not able to mean the complete separation of glycan-oligonucleotide conjugates. As following the reviewer's comment, to demonstrate that the separation of glycan-oligonucleotide conjugate is complete under the acidic conditions used, we used ssDNA which is conjugated with Alexa Fluor[®] 647 at 5'. When the fluorescent dye-conjugated complementary ssDNAs were incubated on i-motif-immobilized surface, the chip exhibited fluorescence intensity according to the concentration of immobilized i-motif DNA. However, the chip showed barely no fluorescence intensity even at high concentration of i-motif DNA when incubation in acidic condition (newly added Figure S5 in the revised Supporting Information). Because Alexa Fluor[®] 647 dye is pH-insensitive from pH 4 and pH 10 (newly added reference 43), the fluorescence change seemed to be due not to the inactivation of the dye by acidic condition but to the separation of the fluorescent dye-conjugated ssDNA. This result indicates that the glycan-oligonucleotide conjugate can be completely separated under the acidic conditions used. We included the related sentences in the revised manuscript (pages 7).

[Page 7, Revised Manuscript]

When the chip was incubated with RCA₁₂₀ after treatment with the pH 4.5 solution, no fluorescence was

observed (Figure 2c). To further validate the separation of glycan-oligosaccharide conjugates at acidic pH, single-stranded oligonucleotide conjugated with Alexa Fluor[®] 647 at 5' was used as a model. There was barely no fluorescence even at high concentration of i-motif DNA when treated with acidic solution (pH 4.5) after incubating Alexa Fluor[®] 647-conjugated complementary oligonucleotides onto the chip (Figure S10). Because Alexa Fluor[®] 647 dye is pH-resistant from pH 4 to pH 10,⁴³ the fluorescence change seemed to be due to the separation of the dye-conjugated oligonucleotides not to the inactivation of the dye in acidic conditions. These results indicated that lactose-oligonucleotide conjugates were completely separated from the chip surface by forming a quadruplex shape of i-motif DNA under acidic conditions.

[Newly added Figure S10]

Figure S10. (a) Hybridization of Alexa Fluor[®] 647-conjugated oligonucleotides with surface-immobilized i-motif DNAs. (b) Denaturation of Alexa Fluor[®] 647-conjugated oligonucleotides from surface-immobilized i-motif DNAs under acidic conditions (pH 4.5).

[Newly added reference]

43. Tran, K. K. & Shen, H. The role of phagosomal pH on the size-dependent efficiency of cross-presentation by dendritic cells. *Biomaterials* **30**, 643-652 (2009).

2. In the enzymatic glycosylation, the product is checked with liquid chromatography. It is assumed that the byproducts and unreacted substrate will be separated at this stage to ensure purity of the product. How is this procedure applied to a large-scale chip preparation with large numbers of glycans on the chip?

Author response:

We thank the reviewer for the good comment. We are currently working on further studies to devise solutions for preparing a large-scale glycan chip with large number of complex glycans on the chip. Among these, we are combining microfluidic system with DNA-based glycan microarray to synthesize enzymatically large number of glycans on chip (see the below supporting figure for on-going work). In preliminary study, we checked the biosyntheses of Gb3 trisaccharide and Gb4 tetrasaccharide using DNA-based glycan chip coupled with microfluidic system (unpublished data). The introduction of microfluidics system with multichannel can not only provide a large number of glycans on the chip by synthesizing different glycans for each channel, but also has the advantages of efficiently reusing glycan-processing enzymes and minimally using expensive reagents (*e.g.*, nucleotide sugars). We think that our proposed structure switchable DNA-based glycan chip can have a large number of glycans by coupling

with microfluidic system containing multichannel. We included the related sentence in the revised manuscript (page 13).

Combination of microfluidic system and DNA-based glycan chip for on-chip biosynthesis of large number of complex glycans. (a) Digital photograph of microfluidic system with two channels. (b) Fabrication of DNA-based glycan chip platform. (c) Introduction of closed microfluidic system. (d) Addition of enzyme (*e.g.*, LgtC) solution containing nucleotide sugar using microfluidic system. (e) Separation of biosynthesized glycan-oligonucleotide conjugate (*e.g.*, Gb3) using microfluidic system. (f) Bio-LC analysis of separated glycan-oligonucleotide conjugates.

[Pages 13]

These results indicated that Globo H hexasaccharide-binding proteins are present on MCF-7 breast cancer cells, which can support a previous study that identified the major binding protein for Globo H hexasaccharide on the cancer cell membrane.⁵⁴

In this work, we clearly demonstrated that the structure switchable DNA-based glycan chip platform enables to biosynthesize complex glycans and analyze glycan-involved applications on the chip, simultaneously. For its practical application, it is necessary to prepare large-scale glycan chip with large number of on-chip biosynthesized complex glycans. Thus, we are working on further studies to devise a solution for synthesizing large number of glycans on the chip by combining with microfluidics system containing multichannel. The introduction of microfluidics system can have advantages of providing large number of glycans on the chip by biosynthesizing different glycans for each channel, efficiently reusing glycan-processing enzymes, and minimally using expensive reagents (*e.g.*, nucleotide sugars). We anticipate that the structure switchable DNA-based glycan chip platform combined with microfluidics systems would enable to realize practical applications of glycan chips with on-chip biosynthesized complex glycans.

3. With appropriate description of the strengths and shortcomings of the method, the work is worth of publication in *Nat. Commun.*

Author response:

We thank the reviewer for the comment. Previously reported glycan microarray systems have used a method of synthesizing glycans chemo-enzymatically and then conjugating with linker to immobilize them on the surface (refs 4-8). This process seems to be not suitable for complex

glycans such as SSEA4 and Globo H hexasaccharides. This is because these glycans have low synthesis yield compared to simple glycans and the conjugation with the linkers has limitations of being capable of eliminating labile sialic acid and occurring significant loss (newly added refs. 12-14). However, our proposed glycan microarray platform can improve the limitations by immobilizing structurally simple disaccharide with high synthetic efficiency and then synthesizing complex glycans on the chip using glycosyltransferases. Structural change of pH-responsive i-motif DNA enables to isolate glycan-oligonucleotide from surface and optimize on-chip glycan biosynthesis reactions, resulting in the synthesis of structurally defined complex glycans on the chip. In addition, our proposed on-chip glycan biosynthesis have a great advantage of using small amounts of expensive nucleotide glycans and glyco-processing enzymes for complex glycan synthesis. We clearly demonstrated that our proposed system enables to synthesize five Globo H series from immobilized lactose disaccharide and analyze glycan-involved interactions on chip simultaneously. Therefore, our structure switchable DNA-based glycan chip platform would be superior to other platforms when fabricating glycan microarray consisting of complex glycans that require the use of expensive glycan-processing enzymes and nucleotide sugars due to their low synthetic efficiency by chemical methods. We included the related sentences in the revised manuscript (pages 3 & 5) with newly added references (12-14).

[Page 3]

Glycan chips have been developed in response to the essential need for the high-throughput analysis of glycan-involved interactions. Currently, glycan chips have been employed in screening therapeutic agents and profiling glycan-involved interactions including glycan-lectin, glycan-cytokine, glycan-antibody, and glycan-virus/bacteria.⁴⁻⁸ However, studies on glycan-related biological processes using glycan chips are still at an early stage relative to our knowledge of the biological functions of proteins and genes. To construct glycan chips, most homogeneous glycans are commonly provided by either chemical synthesis or natural purification using multiphase chromatography. These methods have several limitations, including labor-intensive, costly, and time-consuming processes, the requirement of protection and deprotection steps, and the difficulty of controlling the stereochemistry of glycosidic linkages, resulting in poor purity and low stepwise yield.⁹⁻¹¹ The limitation of access to glycan libraries with diverse structures restricts extensive studies on their roles *in vivo* using glycan chips. In addition, current glycan chip platforms have used a method of conjugating chemo-enzymatically synthesized glycans with linkers to immobilize them on surface.⁴⁻⁸ This method would be unsuitable for complex glycans because these have significantly low synthesis yield compared to simple glycans and the conjugation with the linkers has limitations of being capable of eliminating labile sialic acid and occurring significant loss.¹²⁻¹⁴

Considering that on-chip syntheses of oligonucleotides and peptides have been successfully utilized for genomics and proteomics,¹⁵⁻¹⁷ on-chip enzymatic glycan synthesis would be an attractive tool for glycomics. This method has several merits over conventional methods, including a low-cost and simple process without any additional protection/deprotection, purification, and immobilization steps, the use of small amounts of expensive glycan-processing enzymes and nucleotide sugar donors, the synthesis of many glycosidic linkages in a straightforward manner, and the direct application of the synthesized glycans for glycomics.

[Page 5]

As shown in Figure 1, conjugates of glycan and complementary single-stranded oligonucleotide can hybridize to i-motif DNAs that are immobilized on the chip surface under slightly basic conditions. As the pH is lowered, i-motif DNA tends to strongly form a quadruplex structure, resulting in the denaturation of the DNA double helix and thereby enabling the isolation of glycan-oligonucleotide conjugates from the surface. This property would make possible to optimize on-chip enzymatic glycosylation by analyzing isolated glycan-oligonucleotide conjugates using liquid chromatography, resulting in synthesis of structurally defined complex glycans on chip. Therefore, our proposed glycan

chip platform could improve the limitation of current platforms by immobilizing structurally simple disaccharide with high synthesis yield and then synthesizing complex glycans on the chip using glycosyltransferases under optimized conditions.

To check the feasibility of on-chip enzymatic glycosylation using the structure switchable DNA-based glycan chip platform, Globo H series (Table 1) were selected as target complex glycans, which are aberrantly overexpressed on human tumor cells and known to be involved in tumor progression.^{36,37}

[Newly added references]

12. Esposito, D., Hurevich, M., Castagner, B., Wang, C. –C & Seeberger, P. H. Automated synthesis of sialylated oligosaccharide. *Beilstein J. Org. Chem.* **8**, 1601–1609 (2012).

13. Fair, R. J., Hahm H. S. & Seeberger, P. H. Combination of automated solid-phase and enzymatic oligosaccharide synthesis provides access to $\alpha(2,3)$ -sialylated glycans. *Chem. Comm.* **51**, 6183-6185 (2015).

14. Song, X. et al. Oxidative release of natural glycans for functional glycomics. *Nat. Methods* **13**, 528-536 (2016).

Reviewer #2

The authors describe a glycan array system involving on-chip enzymatic synthesis of glycans and subsequent binding analyses. The concept and practice of on-chip glycan synthesis has been previously described in multiple publications, including the authors' own in 2019 on the subject of on-chip synthesis of GM1. The novelty of the present method comes from the use of a pH regulated i-motif DNA linker which allows reversible immobilization of glycans on the array surface. Thus, the glycans could be eluted from the array surface (at low PH) after each step of enzymatic synthesis for analyses by HPLC to determine the yields (efficiency) of the enzymatic reactions. By this means, one can monitor the yields of the synthesized glycans on the arrays.

To demonstrate the application of the new method, the authors generated an array of five glycans of the globo-series (Gb3, Gb4, Gb5, SSEA4, Globo H hexasaccharide) together with the starting glycan lactose. Lectin binding was used to study the change of glycan structures after each on-chip glycosylation steps initially. This was followed by optimization of the reaction conditions through HPLC analyses of the DNA-linked glycans recovered from the array surface. The arrays that followed were used for binding studies of anti-Globo H antibody VK9, also human breast normal and cancer cells.

Main comments

The method of using the i-motif reversible DNA linker in glycan microarrays is of considerable interest to the field. However, there are several major issues for the authors to address.

1 The first is the sustainability of such arrays. It seems the synthesis and the monitoring of the yields and purities of the desired glycans would need to be repeated for arrays of the same glycans to be generated in the future. Large scale (macro) rather than micro scale syntheses would seem desirable to have the same products to use in the long term.

Author response:

We thank the reviewer for the good comment. We are currently working on further studies to

devise solutions for preparing a large-scale glycan chip with large number of complex glycans on the chip. Among these, we are combining microfluidic system with DNA-based glycan microarray to synthesize enzymatically large number of glycans on chip (see the below supporting figure for on-going work). In preliminary study, we checked the biosyntheses of Gb3 trisaccharide and Gb4 tetrasaccharide using DNA-based glycan chip coupled with microfluidic system (unpublished data). The introduction of microfluidics system with multichannel can not only provide a large number of glycans on the chip by synthesizing different glycans for each channel, but also has the advantages of efficiently reusing glycan-processing enzymes and minimally using expensive reagents (*e.g.*, nucleotide sugars). We think that our proposed structure switchable DNA-based glycan chip can have a large number of glycans by coupling with microfluidic system containing multichannel. We included the related sentence in the revised manuscript (page 13).

Combination of microfluidic system and DNA-based glycan chip for on-chip biosynthesis of large number of complex glycans. (a) Digital photograph of microfluidic system with two channels. (b) Fabrication of DNA-based glycan chip platform. (c) Introduction of closed microfluidic system. (d) Addition of enzyme (*e.g.*, LgtC) solution containing nucleotide sugar using microfluidic system. (e) Separation of biosynthesized glycan-oligonucleotide conjugate (*e.g.*, Gb3) using microfluidic system. (f) Bio-LC analysis of separated glycan-oligonucleotide conjugates.

[Page 13]

These results indicated that Globo H hexasaccharide-binding proteins are present on MCF-7 breast cancer cells, which can support a previous study that identified the major binding protein for Globo H hexasaccharide on the cancer cell membrane.⁵⁴

In this work, we clearly demonstrated that the structure switchable DNA-based glycan chip platform enables to biosynthesize complex glycans and analyze glycan-involved applications on the chip, simultaneously. For its practical application, it is necessary to prepare large-scale glycan chip with large number of on-chip biosynthesized complex glycans. Thus, we are working on further studies to devise a solution for synthesizing large number of glycans on the chip by combining with microfluidics system containing multichannel. The introduction of microfluidics system can have advantages of providing large number of glycans on the chip by biosynthesizing different glycans for each channel, efficiently reusing glycan-processing enzymes, and minimally using expensive reagents (*e.g.*, nucleotide sugars). We anticipate that the structure switchable DNA-based glycan chip platform combined with microfluidics systems would

enable to realize practical applications of glycan chips with on-chip biosynthesized complex glycans.

2. *It is rather surprising that the authors did not give an overview of the glycan microarray systems closely related to the method presented here, i.e. other DNA-based glycan array platforms (e.g. those of Chevolut et al., 2014; Yan et al., 2019) and other arrays of glycans synthesized 'on-chip' (e.g. the system coupled with MS described by Sabine Flitsch group).*

Author response:

We thank the reviewer for the good comment. As following the reviewer's comment, we included an overview of previous other DNA-based glycan array platforms and other arrays related with on-chip glycan synthesis in the revised manuscript (pages 4-5) with newly added references (18-21 & 27)

[Pages 4-5]

DNA hybridization-based immobilization method has been used to immobilize glycans for fabrication of glycan chip.²¹⁻²⁶ The double-stranded DNA (dsDNA) linkers act as a rigid arm, allowing better presentation of glycans and glycoclusters on surface to media.^{21-23,27} They also permit tailoring spatial arrangement.^{21,22,27} These performance leads to glycoside cluster effect,^{21-23,25,27,28} which is key factor for glycan-protein interactions.^{24,26,29,30} In addition, DNA linkers were used as identifier to code individual glycans binding to glycan-binding proteins and cell by conjugating individual oligonucleotide according to glycans.³¹ With these merits, DNA-based glycan chip coupled with mass spectrometry showed the possibility of on-chip glycan biosynthesis by analyzing activities of glycan-processing enzymes on the chip.²⁴

[Newly added references]

21. Chevolut Y. et al. DNA directed immobilization glycocluster array: applications and perspectives. *Curr. Opin. Chem. Biol.* **18**, 46-54 (2014).

23. Zhang, J. et al. DNA-directed immobilization of glycomimetics for glycoarrays application: Comparison with covalent immobilization, and development of an on-chip IC50 measurement assay. *Biosens. Bioelectron.* **24**, 2515-2521 (2009).

24. van Munster, J. M. et al. Application of carbohydrate arrays coupled with mass spectrometry to detect activity of plant-polysaccharide degradative enzymes from the fungus *Aspergillus niger*. *Sci. Rep.* **7**, 43117 (2017).

25. Morvan, F., Vidal, S. Souteyrand, E., Chevolut, Y. & Vasseur J. -J. DNA glycoclusters and DNA-based carbohydrate microarrays: From design to applications. *RSC Adv.* **2**, 12043-12068 (2012).

31. Yan, M. et al. Next-generation glycan microarray enabled by DNA-coded glycan library and next-generation sequencing technology. *Anal. Chem.* **91**, 9221-9228 (2019).

3. *The purity of the glycans on the arrays is a crucial aspect of any glycan array system and this is intended to be the main emphasis for this paper. It is however not clear to the reviewer whether the DNA-tagged glycans recovered from the array surface were subjected only to analytical HPLC for checking the glycosylation conversion efficiency or were actually purified by HPLC and quantified in some way before re-loading to the arrays for the next steps. It is noted that not all of the enzymatic reactions could be optimized to near full conversion, (e.g. the yield of Gb5 pentasaccharide is only about 80%). It would be important for the authors to clarify these points.*

Author response:

We thank the reviewer for the good comment. In this work, the main objective of isolating glycan-oligonucleotide conjugates from the surface was to check the enzymatic glycosylation conversion efficiency for optimization of enzymatic reaction conditions. After optimizing the

enzymatic glycosylation conditions, the freshly synthesized lactose-oligonucleotide conjugate-immobilized chip was separated into five blocks, and five Globo H glycan series were biosynthesized on the chip through the enzymatic reactions under the optimized conditions. To make readers to understand clearly, we included the schematic representation of constructing DNA-based complex glycan chip in the revised Figure 3a.

Actually, we used a reaction time of 72 h to synthesize Gb5 pentasaccharide from Gb4 tetrasaccharide on the chip using LgtD enzyme, but its conversion efficiency was not calculated in the original manuscript because all conversion efficiencies of other enzymes were calculated only until 48 h. As following the reviewer's comment, we calculated the conversion efficiency of LgtD enzyme at the reaction time 72 h and included the data in the revised manuscript (revised Figure 3b) and the Supporting Information (revised Figure S14).

Figure S14. Bio-LC analyses of on-surface biosynthesized Gb5 pentasaccharide from Gb4 tetrasaccharide by LgtD glycosyltransferase for (a) 12 h, (b) 24 h, (c) 48 h, and (d) 72 h. Symbols: blue circle, Glc; yellow circle, Gal; yellow square, GalNAc.

4. Related to the points above, is the lack of quality control data for the arrays generated. Only limited lectin binding results are shown to indicate the changes in the glycans after individual steps of glycosylation. The authors should include full data on lectin binding to all of the arrayed glycans. In addition, it would be important to include antibody binding data to show whether there are minor unreacted starting glycans on the arrays after enzyme reactions. There is a good range of anti-glyco-glycan and anti-ganglioside antibodies that the authors could use for these analyses.

Author response:

We thank the reviewer for the good comment. As following the reviewer's comment, we presented full data on lectin binding to all of the arrayed glycans in the revised Figure 2.

In addition, to check whether there are minor unreacted starting glycans on the arrays after enzymatic glycosylations, we also performed antibody binding analyses for products of enzymatic reactions. Due to absence of commercially available anti-lactose and anti-Gb4 antibodies, it was impossible to confirm that all glycosylation processes were completed. However, using commercially available anti-Gb5, anti-SSEA-4, and anti-Globo H antibodies, we confirmed that there were no fluorescence when the antibodies against starting glycans were treated on the chip after on-chip enzymatic syntheses of Gb4 tetrasaccharide, SSEA-4 hexasaccharide, and Globo H hexasaccharide. These results clearly indicated that the on-chip enzymatic glycosylations completely proceeded under the conditions used. We included the related sentence and data in revised manuscript (page 11 & newly added Figure 3c-d) and the revised Supporting Information (newly added Figure S17).

[Revised Figure 2]

Figure 2. Scanned raw images and quantitative intensity plots for on-chip enzymatic glycosylations of (a) Gb3 trisaccharide, (b) Gb4 tetrasaccharide, (c) Gb5 pentasaccharide, (d) SSEA-4 hexasaccharide, and (e) Globo H hexasaccharide. Synthesized complex glycans were detected by using biotinylated lectins and Alexa Fluor® 647-conjugated streptavidin. Each value is the mean of forty-nine independent spots, and the error bars represent the standard deviation. Abbreviations: GS-IB₄, *Griffonia simplicifolia* isolectin B₄; SBA, Soybean agglutinin lectin; RCA₁₂₀, *Ricinus communis agglutinin* I lectin; MAL II, *Maackia amurensis* Lectin II; LTL, *Lotus tetragonolobus* lectin; LgtC, α -1,4-galactosyltransferase; LgtD, β -1,3-N-acetylgalactosaminyltransferase/ β -1,3-galactosyltransferase; α 2,3-SialT, α -2,3-sialyltransferase; α 1,2-FucT, α -1,2-fucosyltransferase. Symbols: blue circle, Glc; yellow circle, Gal; yellow square, GalNAc; red triangle, Fuc; purple square, Neu5Ac.

[Page 11]

To check whether there are minor unreacted starting glycans on the chip after enzymatic reactions under optimized conditions, we analyzed antibody binding to spots where immobilized glycans reacted with glycosyltransferases (Figure 3a). Due to absence of commercially available anti-lactose and anti-Gb4 antibodies, it was impossible to confirm that all glycosylation processes were completed. However, using anti-Gb5, anti-SSEA-4, and anti-Globo H antibodies, we confirmed that there were no fluorescence when the antibodies against starting glycans were treated on the chip after on-chip enzymatic syntheses of Gb4 tetrasaccharide, SSEA-4 hexasaccharide, and Globo H hexasaccharide (Figures 3c-d & S17). These results clearly indicated that the on-chip enzymatic glycosylations completely proceeded under the conditions used.

[Newly added Figures 3c & 3d]

Figure 3. Scanned raw images for on-chip enzymatic glycosylations of (c) SSEA-4 hexasaccharide and

(d) Globo H hexasaccharide under optimized conditions. Synthesized complex glycans were detected by using DyLight 650-conjugated monoclonal antibodies or monoclonal antibody with Alexa Fluor® 647-conjugated polyclonal secondary antibody. Abbreviations: Anti-Gb5 Ab, DyLight 650-conjugated anti-Gb5 monoclonal antibody; Anti-SSEA-4 Ab, DyLight 650-conjugated anti-SSEA-4 monoclonal antibody; Anti-Globo H Ab, Anti-Globo H monoclonal antibody (VK9); α 2,3-SialT, α -2,3-sialyltransferase; α 1,2-FucT, α -1,2-fucosyltransferase. Symbols: blue circle, Glc; yellow circle, Gal; yellow square, GalNAc; red triangle, Fuc; purple square, Neu5Ac.

[Newly added Figure S17]

Figure S17. Scanned raw images for on-chip enzymatic glycosylations of (a) Gb3 trisaccharide, (b) Gb4 tetrasaccharide, and (c) Gb5 pentasaccharide under optimized conditions. Synthesized complex glycans were detected by using DyLight 650-conjugated monoclonal antibody or monoclonal antibody with Alexa Fluor® 647-conjugated polyclonal secondary antibody. Abbreviations: Anti-Gb3 Ab, Anti-Gb3 monoclonal antibody; Anti-Gb5 Ab, DyLight 650-conjugated anti-Gb5 monoclonal antibody; LgtC, α -1,4-galactosyltransferase; LgtD, β -1,3-N-acetylgalactosaminyltransferase/ β -1,3-galactosyltransferase; N.D., not determined due to non-availability of commercial antibody. Symbols: blue circle, Glc; yellow circle, Gal; yellow square, GalNAc.

5. Further to the points above, there is some variability in the scanned spots presented in Figs 3 and 6. Authors should comment on the consistency or otherwise of all spots scanned. Bar charts for binding data with standard deviation would be good to present.

Author response:

We thank the reviewer for the good comment. As following the reviewer’s comment, we included quantitative intensity bar plot with standard deviation in the revised Figure 2 (from the original Figure 3). We also revised the quantitative intensity bar plots with standard deviation in the original Figures 5 and 6 using the all forty-nine spots and included them in the revised Figure 4.

[Revised Figure 4a-d]

Figure 4. Applications of structure switchable DNA-based glycan chip. (a,b) Analysis of glycan binding specificity of VK9 antibody on the glycan chip. (a) Scanned raw images and (b) quantitative fluorescence intensity plot for the binding of VK9 with biosynthesized Gb3 trisaccharide, Gb4

tetrasaccharide, Gb5 pentasaccharide, SSEA-4 hexasaccharide, and Globo H hexasaccharide. (c,d) Analysis of glycan-binding specificity of MCF-7 breast cancer cells on the glycan chip. (c) Scanned raw images and (d) quantitative fluorescence intensity plot for the binding of MCF-7 breast cancer and MCF-10A breast normal cells with on-chip biosynthesized Gb3 trisaccharide, Gb4 tetrasaccharide, Gb5 pentasaccharide, SSEA-4 hexasaccharide, and Globo H hexasaccharide. Each value is the mean of forty-nine independent spots, and the error bars represent the standard deviation.

6. There is a general lack of key references and experimental details in the manuscript, in particular for array construction and the protocols used for binding assays (see specific points below). The authors are strongly recommended to refer to and comply with the MIRAGE Glycan Array Guidelines This can be done using as supplementary material the template provided in the web page below <https://www.beilstein-institut.de/en/projects/mirage/examples#block-4727>

Author response:

We thank the reviewer for the good comment. As following the MIRAGE Glycan Array Guidelines, we included experimental details for array construction and protocols for binding assays in a newly added Supporting Information.

[Newly added supplementary material]	
Supplementary Glycan Chip DocumentBased on MIRAGE Guidelines	
Classification	Guidelines
1. Samples: Glycan Binding Sample	
Description of Sample	Figures 1 & 2 Biotinylated Ricinus communis agglutinin I (RCA₁₂₀), biotinylated soybean agglutinin (SBA), biotinylated Lotus tetragonolobus lectin (LTL), and biotinylated Maackia amurensis lectin II (MAL II) were purchased from Vector Laboratories (Burlingame, CA, USA). Biotinylated Griffonia simplicifolia isolectin B4 (GS-IB₄) was purchased from Sigma-Aldrich (St. Louis, MO, USA). Alexa Fluor® 647-conjugated streptavidin was purchased from Thermo Fisher Scientific (Waltham, MA, USA).  • RCA₁₂₀/Vector Laboratories/B-1085 • SBA/Vector Laboratories/B-1015 • LTL/Vector Laboratories/B-1325 • MAL II/Vector Laboratories/B-1265 • GS-IB₄/Sigma-Aldrich/L2140 • Streptavidin, Alexa Fluor™ 647 conjugate/Thermo Fisher Scientific/S21374 Figures 3, 4 & S17 Anti-Gb3 monoclonal antibody was purchased from Tokyo Chemical Industry Co., Ltd (Tokyo, Japan). DyLight 650-conjugated anti-Gb5 antibody, DyLight 650-conjugated anti-SSEA-4

	monoclonal antibody, and anti-Globo H monoclonal antibody (VK9) were purchased from Invitrogen (Carlsbad, CA, USA). Alexa Fluor® 647-conjugated goat anti-mouse IgG H&L was purchased from Abcam (Cambridge, MA, USA).  • Anti-Gb3 monoclonal antibody/Tokyo Chemical Industry/A2506 • Anti-Gb5 monoclonal antibody (MC-631), DyLight 650/Invitrogen/MA1-020-D650 • Anti-SSEA-4 monoclonal antibody (MC-813-70), DyLight 650/Invitrogen/ MA1-021-D650 • Anti-Globo H monoclonal antibody (VK9)/Invitrogen/14-9700-82 • Goat anti-mouse IgG H&L (Alexa Fluor® 647)/Abcam/ab150115 Figure 4 MCF-7 cells (ATCC® HTB-22™) and MCF-10A cells (ATCC® CRL-10317) were purchased from ATCC (Manassas, VA, USA).  • MCF7/ATCC/HTB-22/Organism: Homo sapiens, human/Cell Type: epithelial/Tissue: mammary gland, breast; derived from metastatic site: pleural effusion /Disease: adenocarcinoma • MCF10A/ATCC/CRL-10317/Organism: Homo sapiens, human/Cell Type: epithelial /Tissue: mammary gland; breast/Disease: fibrocystic disease • MCF-7 breast cancer cells were cultured in Dulbecco's Modified Eagles Medium (DMEM, High glucose) (HyClone™, SH30243.01) supplemented with 10% (v/v) heat-inactivated FBS, 100 U/mL penicillin, and 100 µg/mL streptomycin. MCF-10A breast normal cells were cultured with Mammary Epithelial Basal Medium (MEBM) (Lonza, CC-3151), which contains bovine pituitary extract, hydrocortisone, human epidermal growth factor, insulin, gentamicin, and amphotericin-B. Both cells were incubated at 37 °C in a humidified atmosphere of 5% CO₂ and 95% air, and they were subcultured every 3 days.
Sample modifications	Not relevant.
Assay protocol	Please see method section in the main text.
2. Glycan Library	
Glycan description for defined glycans	DNA-based glycan chip consists of 6 defined

	glycans (Table 1). The enzymatic synthesis of the glycans are described.
Glycan description for undefined glycans	No glycans are undefined.
Glycan modifications	No modification after lactose-single stranded oligonucleotide conjugates were made.
3. Printing Surface; e.g., Chip Slide	
Description of surface	Gold-coated glass slide
Manufacturer	Glass slide (Marienfeld GmbH & Co. KG)
Custom preparation of surface	Please see method section in the main text.
Non-covalent immobilization	All glycans are conjugated with single-stranded oligonucleotide. Complementary i-motif DNA was modified with thiolated T10 at 5' to immobilize them onto gold-coated glass slide.
4. Arrayer (Printer)	
Description of Arrayer	Microsys 5100 microarrayer (Cartesian Technologies)
Dispensing mechanism	ArrayIt Brand Products, ChipMaker CMP4
Glycan deposition	Manufacturer estimation is 1 nL per spot. However, actual delivery volume of each printed spot is not determined. i-motif was “pre-spotted” 10-times on plain glass slide before being spotted on gold-coated slides. Each array contains 49 replicate spots for each individual glycan.
Printing conditions	i-motif DNA was diluted 70 μ M in 100 mM sodium phosphate (pH 4.5) containing 10% (v/v) N,N -dimethylformamide and 1 M NaCl. 20 μ L of i-motif DNA was transferred to a 96-well microtiter plate and printed at ambient temperature and relative humidity of 75%.
5. Glycan Chip with “MAP”	
Array layout	For analyzing on-chip enzymatic glycosylation of cancer-associated complex glycans, each slide consists of 2 \times 4 (8) subarray pattern with each subarray containing 7 \times 7 features. Array Layout file = “For analyzing on-chip enzymatic glycosylation of cancer-associated complex glycans.GAL” For interaction analyses of cells and VK9 antibody with on-chip synthesized Globo H-related complex glycans, each slide consists of 2 \times 5 (10) subarray pattern with each subarray containing 7 \times 7 features. Array Layout file = “Interaction analyses of cells

	and antibody with on-chip biosynthesized Globo H-related complex glycans.GAL”
Glycan identification and quality control	Glycans biosynthesized on chip were assessed by using Bio-LC (Figures 3, S12-S16 & S18-S20), lectin binding (Figure 2), and antibody binding (Figures 3, 4 & S17). Used lectins were GS-IB ₄ , SBA, RCA ₁₂₀ , MAL II, and LTL to monitor enzymatic glycosylations. Used antibodies were anti-Gb3, anti-Gb5, anti-SSEA-4, and anti-Globo H antibodies to check whether there are minor unreacted starting glycan after each enzymatic reaction under optimized conditions used.
6. Detector and Data Processing	
Scanning hardware	GenePix® 4100A
Scanning settings	Scanning resolution: 10 µm resolution, full scan area Laser channel: 635 PMT Voltages: Adjusted for each sample to achieve maximum signal without saturation of any single spot. Scan power: Adjusted for each sample to achieve maximum signal without saturation of any single spot.
Image analysis software	GenePix Pro 7 Software
Data processing	Output .txt files containing calculated data were processed in MS Excel to determine the mean signal value of 49 replicate spots.
7. Glycan Chip Data Presentation	
Data presentation	The chip binding results are in Figures 1, 2, 3 & 4 and Supplementary Figures S17, S21, & S23. Binding results in Figures 2, 3, 4, and Supplementary Figures S21 and S23 are presented as 2D bar graphs with bars representing averaged mean signal of each glycan and error bars representing standard deviation.
8. Interpretation and Conclusion from Chip Data	
Data interpretation	No software or algorithms were used to interpret processed data.
Conclusions	Lectin binding results confirmed that the Globo H hexasaccharide series was successfully biosynthesized on lactose disaccharide-immobilized surface (Figure 2). VK9 antibody exhibited high binding affinity to Globo H hexasaccharide biosynthesized on the chip (Figure 4), indicating that Globo H hexasaccharide was successfully biosynthesized

	on lactose disaccharide-immobilized surface. MCF-7 cancer cells strongly recognized Globo H hexasaccharide on the chip, while MCF-10A breast normal cells did not (Figure 4). This indicated DNA-based glycan chip combined with on-chip enzymatic glycosylation could be used to screen cancer cell-specific glycans.
--	--

7. *The authors claim that this new array platform developed will present new directions for diverse application. It is however not clear what the key advantages are of this new platform compared with the existing advanced array systems of sequence-defined glycans, e.g. of the NCFG (previously CFG) covalent glycan arrays, the neoglycolipid microarray system of Imperial College Glycosciences Laboratory, and also the microarrays of the synthetic glycans of Seeberger group and chemo-enzymatically synthesized by the groups of Boons and George Peng Wang. Many of these array systems have hundreds of well characterized sequence-defined glycans in their glycan library, and some have the whole range of the globo- and ganglioside glycans. Considering the currently available array resources and the recent advances in automated solid phase enzymatic glycan synthesis, the authors should comment on the special features and future applications of the new array platform in the wider scene.*

Author response:

We thank the reviewer for the good comment. Previously reported glycan microarray systems have used a method of synthesizing glycans chemo-enzymatically and then conjugating with linker to immobilize them on the surface (refs 4-8). This process seems to be not suitable for complex glycans such as SSEA4 and Globo H hexasaccharides. This is because these glycans have low synthesis yield compared to simple glycans and the conjugation with the linkers has limitations of being capable of eliminating labile sialic acid and occurring significant loss (newly added refs. 12-14). However, our proposed glycan microarray platform can improve the limitations by immobilizing structurally simple disaccharide with high synthetic efficiency and then synthesizing complex glycans on the chip using glycosyltransferases. Structural change of pH-responsive i-motif DNA enables to isolate glycan-oligonucleotide from surface and optimize on-chip glycan biosynthesis reactions, resulting in the synthesis of structurally defined complex glycans on the chip. In addition, our proposed on-chip glycan biosynthesis have a great advantage of using small amounts of expensive nucleotide glycans and glyco-processing enzymes for complex glycan synthesis. We clearly demonstrated that our proposed system enables to synthesize five Globo H series from immobilized lactose disaccharide and analyze glycan-involved interactions on chip simultaneously. Therefore, our structure switchable DNA-based glycan chip platform would be superior to other platforms when fabricating glycan microarray consisting of complex glycans that require the use of expensive glycan-processing enzymes and nucleotide sugars due to their low synthetic efficiency by chemical methods.

However, it is necessary to prepare large-scale glycan chip with large number of glycans on the chip for practical application of our developed platform. As explained earlier, we are currently working on further studies to devise a solution for enzymatically synthesizing large number of glycans on the chip by combining with microfluidics system (see the below supporting figure for on-going work). In preliminary study, we checked the biosyntheses of Gb3 trisaccharide and Gb4 tetrasaccharide using DNA-based glycan chip coupled with microfluidic system (unpublished

data). We think that our proposed structure switchable DNA-based glycan chip can have a large number of glycans by coupling with microfluidic system containing multichannel.

An increase in the number of available glycan-processing enzymes can contribute to the improved synthetic efficiency of complex glycans and the synthesis of new glycans. Thus, it is very important to screen the activity of a large number of new glycan-processing enzymes for synthesis of diverse complex glycans. Our proposed glycan chip platform can be successfully used to analyze the activity of various glycan-processing enzymes in a label-free manner on a single chip because immobilized glycans can be individually separated by structural change of pH-responsive i-motif DNA. We hope to increase the number of available glycan-processing enzymes by screening their activities using our structure switchable DNA-based glycan chip platform.

We included the related sentences in the revised manuscript (pages 3, 5, 13) with newly added references (12-14).

[Page 3]

Glycan chips have been developed in response to the essential need for the high-throughput analysis of glycan-involved interactions. Currently, glycan chips have been employed in screening therapeutic agents and profiling glycan-involved interactions including glycan-lectin, glycan-cytokine, glycan-antibody, and glycan-virus/bacteria.⁴⁻⁸ However, studies on glycan-related biological processes using glycan chips are still at an early stage relative to our knowledge of the biological functions of proteins and genes. To construct glycan chips, most homogeneous glycans are commonly provided by either chemical synthesis or natural purification using multiphase chromatography. These methods have several limitations, including labor-intensive, costly, and time-consuming processes, the requirement of protection and deprotection steps, and the difficulty of controlling the stereochemistry of glycosidic linkages, resulting in poor purity and low stepwise yield.⁹⁻¹¹ The limitation of access to glycan libraries with diverse structures restricts extensive studies on their roles *in vivo* using glycan chips. In addition, current glycan chip platforms have used a method of conjugating chemo-enzymatically synthesized glycans with linkers to immobilize them on surface.⁴⁻⁸ This method would be unsuitable for complex glycans because these have significantly low synthesis yield compared to simple glycans and the conjugation with the linkers has limitations of being capable of eliminating labile sialic acid and occurring significant loss.¹²⁻¹⁴

Considering that on-chip syntheses of oligonucleotides and peptides have been successfully utilized for genomics and proteomics,¹⁵⁻¹⁷ on-chip enzymatic glycan synthesis would be an attractive tool for glycomics. This method has several merits over conventional methods, including a low-cost and simple process without any additional protection/deprotection, purification, and immobilization steps, the use of small amounts of expensive glycan-processing enzymes and nucleotide sugar donors, the synthesis of many glycosidic linkages in a straightforward manner, and the direct application of the synthesized glycans for glycomics.

[Page 5]

As shown in Figure 1, conjugates of glycan and complementary single-stranded oligonucleotide can hybridize to i-motif DNAs that are immobilized on the chip surface under slightly basic conditions. As the pH is lowered, i-motif DNA tends to strongly form a quadruplex structure, resulting in the denaturation of the DNA double helix and thereby enabling the isolation of glycan-oligonucleotide conjugates from the surface. This property would make possible to optimize on-chip enzymatic glycosylation by analyzing isolated glycan-oligonucleotide conjugates using liquid chromatography, resulting in synthesis of structurally defined complex glycans on chip. Therefore, our proposed glycan chip platform could improve the limitation of current platforms by immobilizing structurally simple disaccharide with high synthesis yield and then synthesizing complex glycans on the chip using glycosyltransferases under optimized conditions.

To check the feasibility of on-chip enzymatic glycosylation using the structure switchable DNA-based glycan chip platform, Globo H series (Table 1) were selected as target complex glycans, which are aberrantly overexpressed on human tumor cells and known to be involved in tumor progression.^{36,37}

[Newly added references]

12. Esposito, D., Hurevich, M., Castagner, B., Wang, C. -C & Seeberger, P. H. Automated synthesis of sialylated oligosaccharide. *Beilstein J. Org. Chem.* **8**, 1601–1609 (2012).
13. Fair, R. J., Hahm H. S. & Seeberger, P. H. Combination of automated solid-phase and enzymatic oligosaccharide synthesis provides access to $\alpha(2,3)$ -sialylated glycans. *Chem. Comm.* **51**, 6183–6185 (2015).
14. Song, X. et al. Oxidative release of natural glycans for functional glycomics. *Nat. Methods* **13**, 528–536 (2016).

[Supporting figure for on-going work]

Combination of microfluidic system and DNA-based glycan chip for on-chip biosynthesis of large number of complex glycans. (a) Digital photograph of microfluidic system with two channels. (b) Fabrication of DNA-based glycan chip platform. (c) Introduction of closed microfluidic system. (d) Addition of enzyme (*e.g.*, LgtC) solution containing nucleotide sugar using microfluidic system. (e) Separation of biosynthesized glycan-oligonucleotide conjugate (*e.g.*, Gb3) using microfluidic system. (f) Bio-LC analysis of separated glycan-oligonucleotide conjugates.

[Page 13]

These results indicated that Globo H hexasaccharide-binding proteins are present on MCF-7 breast cancer cells, which can support a previous study that identified the major binding protein for Globo H hexasaccharide on the cancer cell membrane.⁵⁴

In this work, we clearly demonstrated that the structure switchable DNA-based glycan chip platform enables to biosynthesize complex glycans and analyze glycan-involved applications on the chip, simultaneously. For its practical application, it is necessary to prepare large-scale glycan chip with large number of on-chip biosynthesized complex glycans. Thus, we are working on further studies to devise a solution for synthesizing large number of glycans on the chip by combining with microfluidics system containing multichannel. The introduction of microfluidics system can have advantages of providing large number of glycans on the chip by biosynthesizing different glycans for each channel, efficiently reusing glycan-processing enzymes, and minimally using expensive reagents (*e.g.*, nucleotide sugars). We anticipate that the structure switchable DNA-based glycan chip platform combined with microfluidics systems would enable to realize practical applications of glycan chips with on-chip biosynthesized complex glycans.

Other specific points

8. Line 112, reference should be given for the thiol-ene photochemical reaction. The lactose-DNA conjugate should be fully characterized and assigned by NMR, and also by MS.

Author response:

We thank the reviewer for the comment. As following the reviewer's comment, we included the reference for the thiol-ene photochemical reaction in the revised manuscript (page 15) with newly added reference (55). We also characterized the lactose-oligonucleotide conjugates by ^1H NMR and MS analyses. We included these data in the revised Supporting Information (revised Figures S3 & newly added Figure S9).

[Page 15]

Synthesis of glycan-oligonucleotide conjugates. Lactose-oligonucleotide conjugates were synthesized by conjugating allyl lactose with thiol-modified oligonucleotides via photochemical reaction using a photoinitiator.⁵⁵ Single-stranded oligonucleotide pretreated with 1,4-dithiothreitol (100 nmol, 1 equiv.), 1-O-allyl-D-lactose (100 nmol, 1 equiv.), and photoinitiator (10 nmol, DMPA) were mixed in 1 mL of deionized water.

[Newly added reference]

55. Lowe, A. B. Thiol-ene "click" reactions and recent applications in polymer and materials synthesis: a first update. *Poly, Chem.* 5, 4820-4870 (2014).

[Revised Figure S3]

Figure S3. ^1H NMR spectrum of lactose-oligonucleotide conjugates. ^1H NMR spectrum of lactose-oligonucleotide conjugates. ^1H NMR (500 MHz, D_2O) δ 8.26-7.34 (m, oligonucleotide-H), 6.19-6.06 (m, oligonucleotide-H), 6.03-5.95 (m, 2H), 5.41-5.38 (dd, $J = 17.2, 1.3$ Hz, 1H), 5.31-5.29 (d, $J = 10.7$ Hz, 1H), 4.55 (d, $J = 8$ Hz, 1H), 4.46 (d, $J = 7.8$ Hz, 1H), 4.43-4.34 (m, 1H), 4.26-4.22 (m, 1H), 4.01-3.98 (dd, $J = 12.3, 2$ Hz, 1H), 3.94 (d, $J = 3.3$ Hz, 1H), 3.83-3.58 (m, 8H), 3.57-3.53 (t, $J = 8.9$ Hz, 1H), 3.36-3.33 (t, $J = 8.6$ Hz, 1H), 3.23-3.19 (q, oligonucleotide-H, $J = 7.3$ Hz), 1.30-1.27 (t, oligonucleotide-H, $J = 7.4$ Hz) ppm.

[Newly added Figure S9]

Figure S9. MALDI-TOF MS analysis of lactose-oligonucleotide conjugates. MALDI-TOF MS: calculated for [lactose-oligonucleotide conjugate] 7289.47, found 7290.94 $[\text{M}+\text{H}]^+$ and 7311.51 $[\text{M}+\text{Na}]^+$.

9. Line 122, the chemistry used for covalent immobilization of the i-motif DNAs should be described here.

Author response:

We thank the reviewer for the comment. As following the reviewer's comment, we included the chemistry for covalent immobilization of the i-motif DNA linker in the revised manuscript (page 7) with newly added reference (42).

[Page 7]

First, 70 μ M i-motif DNAs dissolved in phosphate-buffered saline (PBS) solution (pH 4.5) were robotically spotted onto a gold chip and incubated overnight in a humidified chamber. **i-motif DNAs were attached onto a gold surface via gold-thiol interactions.**⁴² After immobilization of i-motif DNAs, the chip was blocked with poly(ethylene glycol) methyl ether thiol in PBS (pH 4.5). The lactose-oligonucleotide conjugates were incubated onto an i-motif DNA-immobilized surface that was pretreated with PBS buffer (pH 9.0).

[Newly added reference]

42. Herne, T. M. and Tarloy, M. J. Characterization of DNA probes immobilization on gold surfaces. *J. Am. Chem. Soc.* **119**, 8916-8920 (1997).

10. Lines 227-233, the GM1 related work doesn't fit under the heading.

Author response:

We thank the reviewer for the good comment. As following the reviewer's comment, we included the GM1-related work in the newly added heading (page 12).

[Page 12]

Interaction analysis of cholera toxin B subunit with on-chip biosynthesized GM1-related complex glycans. To further validate the feasibility of the **structure switchable** DNA-based glycan chip for on-chip complex glycan biosynthesis, GM1 pentasaccharide and its related glycans were also synthesized on the lactose disaccharide-immobilized surface using glycosyltransferases under their optimized conditions (Table S2 & Figures **S18-S20**). Unlike GM3 trisaccharide and GM2 tetrasaccharide, GM1 pentasaccharide was biosynthesized at low efficiency. This result was due to the low catalytic activity of the CgtB enzyme (17 U/L), which was consistent with a previous study.⁵⁰ **We analyzed the interaction of the cholera toxin B subunit and GM1 pentasaccharide-related complex glycans biosynthesized on the chip surface (Figure S21). This result showed that the intrinsic selectivity of the cholera toxin B subunit to the on-chip biosynthesized glycans was consistent with previous studies.**^{5,19}

11. Line 291, reaction scale should be described.

Author response:

We thank the reviewer for the comment. As following the reviewer's comment, we included our reaction scale for synthesis of glycan-oligonucleotide conjugates in the revised manuscript (page 16).

[page 16]

Synthesis of glycan-oligonucleotide conjugates. Lactose-oligonucleotide conjugates were synthesized by photochemical reaction using a photoinitiator. Single-stranded oligonucleotide pretreated with 1,4-dithiothreitol (**100 nmol**, 1 equiv.), 1-O-allyl-D-lactose (**100 nmol**, 1 equiv.), and photoinitiator (**10 nmol**, DMPA) were mixed in 1 mL of deionized water. The mixture was stirred under UV light (365 nm) using

a UVP Blak-Ray® XX-15L UV bench lamp (15 W; Thermo Fisher Scientific) for 2 h.

12. Line 310, more details (references) should be given on how the glass slides were coated with gold.

Author response:

We thank the reviewer for the comment. As following the reviewer's comment, we included detailed method for preparation of gold-coated glass slides in the revised manuscript (page 17) with newly added references (56, 57).

[Page 17]

Fabrication of the DNA-based glycan chip platform. Glass slides (76 × 26 × 1 mm; Marienfeld GmbH & Co. KG, Lauda-Königshofen, Germany) were coated with ~100 nm thick gold and an ~10 nm titanium adhesive layer as described previously.^{56,57} Glass slides were cleaned by ultrasonication in trichloroethylene, acetone, isopropyl alcohol, and pure water, and dried by centrifugation. Titanium and gold films were formed on glass slide at a deposition of 1 Å/s and a chamber pressure of 3×10^{-6} mbar by E-beam evaporator (KVE-4000; Korea Vacuum Tech, Gimpo, Korea). Prepared slide substrates were placed on the sample holder disc and a titanium adhesion layer of 10 nm thickness was firstly deposited, followed by a gold layer of thickness 100 nm. The film thickness was monitored using quartz crystal microbalance (QCM; INFICON, Bad Ragaz, Switzerland). After the completion of deposition, gold-coated glass slides were rinsed with acetone and methanol.

[Newly added references]

56. Barton, S. J. et al. Improved performance of near infrared excitation Raman spectroscopy using reflective thin-film gold on glass substrates for cytology samples. *Anal. Methods*. **11**, 6023-6032 (2019).
57. Todeschini, M. et al. Influence of Ti and Cr adhesion layers on ultrathin au films. *ACS Appl. Mater. Interfaces*. **9**, 37374-37385 (2017).

13. Line 324, the volume of 1 nmol lactose-oligonucleotide conjugate should be given for the glycan immobilization step.

Author response:

We thank the reviewer for the comment. As following the reviewer's comment, we included the volume of the solution containing 1 nmol lactose-oligonucleotide conjugate in the revised manuscript (page 18).

[Page 18]

Glycan immobilization was conducted by reacting the i-motif DNA-immobilized chip with a 40 μ L hybridization solution (100 mM sodium phosphate and 1 M NaCl; pH 9.0) containing 1 nmol lactose-oligonucleotide conjugates for 3 h.

14. Line 353, the volume of enzyme solution should be given.

Author response:

We thank the reviewer for the comment. For on-chip biosynthesis of Globo H hexasaccharide, SSEA-4 hexasaccharide, Gb5 pentasaccharide, Gb4 tetrasaccharide, and Gb3 trisaccharide, the volumes of all enzyme solutions were 25 μ L. As following the reviewer's comment, we included the volume of enzyme solution for fabrication of DNA-based glycan chip consisting of Globo H series in the revised manuscript (page 19).

[Page 19]

Commercially available glycosyltransferases were used for on-chip glycosylation of complex glycans.⁵⁷ The lactose-immobilized surface was divided into five different blocks using GeneFrame®. For synthesis of Gb3 trisaccharide, a 25 µL solution of LgtC (4 mU), UDP-Gal (10 mM), Tris-HCl (pH 7, 100 mM), and MgCl₂ (10 mM) was dropped into all five blocks, and the slide was incubated at 37 °C for 48 h in a humidified chamber. For synthesis of Gb4 tetrasaccharide from Gb3 trisaccharide, a 25 µL solution of LgtD (4 mU), UDP-GalNAc (10 mM), Tris-HCl (100 mM; pH 7.0), and MgCl₂ (10 mM) was dropped into four blocks of Gb3 trisaccharide-synthesized five blocks, and the slide was incubated at 37 °C for 48 h in a humidity chamber. For galactosylation (synthesis of Gb5 pentasaccharide) of Gb4 tetrasaccharide, a 25 µL solution of LgtD (4 mU), UDP-Gal (10 mM), Tris-HCl (100 mM; pH 7.0), and MgCl₂ (10 mM) was dropped into three blocks of Gb4 tetrasaccharide-synthesized four blocks, and the slide was incubated at 37 °C for 60 h in a humidified chamber. For fucosylation (synthesis of Globo H hexasaccharide) of Gb5 pentasaccharide, a 25 µL solution of α1,2-FucT (4 mU), GDP-Fucose (10 mM), Tris-HCl (100 mM; pH 7.0), and MgCl₂ (10 mM) were dropped into one block of Gb5 pentasaccharide-synthesized three blocks and the slide was incubated at 37 °C for 48 h in a humidified chamber. For the synthesis of SSEA-4 hexasaccharide, a 25 µL solution of α2,3-SialT (4 mU), CMP-Neu5Ac (10 mM), Tris-HCl (100 mM; pH 7.5), and MgCl₂ (20 mM) was dropped into one block of Gb5 pentasaccharide-synthesized three blocks, and the slide was incubated at 37 °C for 48 h in a humidified chamber. After each reaction, the slides were washed once with washing buffer I and twice with washing buffer II. The slides were then dried by centrifugation at 213 × g for 3 min.

15. Table 1, Symbolic structures should be drawn from non-reducing end to reducing end (the same direction of the text sequence).

Author response:

We thank the reviewer for the good comment. As following the reviewer's comment, we corrected the symbolic structures in the revised Table 1.

[Revised Table 1]**Table 1.** Glycans used in this work and their sequences.

Glycan	Saccharide	Sequence	Symbol ^a
Globo H	hexasaccharide (globohexaose)	Fucα1-2Galβ1-3GalNAcβ1-3Galα1-4Galβ1-4Glc	
SSEA 4	hexasaccharide (globohexaose)	Neu5Acα2-3Galβ1-3GalNAcβ1-3Galα1-4Galβ1-4Glc	
Gb5	pentasaccharide (globopentaose)	Galβ1-3GalNAcβ1-3Galα1-4Galβ1-4Glc	
Gb4	tetrasaccharide (globotetraose)	GalNAcβ1-3Galα1-4Galβ1-4Glc	
Gb3	trisaccharide (globotriaose)	Galα1-4Galβ1-4Glc	
Lactose	disaccharide	Galβ1-4Glc	

^a ●, glucose (Glc); ●, galactose (Gal); ■, N-acetylgalactosamine (GalNAc); ◆, N-acetylneuraminic acid (Neu5Ac); ▲, fucose (Fuc)

16. Figures S5-S8, conditions for HPLC analyses should be given in the footnote. How were the products characterised? Any MS data available?

Author response:

We thank the reviewer for the comment. We included analysis condition for Bio-LC in the

revised Figures S12-S16. The products were characterized by using ^1H NMR. We included the data and related information in the revised manuscript (pages 6, 14, 16, and 18) and Supporting Information (Figures S3-S8) with newly added reference (59). In addition, MALDI-TOF MS analyses of the products identified by ^1H NMR analyses were performed using commonly used 3-HPA matrix. However, as shown in the response only figure 1-6, the corresponding MS peaks could not be identified in other conjugates except for the lactose-oligonucleotide conjugate. Accordingly, we performed matrix optimization for MS analysis using various matrices (2,5-DHB, super DHB, THAP, α -CHCA, 6-aza-2-thiothymine, sinapinic acid, 3,4-diaminobenzophenone, 9-nitroanthracene, fucose, 5-chloro-2-benzothiazolethiol, 2-amino-4-methyl-5-nitropyridine, and 2-amino-4-methyl-3-nitropyridine). However, although many our efforts, we could not identify MS peaks corresponding to the glycan-oligonucleotide conjugates. These seem that the glycan-oligonucleotide conjugates are not ionized or fragmentation occurs in the used matrices. We only included MS data of lactose-oligonucleotide conjugates in the revised Supporting Information (newly added Figure S9).

Figure S15. HPLC analyses of on-surface biosynthesized SSEA-4 hexa saccharide from Gb5 pentasaccharide by α -2,3-SialT glycosyltransferase for (a) 12 h, (b) 24 h, and (c) 48 h. Symbols: blue circle, Glc; yellow circle, Gal; yellow square, GalNAc; purple square, Neu5Ac.

The products were analyzed by liquid chromatography (LC-5000, Thermo Fisher Scientific) using a Carbo-Pac PA100 column (4 mm \times 250 mm, Dionex, Sunnyvale, CA, USA), isocratic elution mode with 100 mM sodium hydroxide, a flow rate of 0.25 ml/min, and an Ag/AgCl reference electrode for electrochemical detection.

Figure S16. HPLC analyses of on-surface biosynthesized Globo H hexa saccharide from Gb5 pentasaccharide by α -1,2-FucT glycosyltransferase for (a) 24 h, and (b) 48 h. Symbols: blue circle, Glc; yellow circle, Gal; yellow square, GalNAc; red triangle, Fuc.

The products were analyzed by liquid chromatography (LC-5000, Thermo Fisher Scientific) using a Carbo-Pac PA100 column (4 mm \times 250 mm, Dionex, Sunnyvale, CA, USA), isocratic elution mode with 100 mM sodium hydroxide, a flow rate of 0.25 ml/min, and an Ag/AgCl reference electrode for electrochemical detection.

[Page 6]

The synthesized lactose-oligonucleotide conjugates were purified using high-pressure liquid chromatography (Figure S2) and confirmed by ^1H nuclear magnetic resonance (NMR) and mass spectrometry (MS) analyses (Figures S3-S9).

[Page 14]

Materials. Poly(ethylene glycol) methyl ether thiol (average M_n 800), 1,4-dithiothreitol, 4-dimethylaminopyridine, 2,2-dimethoxy-2-phenylacetophenone (DMPA), uridine 5'-diphospho-N-acetylgalactosamine (UDP-GalNAc) disodium salt, uridine 5'-diphosphogalactose (UDP-Gal) disodium salt, biotinylated Griffonia simplicifolia isolectin B4 (GS-IB₄), and Envi-Carb SPE column were purchased from Sigma-Aldrich (St. Louis, MO, USA). Cytidine-5'-monophospho-N-acetylneuraminic acid (CMP-Neu5Ac) sodium salt and Pasteurella multocida α -2,3-sialyltransferase (α 2,3-SialT) were obtained from GeneChem Inc.

[Page 16]

Nuclear magnetic resonance (NMR) analysis of glycan-oligonucleotide conjugates: ^1H NMR spectrum (solvent D_2O) of glycan-oligonucleotide conjugate was acquired using an NMR spectrometer (500 MHz; Bruker, Karlsruhe, Germany). Data are reported as follows: chemical shifts (δ ppm), multiplicity (s = singlet, d = doublet, q = quartet, m = multiplet), and coupling constants (Hz).

Matrix assisted laser desorption/ionization time-of-flight (MALDI-TOF) mass spectrometry (MS) analyses of glycan-oligonucleotide conjugates: MALDI-TOF spectrum was measured on AXIMA LNR MALDI-TOF MS (Shimadzu, Kyoto, Japan) using 3-hydroxypyridine-2-carboxylic acid (3-HPA) as a matrix (50 mg/mL in deionized water). Dried sample was mixed with 10 μL of matrix solution directly on the MALDI target, followed by vacuum drying.

[Page 18]

The eluted sample was purified by solid-phase extraction chromatography additionally.⁵⁹ Envi-Carb SPE column was equilibrated in a 15 mL conical tube using 80% (v/v) acetonitrile in 0.1% (v/v) trifluoroacetic acid and ultrapure water and then, spun at $60 \times g$ for 50 s. 1 mL sample was added to the Envi-Carb column. The column was washed with 2 mL of ultrapure water, 2 mL of 25% (v/v) acetonitrile, 1 mL of ultrapure water, and 2 mL of 10 mM triethylammonium acetate (pH 7.0) sequentially. The final product was eluted with 2 mL of 25% (v/v) acetonitrile in 50 mM triethylammonium acetate (pH 7.0) and dried to remove the solvent. In addition, the slides were incubated with the complexes of biotinylated GS-IB₄, RCA120, SBA, LTL, and MAL II labeled by streptavidin-Alexa Fluor® 647 to check the products of enzymatic reactions.

[Newly added reference]

59. Barnes, J. et al. Isolation and analysis of sugar nucleotides using solid phase extraction and fluorophore assisted carbohydrate electrophoresis. *MethodsX*. **3**, 251-260 (2016).

[Revised Figure S3]

Figure S3. ^1H NMR spectrum of lactose-oligonucleotide conjugates. ^1H NMR (500 MHz, D_2O) δ 8.26-7.34 (m, oligonucleotide-H), 6.19-6.06 (m, oligonucleotide-H), 6.03-5.95 (m, 2H), 5.41-5.38 (dd, $J = 17.2, 1.3$ Hz, 1H), 5.31-5.29 (d, $J = 10.7$ Hz, 1H), 4.55 (d, $J = 8$ Hz, 1H), 4.46 (d, $J = 7.8$ Hz, 1H), 4.43-4.34 (m, 1H), 4.26-4.22 (m, 1H), 4.01-3.98 (dd, $J = 12.3, 2$ Hz, 1H), 3.94 (d, $J = 3.3$ Hz, 1H), 3.83-3.58 (m, 8H), 3.57-3.53 (t, $J = 8.9$ Hz, 1H), 3.36-3.33 (t, $J = 8.6$ Hz, 1H), 3.23-3.19 (q, oligonucleotide-H, $J = 7.3$ Hz), 1.30-1.27 (t, oligonucleotide-H, $J = 7.4$ Hz) ppm.

[Newly added Figure S4]

Figure S4. ^1H NMR spectrum of Gb3-oligonucleotide conjugates. ^1H NMR (500 MHz, D_2O) δ 6.01 (m, 1H), 6.00-5.99 (dd, $J = 20.4, 8.5$ Hz, 3H), 5.67-5.65 (m, 2H), 4.40-4.39 (d, $J = 2.9$ Hz, 2H), 4.36-4.35 (d, $J = 2.5$ Hz, 1H), 4.31-4.18 (m, 6H), 4.05 (s, 1H), 3.95-3.92 (dd, $J = 10.1, 2.5$ Hz, 2H), 3.83-3.74 (m, 5 H), 3.68-3.67 (d, $J = 6.5$ Hz, 2H), 3.37-3.36 (m, 1H), 3.25-3.20 (q, oligonucleotide-H, $J = 7.2$ Hz), 1.32-1.29 (t, oligonucleotide-H, $J = 7.2$ Hz) ppm.

[Newly added Figure S5]

Figure S5. ^1H NMR spectrum of Gb4-oligonucleotide conjugates. ^1H NMR (500 MHz, D_2O) δ 6.01 (m, 1H), 6.00-5.98 (d, $J = 8.6$ Hz, 3H), 5.58-5.56 (m, 2H), 4.48-4.47 (d, $J = 6.5$ Hz, 1H), 4.40-4.37 (m, 4H),

4.31-4.26 (m, 5H), 4.22-4.20 (m, 3H), 4.06-4.05 (d, J = 2.5 Hz, 2H), 4.00-3.95 (m, 3H), 3.88-3.72 (m, 5H), 3.69-3.65 (m, 3H), 3.59-3.55 (m, 1H), 3.29-3.28 (m, 1H), 2.10 (s, 3H), 3.25-3.20(q, oligonucleotide-H, J = 7.2 Hz), 1.32-1.29 (t, oligonucleotide-H, J = 7.0 Hz) ppm.

[Newly added Figure S6]

Figure S6. ^1H NMR spectrum of Gb5-oligonucleotide conjugates. ^1H NMR (500 MHz, D_2O) δ 6.00 (m, 5H), 5.67-5.64 (dd, J = 12.6, 3.8 Hz, 2H), 5.57-5.55 (dd, J = 7.1, 3.4 Hz, 2H), 4.40-4.38 (m, 6H), 4.36-4.34 (m, 2H), 4.30-4.29 (m, 5H), 4.26-4.25 (m, 2H), 4.23-4.18 (m, 4H), 4.15 (m, 1H), 4.11-4.10 (m, 1H), 4.06-4.04 (m, 2H), 4.00-3.97 (dd, J = 10.7, 3.5 Hz, 1H), 3.95-3.92 (dd, J = 10.2, 3.4 Hz, 1H), 3.83-3.74 (m, 6H), 3.45-3.43 (d, J = 7.9 Hz, 1H), 2.10 (s, 3H), 3.24-3.20(q, oligonucleotide-H, J = 7.4 Hz), 1.31-1.28 (t, oligonucleotide-H, J = 7.3 Hz) ppm.

[Newly added Figure S7]

Figure S7. ^1H NMR spectrum of SSEA-4-oligonucleotide conjugates. ^1H NMR (500 MHz, D_2O) δ 6.00 (s, 1H), 5.98 (s, 1H), 5.96-5.94 (d, J = 8 Hz, 2H), 5.81-5.80 (d, J = 7.7 Hz, 2H), 4.33-4.32 (m, 10H), 4.29-4.28 (m, 4H), 4.22-4.21 (m, 4H), 4.20 (m, 2H), 4.15-4.01 (m, 4H), 3.95-3.94 (d, J = 2.7 Hz, 2H), 3.93-3.92 (d, J = 2.8 Hz, 2H), 3.83-3.79 (m, 7H), 3.75-3.74 (m, 5H), 2.74-2.71 (dd, J = 14.6, 4.7 Hz, 1H), 2.23 (s, 3H), 2.04 (s, 3H), 1.78-1.74 (t, J = 12 Hz, 1H), 3.23-3.19(q, oligonucleotide-H, J = 7.3 Hz), 1.30-1.27 (t, oligonucleotide-H, J = 7.3 Hz) ppm.

[Newly added Figure S8]

Figure S8. ^1H NMR spectrum of Globo H-oligonucleotide conjugates. ^1H NMR (500 MHz, D_2O) δ 6.01 (m, 1H), 5.95-5.94 (d, $J = 2.15$ Hz, 3H), 5.94-5.93 (d, $J = 2.4$ Hz, 3H), 5.55-5.54 (m, 1H), 4.56-4.54 (m, 3H), 4.49-4.47 (m, 3H), 4.38-4.37 (m, 2H), 4.36-4.34 (m, 6H), 4.30-4.29 (m, 1H), 4.27-4.24 (m, 2H), 4.21-4.20 (m, 7H), 4.10-4.09 (m, 7H), 4.04 (d, $J = 2.9$ Hz, 1H), 3.98-3.96 (m, 1H), 3.78-3.76 (d, $J = 7.4$ Hz, 2H), 3.67-3.63 (m, 1H), 3.58-3.56 (m, 1H), 3.36-3.33 (m, 1H), 2.09 (s, 3H), 1.20-1.19 (d, $J = 7.1$ Hz, 3H), 3.23-3.19 (q, oligonucleotide-H, $J = 7.3$ Hz), 1.30-1.27 (t, oligonucleotide-H, $J = 7.3$ Hz) ppm.

[Response only figure 1: MALDI-TOF MS analysis of lactose-oligonucleotide conjugate]

[Response only figure 2: MALDI-TOF MS analysis of Gb3-oligonucleotide conjugate]

[Response only figure 3: MALDI-TOF MS analysis of Gb4-oligonucleotide conjugate]

[M+Na]⁺

17. In the Material and Methods sections, there is a heading Nuclear magnetic resonance (NMR) analysis of glycan-oligonucleotide conjugates. However only the lactose conjugate is investigated by NMR.

Author response:

We thank the reviewer for the comment. As following the reviewer's comment, we performed ¹H NMR analyses for all glycan-oligonucleotide conjugates (Lactose-, Gb3-, Gb4-, Gb5-, SSEA-4-, and Globo H-oligonucleotide conjugates). We included the data and related information in the revised manuscript (pages 6, 14, 16, and 18) and Supporting Information (Newly added Figures S3-S8) with newly added reference (59).

[Page 6]

The synthesized lactose-oligonucleotide conjugates were purified using high-pressure liquid chromatography (Figure S2) and confirmed by ¹H nuclear magnetic resonance (NMR) and mass spectrometry (MS) analyses (Figures S3-S9).

[Page 14]

Materials. Poly(ethylene glycol) methyl ether thiol (average Mn 800), 1,4-dithiothreitol, 4-dimethylaminopyridine, 2,2-dimethoxy-2-phenylacetophenone (DMPA), uridine 5'-diphospho-N-acetylgalactosamine (UDP-GalNAc) disodium salt, uridine 5'-diphosphogalactose (UDP-Gal) disodium salt, biotinylated Griffonia simplicifolia isolectin B4 (GS-IB₄), and Envi-Carb SPE column were purchased from Sigma-Aldrich (St. Louis, MO, USA). Cytidine-5'-monophospho-N-acetylneuraminic acid (CMP-Neu5Ac) sodium salt and Pasteurella multocida α-2,3-sialyltransferase (α2,3-SialT) were obtained from GeneChem Inc.

[Page 16]

Nuclear magnetic resonance (NMR) analysis of glycan-oligonucleotide conjugates: ¹H NMR spectrum (solvent D₂O) of glycan-oligonucleotide conjugate was acquired using an NMR spectrometer (500 MHz; Bruker, Karlsruhe, Germany). Data are reported as follows: chemical shifts (δ ppm), multiplicity (s = singlet, d = doublet, q = quartet, m = multiplet), and coupling constants (Hz).

[Page 18]

The eluted sample was purified by solid-phase extraction chromatography additionally.⁵⁹ Envi-Carb SPE column was equilibrated in a 15 mL conical tube using 80% (v/v) acetonitrile in 0.1% (v/v) trifluoroacetic acid and ultrapure water and then, spun at 60 × g for 50 s. 1 mL sample was added to the Envi-Carb column. The column was washed with 2 mL of ultrapure water, 2 mL of 25% (v/v) acetonitrile, 1 mL of ultrapure water, and 2 mL of 10 mM triethylammonium acetate (pH 7.0) sequentially. The final product was eluted with 2 mL of 25% (v/v) acetonitrile in 50 mM triethylammonium acetate (pH 7.0) and dried to remove the solvent. In addition, the slides were incubated with the complexes of biotinylated GS-IB₄, RCA120, SBA, LTL, and MAL II labeled by streptavidin-Alexa Fluor® 647 to check the products of enzymatic reactions.

[Newly added reference]

59. Barnes, J. et al. Isolation and analysis of sugar nucleotides using solid phase extraction and fluorophore assisted carbohydrate electrophoresis. *MethodsX*. **3**, 251-260 (2016).

[Revised Figure S3]

Figure S3. ^1H NMR spectrum of lactose-oligonucleotide conjugates. ^1H NMR (500 MHz, D_2O) δ 8.26-7.34 (m, oligonucleotide-H), 6.19-6.06 (m, oligonucleotide-H), 6.03-5.95 (m, 2H), 5.41-5.38 (dd, $J = 17.2, 1.3$ Hz, 1H), 5.31-5.29 (d, $J = 10.7$ Hz, 1H), 4.55 (d, $J = 8$ Hz, 1H), 4.46 (d, $J = 7.8$ Hz, 1H), 4.43-4.34 (m, 1H), 4.26-4.22 (m, 1H), 4.01-3.98 (dd, $J = 12.3, 2$ Hz, 1H), 3.94 (d, $J = 3.3$ Hz, 1H), 3.83-3.58 (m, 8H), 3.57-3.53 (t, $J = 8.9$ Hz, 1H), 3.36-3.33 (t, $J = 8.6$ Hz, 1H), 3.23-3.19 (q, oligonucleotide-H, $J = 7.3$ Hz), 1.30-1.27 (t, oligonucleotide-H, $J = 7.4$ Hz) ppm.

[Newly added Figure S4]

Figure S4. ^1H NMR spectrum of Gb3-oligonucleotide conjugates. ^1H NMR (500 MHz, D_2O) δ 6.01 (m, 1H), 6.00-5.99 (dd, $J = 20.4, 8.5$ Hz, 3H), 5.67-5.65 (m, 2H), 4.40-4.39 (d, $J = 2.9$ Hz, 2H), 4.36-4.35 (d, $J = 2.5$ Hz, 1H), 4.31-4.18 (m, 6H), 4.05 (s, 1H), 3.95-3.92 (dd, $J = 10.1, 2.5$ Hz, 2H), 3.83-3.74 (m, 5H), 3.68-3.67 (d, $J = 6.5$ Hz, 2H), 3.37-3.36 (m, 1H), 3.25-3.20 (q, oligonucleotide-H, $J = 7.2$ Hz), 1.32-1.29 (t, oligonucleotide-H, $J = 7.2$ Hz) ppm.

[Newly added Figure S5]

Figure S5. ^1H NMR spectrum of Gb4-oligonucleotide conjugates. ^1H NMR (500 MHz, D_2O) δ 6.01 (m, 1H), 6.00-5.98 (d, $J = 8.6$ Hz, 3H), 5.58-5.56 (m, 2H), 4.48-4.47 (d, $J = 6.5$ Hz, 1H), 4.40-4.37 (m, 4H), 4.31-4.26 (m, 5H), 4.22-4.20 (m, 3H), 4.06-4.05 (d, $J = 2.5$ Hz, 2H), 4.00-3.95 (m, 3H), 3.88-3.72 (m, 5H), 3.69-3.65 (m, 3H), 3.59-3.55 (m, 1H), 3.29-3.28 (m, 1H), 2.10 (s, 3H), 3.25-3.20 (q, oligonucleotide-H, $J = 7.2$ Hz), 1.32-1.29 (t, oligonucleotide-H, $J = 7.0$ Hz) ppm.

[Newly added Figure S6]

Figure S6. ^1H NMR spectrum of Gb5-oligonucleotide conjugates. ^1H NMR (500 MHz, D_2O) δ 6.00 (m, 5H), 5.67-5.64 (dd, $J = 12.6, 3.8$ Hz, 2H), 5.57-5.55 (dd, $J = 7.1, 3.4$ Hz, 2H), 4.40-4.38 (m, 6H), 4.36-4.34 (m, 2H), 4.30-4.29 (m, 5H), 4.26-4.25 (m, 2H), 4.23-4.18 (m, 4H), 4.15 (m, 1H), 4.11-4.10 (m, 1H), 4.06-4.04 (m, 2H), 4.00-3.97 (dd, $J = 10.7, 3.5$ Hz, 1H), 3.95-3.92 (dd, $J = 10.2, 3.4$ Hz, 1H), 3.83-3.74 (m, 6H), 3.45-3.43 (d, $J = 7.9$ Hz, 1H), 2.10 (s, 3H), 3.24-3.20(q, oligonucleotide-H, $J = 7.4$ Hz), 1.31-1.28 (t, oligonucleotide-H, $J = 7.3$ Hz) ppm.

[Newly added Figure S7]

Figure S7. ^1H NMR spectrum of SSEA-4-oligonucleotide conjugates. ^1H NMR (500 MHz, D_2O) δ 6.00 (s, 1H), 5.98 (s, 1H), 5.96-5.94 (d, $J = 8$ Hz, 2H), 5.81-5.80 (d, $J = 7.7$ Hz, 2H), 4.33-4.32 (m, 10H), 4.29-4.28 (m, 4H), 4.22-4.21 (m, 4H), 4.20 (m, 2H), 4.15-4.01 (m, 4H), 3.95-3.94 (d, $J = 2.7$ Hz, 2H), 3.93-3.92 (d, $J = 2.8$ Hz, 2H), 3.83-3.79 (m, 7H), 3.75-3.74 (m, 5H), 2.74-2.71 (dd, $J = 14.6, 4.7$ Hz, 1H), 2.23 (s, 3H), 2.04 (s, 3H), 1.78-1.74 (t, $J = 12$ Hz, 1H), 3.23-3.19(q, oligonucleotide-H, $J = 7.3$ Hz), 1.30-1.27 (t, oligonucleotide-H, $J = 7.3$ Hz) ppm.

[Newly added Figure S8]

Figure S8. ^1H NMR spectrum of Globo H-oligonucleotide conjugates. ^1H NMR (500 MHz, D_2O) δ 6.01 (m, 1H), 5.95-5.94 (d, $J = 2.15$ Hz, 3H), 5.94-5.93 (d, $J = 2.4$ Hz, 3H), 5.55-5.54 (m, 1H), 4.56-4.54 (m, 3H), 4.49-4.47 (m, 3H), 4.38-4.37 (m, 2H), 4.36-4.34 (m, 6H), 4.30-4.29 (m, 1H), 4.27-4.24 (m, 2H), 4.21-4.20 (m, 7H), 4.10-4.09 (m, 7H), 4.04 (d, $J = 2.9$ Hz, 1H), 3.98-3.96 (m, 1H), 3.78-3.76 (d, $J = 7.4$ Hz, 7.4 Hz)

Hz, 2H), 3.67-3.63 (m, 1H), 3.58-3.56 (m, 1H), 3.36-3.33 (m, 1H), 2.09 (s, 3H), 1.20-1.19 (d, J = 7.1 Hz, 3H), 3.23-3.19 (q, oligonucleotide-H, J = 7.3 Hz), 1.30-1.27 (t, oligonucleotide-H, J = 7.3 Hz) ppm.

18. *The previous study cited by the authors identified a major binding protein for Globo H hexasaccharide on the cancer cell membrane; it is mentioned but not followed up.*

Author response:

We thank the reviewer for the good comment. We additionally performed flow cytometry analyses of MCF-7 breast cancer and MCF-10A breast normal cells for Globo H hexasaccharide binding to support the interaction of cancer cells with the glycan chip. For flow cytometry analyses, Globo H hexasaccharide-Alexa Fluor® 488 conjugates were prepared as previously described method (newly added reference 59). The conjugates were analyzed by high-resolution LC-MS (newly added Figure S22). Consistent with the results of the glycan chip (revised Figures 4c & 4d), flow cytometry analyses clearly demonstrated that MCF-7 cancer cells strongly bound to Globo H hexasaccharide, whereas MCF-10A normal cells did not (newly added Figures 4e-g). With the glycan chip data, this result could support a previous study about identification of a major binding protein for Globo H hexasaccharide on the cancer cell membrane (reference 54). We revised the original sentence “which can be supported by a previous study that identified the major binding protein for Globo H hexasaccharide on the cancer cell membrane” to “which could support a previous study that identified the major binding protein for Globo H hexasaccharide on the cancer cell membrane” (pages 13). We also included the related sentences in the revised manuscript (pages 13 & 19-20) with newly added reference (60).

[Page 13]

MCF-7 breast cancer cells strongly recognized Globo H hexasaccharide on the chip, while MCF-10A breast normal cells did not (Figures 4c & 4d). This could be also supported by flow cytometry analyses of MCF-7 cancer and MCF-10A normal cells for Globo H hexasaccharide binding (Figures 4e-g). Except for that of Globo H hexasaccharide, the binding affinities of both cells for other biosynthesized complex glycans were similar. These results indicated that Globo H hexasaccharide-binding proteins can be present on MCF-7 breast cancer cells, which could support a previous study that identified the major binding protein for Globo H hexasaccharide on the cancer cell membrane.⁵⁴

[Pages 19-20]

Fluorescence-activated cell sorting (FACS) analysis. MCF-7 breast cancer cells were cultured in DMEM (high glucose) supplemented with 10% (v/v) heat-inactivated FBS, 100 U/mL penicillin, and 100 µg/mL streptomycin. MCF-10A breast normal cells were cultured with Mammary Epithelial Basal Medium, which contains bovine pituitary extract, hydrocortisone, human epidermal growth factor, insulin, gentamicin, and amphotericin-B. Both cells were incubated at 37 °C in a humidified atmosphere of 5% CO₂ and 95% air, and they were subcultured every 3 days. After incubation, the cells were detached and centrifuged.

To prepare dye-conjugated Globo H hexasaccharide, Alexa Fluor® 488 hydrazide (1.75 µmol, 1 equiv.) and Globo H hexasaccharide (0.88 µmol, 0.5 equiv.) were mixed in 1 mL of 100 mM PBS buffer (pH 7.0).⁶⁰ The mixture was incubated at 37 °C for 6 h in a humidity chamber. After the reaction, the mixture was analyzed by liquid chromatograph-mass spectrometry (LC-MS; Waters, Milford, MA, USA) (Figure S22).

To analyze cell-Globo H hexasaccharide interactions on cell surface, both cells were treated with Alexa Fluor® 488 conjugated-Globo H hexasaccharide in culture medium at 37 °C for 1 h in a humidified atmosphere of 5% CO₂ and 95% air, respectively. Each solution was centrifuged to remove remaining dye-conjugated Globo H hexasaccharide. After washing with culture medium and DPBS, glycan-treated and non-treated cells resuspended in DPBS were placed into the wells of non-coated 96-

well plate. These cells were sorted out by FACS (Beckman Coulter, Brea, CA, USA). Data were acquired and analyzed by using CytExpert software (Beckman Coulter).

[Newly added reference]

60. Lee, B.-S. et al. Biotinylation of peptides/proteins using biocytin hydrazide. *J. Chin. Chem. Soc.* **54**, 541-548 (2007).

[Newly added Figures 4e-g]

Figure 4. (e) Schematic presentation for analyzing the binding of MCF-7 breast cancer and MCF-10A breast normal cells to Globo H hexasaccharide using flow cytometry. FACS analyses for the binding (f) MCF-7 breast cancer and (g) MCF-10A breast normal cells with Globo H hexasaccharide.

[Newly added Figure S22]

Figure S22. High-resolution LC-MS analysis of Globo H hexasaccharide-Alexa Fluor[®] 488 conjugate. LC-MS: calculated for [Globo H hexasaccharide-Alexa Fluor[®] 488 conjugate] 1545.38, found 1643.0857 [M+HSO₄]⁻.

19. L90, L95, L241; it is stated (L90) that glycans of Globo H series are overexpressed on human tumour cells, therefore, on the face of it, it is not immediately apparent why there should be

interactions between breast cells (expressing Globo H glycans) and on-chip-biosynthesized Globo H glycans (L95). Some reconciliation by the authors is required at this stage and before L241 where the hypothesis that there are present on breast cancer cells specific binding proteins for Globo-series glycans. The issue of self-neutralization between glycans and glycan-binding proteins expressed on the same cells should be considered.

Author response:

We thank the reviewer for good comment. As following the reviewer's comment, we reconciled two sentences in the revised manuscript to make readers to understand more clearly. We moved the sentence "We hypothesized that specific binding proteins for glycans could be present on breast cancer cells for tumor progression and metastasis when considering that globo-series glycosphingolipids Gb5, SSEA-4, and Globo H are specifically overexpressed on cancer cells" into the Introduction section (pages 5-6) with revised references (38-40).

We additionally performed flow cytometry analyses of MCF-7 breast cancer and MCF-10A breast normal cells for Globo H hexasaccharide binding to support the interaction of cancer cells with the glycan chip. For flow cytometry analyses, Globo H hexasaccharide-Alexa Fluor® 488 conjugates were prepared as previously described method (newly added reference 59). The conjugates were analyzed by high-resolution LC-MS (newly added Figure S22). Consistent with the results of the glycan chip (revised Figures 4c & 4d), flow cytometry analyses clearly demonstrated that MCF-7 cancer cells strongly bound to Globo H hexasaccharide, whereas MCF-10A normal cells did not (newly added Figures 4e-g). These results indicated that Globo H hexasaccharide-binding proteins could be present on the MCF-7 breast cancer cells. We also included the related sentences in the revised manuscript (pages 13 & 19-20) with newly added reference (60).

[Pages 5-6]

To check the feasibility of on-chip enzymatic glycosylation using the **structure switchable** DNA-based glycan chip platform, Globo H series (Table 1) were selected as target complex glycans, which are aberrantly overexpressed on human tumor cells and known to be involved in tumor progression.^{36,37} **Five Globo H-related complex glycans (from trisaccharide to hexasaccharide) were** successfully synthesized from chip surface-immobilized lactose (disaccharide) using the pH-responsive i-motif DNA linker under optimized conditions of stepwise enzymatic glycosylation reactions. **We hypothesized that specific binding proteins for glycans could be present on breast cancer cells for tumor progression and metastasis when considering that globo-series glycosphingolipids Gb5, SSEA-4, and Globo H are specifically overexpressed on cancer cells.**³⁸⁻⁴⁰ We analyzed interactions between breast cancer cells and on-chip-biosynthesized Globo H-related complex glycans to examine the glycan-binding specificity.

[Page 13]

MCF-7 breast cancer cells strongly recognized Globo H hexasaccharide **on the chip**, while MCF-10A breast normal cells did not (Figures 4c & 4d). **This could be also supported by flow cytometry analyses of MCF-7 cancer and MCF-10A normal cells for Globo H hexasaccharide binding (Figures 4e-g).** Except for that of Globo H hexasaccharide, the binding affinities of both cells for other biosynthesized complex glycans were similar. These results indicated that Globo H hexasaccharide-binding proteins can be present on MCF-7 breast cancer cells, which **could support** a previous study that identified the major binding protein for Globo H hexasaccharide on the cancer cell membrane.⁵⁴

[Pages 19-20]

Fluorescence-activated cell sorting (FACS) analysis. MCF-7 breast cancer cells were cultured in DMEM (high glucose) supplemented with 10% (v/v) heat-inactivated FBS, 100 U/mL penicillin, and 100 µg/mL streptomycin. MCF-10A breast normal cells were cultured with Mammary Epithelial Basal

Medium, which contains bovine pituitary extract, hydrocortisone, human epidermal growth factor, insulin, gentamicin, and amphotericin-B. Both cells were incubated at 37 °C in a humidified atmosphere of 5% CO₂ and 95% air, and they were subcultured every 3 days. After incubation, the cells were detached and centrifuged.

To prepare dye-conjugated Globo H hexasaccharide, Alexa Fluor[®] 488 hydrazide (1.75 μmol, 1 equiv.) and Globo H hexasaccharide (0.88 μmol, 0.5 equiv.) were mixed in 1 mL of 100 mM PBS buffer (pH 7.0).⁶⁰ The mixture was incubated at 37 °C for 6 h in a humidity chamber. After the reaction, the mixture was analyzed by liquid chromatograph-mass spectrometry (LC-MS; Waters, Milford, MA, USA) (Figure S22).

To analyze cell-Globo H hexasaccharide interactions on cell surface, both cells were treated with Alexa Fluor[®] 488 conjugated-Globo H hexasaccharide in culture medium at 37 °C for 1 h in a humidified atmosphere of 5% CO₂ and 95% air, respectively. Each solution was centrifuged to remove remaining dye-conjugated Globo H hexasaccharide. After washing with culture medium and DPBS, glycan-treated and non-treated cells resuspended in DPBS were placed into the wells of non-coated 96-well plate. These cells were sorted out by FACS (Beckman Coulter, Brea, CA, USA). Data were acquired and analyzed by using CytExpert software (Beckman Coulter).

[Newly added reference]

60. Lee, B.-S. et al. Biotinylation of peptides/proteins using biocytin hydrazide. *J. Chin. Chem. Soc.* **54**, 541-548 (2007).

[Newly added Figures 4e-g]

Figure 4. (e) Schematic presentation for analyzing the binding of MCF-7 breast cancer and MCF-10A breast normal cells to Globo H hexasaccharide using flow cytometry. FACS analyses for the binding (f) MCF-7 breast cancer and (g) MCF-10A breast normal cells with Globo H hexasaccharide.

[Newly added Figure S22]

20. Bio-LC analyses, Figure S5-S11; the authors should comment as to the slight shifts in the retention times, especially of the major peaks, between 12h, 24h and 48h (a,b,c, respectively).

Author response:

We thank the reviewer for the comment. Actually, we analyzed products of on-chip enzymatic glycosylation reactions using a Bio-LC without a thermostat. Although the air conditioner was used to maintain the temperature of the laboratory with a Bio-LC, there seems to be slight shift in retention times due to a slight change in temperature. To make reader to understand clearly, we included the related sentence in revised manuscript (page 18).

[Page 18]

After incubating for 2 h, the solutions were collected, desalted using a NAP-10 column, and evaporated. The products were analyzed by liquid chromatography (ICS-5000; Thermo Fisher Scientific) using a CarboPac PA100 column (4 mm × 250 mm; Dionex, Sunnyvale, CA, USA), isocratic elution mode with 100 mM sodium hydroxide, and an Ag/AgCl reference electrode for electrochemical detection. **Because the Bio-LC used did not have a thermostat, an air conditioner was used to maintain a constant column temperature.**

21. A Schematic presentation (flow chart of the sequential steps, including the recovery of the glycans from the chips, LC analysis, rehybridization of the glycans (purified?) on array, binding etc would be helpful

Author response:

We thank the reviewer for the good comment. As following the reviewer's comment, we included the schematic presentation in the revised Figure 3a.

[Revised Figure 3a]

Reviewer #3

In this work, Cha and co-workers developed a glycan chip for biosynthesizing glycans based on structure changeable DNA linker. Compared with reported glycan chips, the authors introduced a pH-regulated i-motif to control the immobilization and isolation of glycans synthesized. However, compared with the reported literature (Chem. Commun., 2019, 55, 71-74), I don't clearly know the novelty of this study. Moreover, the authors' conclusions were not clearly supported by convincing data. This work does not warrant the publication in Nature Communications. Here are my concerns:

1. The proposed on-chip glycans biosynthesizing strategy is not improved so much when compared with reported strategies (Chem. Commun., 2019, 55, 71-74). Just tethering an i-motif DNA linker on chips lacks novelty. Isolation and quantification of glycans can be achieved by other strategies, such as polypeptide. Except for glycan isolation, I strongly suggest the authors make a detailed introduction of the improvement in this work.

Author response:

We thank the reviewer for the good comment. Previously reported glycan microarray systems have used a method of synthesizing glycans chemo-enzymatically and then conjugating with linker to immobilize them on the surface (refs 4-8). This process seems to be not suitable for complex glycans such as SSEA4 and Globo H hexasaccharides. This is because these glycans have low synthesis yield compared to simple glycans and the conjugation with the linkers has limitations of being capable of eliminating labile sialic acid and occurring significant loss (newly added refs. 12-14). However, our proposed glycan microarray platform can improve the limitations by immobilizing structurally simple disaccharide with high synthetic efficiency and then synthesizing complex glycans on the chip using glycosyltransferases. Structural change of pH-responsive i-motif DNA enables to isolate glycan-oligonucleotide from surface and optimize on-chip glycan biosynthesis reactions, resulting in the synthesis of structurally defined complex

glycans on the chip. In addition, our proposed on-chip glycan biosynthesis have a great advantage of using small amounts of expensive nucleotide glycans and glyco-processing enzymes for complex glycan synthesis. We clearly demonstrated that our proposed system enables to synthesize five Globo H series from immobilized lactose disaccharide and analyze glycan-involved interactions on chip simultaneously. Therefore, our structure switchable DNA-based glycan chip platform would be superior to other platforms when fabricating glycan microarray consisting of complex glycans that require the use of expensive glycan-processing enzymes and nucleotide sugars due to their low synthetic efficiency by chemical methods.

However, it is necessary to prepare large-scale glycan chip with large number of glycans on the chip for practical application of our developed platform. As explained earlier, we are currently working on further studies to devise a solution for enzymatically synthesizing large number of glycans on the chip by combining with microfluidics system (see the below supporting figure for on-going work). In preliminary study, we checked the biosyntheses of Gb3 trisaccharide and Gb4 tetrasaccharide using DNA-based glycan chip coupled with microfluidic system (unpublished data). We think that our proposed structure switchable DNA-based glycan chip can have a large number of glycans by coupling with microfluidic system containing multichannel.

An increase in the number of available glycan-processing enzymes can contribute to the improved synthetic efficiency of complex glycans and the synthesis of new glycans. Thus, it is very important to screen the activity of a large number of new glycan-processing enzymes for synthesis of diverse complex glycans. Our proposed glycan chip platform can be successfully used to analyze the activity of various glycan-processing enzymes in a label-free manner on a single chip because immobilized glycans can be individually separated by structural change of pH-responsive i-motif DNA. We hope to increase the number of available glycan-processing enzymes by screening their activities using our structure switchable DNA-based glycan chip platform.

We included the related sentences in the revised manuscript (pages 3, 5, 13, 14) with newly added references (12-14).

[Page 3]

Glycan chips have been developed in response to the essential need for the high-throughput analysis of glycan-involved interactions. Currently, glycan chips have been employed in screening therapeutic agents and profiling glycan-involved interactions including glycan-lectin, glycan-cytokine, glycan-antibody, and glycan-virus/bacteria.⁴⁻⁸ However, studies on glycan-related biological processes using glycan chips are still at an early stage relative to our knowledge of the biological functions of proteins and genes. To construct glycan chips, most homogeneous glycans are commonly provided by either chemical synthesis or natural purification using multiphase chromatography. These methods have several limitations, including labor-intensive, costly, and time-consuming processes, the requirement of protection and deprotection steps, and the difficulty of controlling the stereochemistry of glycosidic linkages, resulting in poor purity and low stepwise yield.⁹⁻¹¹ The limitation of access to glycan libraries with diverse structures restricts extensive studies on their roles *in vivo* using glycan chips. In addition, current glycan chip platforms have used a method of conjugating chemo-enzymatically synthesized glycans with linkers to immobilize them on surface.⁴⁻⁸ This method would be unsuitable for complex glycans because these have significantly low synthesis yield compared to simple glycans and the conjugation with the linkers has limitations of being capable of eliminating labile sialic acid and occurring significant loss.¹²⁻¹⁴

Considering that on-chip syntheses of oligonucleotides and peptides have been successfully utilized for genomics and proteomics,¹⁵⁻¹⁷ on-chip enzymatic glycan synthesis would be an attractive tool for glycomics. This method has several merits over conventional methods, including a low-cost and simple process without any additional protection/deprotection, purification, and immobilization steps, the use of small amounts of expensive glycan-processing enzymes and nucleotide sugar donors, the synthesis of many glycosidic

linkages in a straightforward manner, and the direct application of the synthesized glycans for glycomics.

[Page 5]

As shown in Figure 1, conjugates of glycan and complementary single-stranded oligonucleotide can hybridize to i-motif DNAs that are immobilized on the chip surface under slightly basic conditions. As the pH is lowered, i-motif DNA tends to strongly form a quadruplex structure, resulting in the denaturation of the DNA double helix and thereby enabling the isolation of glycan-oligonucleotide conjugates from the surface. This property would make possible to optimize on-chip enzymatic glycosylation by analyzing isolated glycan-oligonucleotide conjugates using liquid chromatography, resulting in synthesis of structurally defined complex glycans on chip. Therefore, our proposed glycan chip platform could improve the limitation of current platforms by immobilizing structurally simple disaccharide with high synthesis yield and then synthesizing complex glycans on the chip using glycosyltransferases under optimized conditions.

To check the feasibility of on-chip enzymatic glycosylation using the structure switchable DNA-based glycan chip platform, Globo H series (Table 1) were selected as target complex glycans, which are aberrantly overexpressed on human tumor cells and known to be involved in tumor progression.^{36,37}

[Newly added references]

12. Esposito, D., Hurevich, M., Castagner, B., Wang, C. -C & Seeberger, P. H. Automated synthesis of sialylated oligosaccharide. *Beilstein J. Org. Chem.* **8**, 1601–1609 (2012).

13. Fair, R. J., Hahn H. S. & Seeberger, P. H. Combination of automated solid-phase and enzymatic oligosaccharide synthesis provides access to $\alpha(2,3)$ -sialylated glycans. *Chem. Comm.* **51**, 6183–6185 (2015).

14. Song, X. et al. Oxidative release of natural glycans for functional glycomics. *Nat. Methods* **13**, 528–536 (2016).

[Supporting figure for on-going work]

Combination of microfluidic system and DNA-based glycan chip for on-chip biosynthesis of large number of complex glycans. (a) Digital photograph of microfluidic system with two channels. (b) Fabrication of DNA-based glycan chip platform. (c) Introduction of closed microfluidic system. (d) Addition of enzyme (*e.g.*, LgtC) solution containing nucleotide sugar using microfluidic system. (e) Separation of biosynthesized glycan-oligonucleotide conjugate (*e.g.*, Gb3) using microfluidic system. (f) Bio-LC analysis of separated glycan-oligonucleotide conjugates.

[Pages 13-14]

These results indicated that Globo H hexasaccharide-binding proteins are present on MCF-7 breast cancer cells, which can support a previous study that identified the major binding protein for Globo H hexasaccharide on the cancer cell membrane.⁵⁴

In this work, we clearly demonstrated that the structure switchable DNA-based glycan chip platform

enables to biosynthesize complex glycans and analyze glycan-involved applications on the chip, simultaneously. For its practical application, it is necessary to prepare large-scale glycan chip with large number of on-chip biosynthesized complex glycans. Thus, we are working on further studies to devise a solution for synthesizing large number of glycans on the chip by combining with microfluidics system containing multichannel. The introduction of microfluidics system can have advantages of providing large number of glycans on the chip by biosynthesizing different glycans for each channel, efficiently reusing glycan-processing enzymes, and minimally using expensive reagents (*e.g.*, nucleotide sugars). We anticipate that the structure switchable DNA-based glycan chip platform combined with microfluidics systems would enable to realize practical applications of glycan chips with on-chip biosynthesized complex glycans.

In particular, the structure switchable DNA-based glycan chip platform might be used to analyze activities of various glycan-processing enzymes in a label-free manner on a single chip because immobilized glycans can be individually separated by structural change of pH-responsive i-motif DNA. We anticipate that the number of available glycan-processing enzymes is increased by efficiently screening their activities using our developed glycan chip platform.

2. In Figure 2, it could be easily noticed that after rehybridization of isolated lactose-oligonucleotide conjugates, the hybridization efficiency is greatly reduced (Figure 2b vs. 2d). Therefore, more pH cycles should be tested to evaluate the efficiency of i-motif for controlling the DNA-based glycan chip platform.

Author response:

We thank the reviewer for the good comment. In this work, the main objective of isolating glycan-oligonucleotide conjugates from the surface was to check the enzymatic glycosylation conversion efficiency for optimization of enzymatic reactions conditions. After optimizing the enzymatic glycosylation conditions, the chip immobilized with freshly synthesized lactose-oligonucleotide conjugates was separated into five blocks, and five Globo H glycan series were biosynthesized on the chip through the enzymatic reactions under the optimized conditions. In the original Figure 2, we tried to check whether the immobilization and isolation of glycan-oligonucleotide conjugates are controlled by structural changes of i-motif DNA linker according to pH change. We think that the original sentence “To check whether the immobilized lactose-oligonucleotide conjugates could be separated and reimmobilized by reversible structural change of the i-motif on the surface” may be confusing for the readers to understand. Thus, we revised the sentence to “To check whether the immobilization and isolation of lactose-oligonucleotide conjugates could be controlled by reversible structural change of i-motif DNAs on the surface”. In addition, as following the reviewer’s comment, we also tested more pH cycles by repetition of second and third round rehybridizations of isolated lactose-oligonucleotide conjugates (revised Figures 1e & 1g). They showed slightly reduced fluorescence intensities, but it was clearly demonstrated that immobilization and isolation are controlled by pH change. The somewhat decrease of hybridization efficiency might be due to the loss of lactose-oligonucleotide conjugates during repeated separation and hybridization reactions. We included the related sentences in the revised manuscript (pages 7-8) with the revised Figure 1.

[Pages 7-8]

DNA hybridization-based lactose immobilization was confirmed through interaction analysis of *Ricinus communis* agglutinin I (RCA₁₂₀) lectin, which specifically binds to terminal galactose (Gal). While the complementary ssDNA-treated surface did not show strong fluorescence intensity (Figure 1b), the lactose-oligonucleotide conjugate-treated surface did (Figure 1c). To check whether immobilization and isolation of lactose-oligonucleotide conjugates could be controlled by reversible structural change of i-motif DNAs on the surface, the fabricated chip was sequentially treated with pH 4.5 and pH 9.0 PBS

buffers. When the chip was incubated with RCA₁₂₀ after treatment with the pH 4.5 solution, no fluorescence was observed (Figures 1d & 1f). To further validate the separation of glycan-oligosaccharide conjugates at acidic pH, single-stranded oligonucleotide conjugated with Alexa Fluor[®] 647 at 5' was used as a model. There was barely no fluorescence even at high concentration of i-motif DNA when treated with acidic solution (pH 4.5) after incubating Alexa Fluor[®] 647-conjugated complementary oligonucleotides onto the chip (Figure S10). Because Alexa Fluor[®] 647 dye is pH-resistant from pH 4 to pH 10,⁴⁰ the fluorescence change seemed to be due to the separation of the dye-conjugated oligonucleotides not to the inactivation of the dye in acidic conditions. These results indicated that lactose-oligonucleotide conjugates were completely separated from the chip surface by forming a quadruplex shape of i-motif DNA under acidic conditions. When the isolated lactose-oligonucleotide conjugates dissolved in pH 9.0 buffer were added again on the chip immobilized with i-motif DNAs, the lactose-oligonucleotide conjugates were successfully rehybridized (Figures 1e & 1g). These results presented that the pH-dependent structural change of i-motif DNAs causes the glycan-oligonucleotide conjugates to be separated from the surface and reimmobilized on the surface.

[Revised Figure 1]

Figure 1. (a) Schematic illustration of a structure switchable DNA-based glycan chip platform using a pH-responsive i-motif DNA linker. (b-g) Fabrication of i-motif DNA linker-based glycan chip platform. (b) Hybridization of complementary single-stranded oligonucleotides and (c) the first round hybridization of lactose-oligonucleotide conjugates with surface-immobilized i-motif DNAs under basic conditions (pH 9.0). (d) The first round denaturation of lactose-oligonucleotide conjugates from surface-immobilized i-motif DNAs under acidic conditions (pH 4.5). (e) The second round hybridization of isolated lactose-oligonucleotide conjugates with surface-immobilized i-motif DNAs under basic conditions. (f) The second round denaturation of lactose-oligonucleotide conjugates from surface-immobilized i-motif DNAs under acidic conditions. (g) The third round hybridization of isolated lactose-oligonucleotide conjugates with surface-immobilized i-motif DNAs under basic conditions. The hybridized lactose-oligonucleotide conjugates were detected by using biotinylated RCA₁₂₀ lectin and Alexa Fluor[®] 647-conjugated streptavidin.

3. In Figure 5, the authors tested the specific binding process of synthesized Globo H hexasaccharide and VK9 on the chip. But more experimental data should be provided. The author should measure the affinity coefficient between the synthesized Globo H hexasaccharide and VK9. In addition, the entire measurement process should be operated in human serum at 37 °C to prove its stability. (For example : Proc. Natl. Acad. Sci., 109(9), 3317-3322: Figure 2A, interaction between dimer peptide and PSD-95 protein).

Author response:

We thank the reviewer for the comment. Actually, the objective of this experiment was to check whether our structure switchable DNA-based glycan chip platform can be used for practical

application by analyzing substrate specificity of already characterized VK9 antibody with the on-chip biosynthesized glycans. It has been established that a mouse monoclonal IgG3 antibody (VK9) has high binding specificity to Globo H hexasaccharide (newly added reference 50). In addition, Wang and co-workers have analyzed surface dissociation constants of VK9 on glycan chip consisted of Globo H and related structures (reference 51). VK9 antibody bound to only biosynthesized Globo H without any cross-reactivity to other Globo H analogs (SSEA4 hexasaccharide, Gb5 pentasaccharide, Gb4 tetrasaccharide, and Gb3 trisaccharide) (revised Figures 4a & 4b), which was consistent of a previously reported relative binding specificity of VK9 to Globo H series (reference 51). This result demonstrated that the structure switchable DNA-based glycan chip combined with on-chip glycan biosynthesis can be successfully used for practical applications. Therefore, we thought that analysis of the affinity coefficient between synthesized Globo H series and VK9 in human serum is out of focus for this work. Instead, we included the related sentence in the revised manuscript to help readers understand more clearly (page 12) with newly added reference (50).

[Page 12]

The relative glycan-binding specificity of a mouse IgG anti-Globo H monoclonal antibody (VK9) was investigated for the on-chip biosynthesized Globo H-related complex glycans. It has been established that VK9 antibody has a high binding affinity to Globo H hexasaccharide without any cross-reactivity to other Globo H analogs.^{50,51} Scanned fluorescence imaging showed that VK9 had a much higher binding affinity to biosynthesized Globo H hexasaccharide than to the other synthesized glycans (Figures 4a & 4b), consistent with a previously reported binding specificity of VK9.⁵¹ The difference in fluorescence intensities of Globo H and SSEA-4 hexasaccharides showed that the Fuc moiety plays a significant role in VK9 binding, which is supported by previous reports.^{51,52} These results confirmed that the Globo H hexasaccharide series was successfully biosynthesized on the lactose disaccharide-immobilized surface.

[Newly added reference]

50. Kudryashov, V. et al. Characterization of a mouse monoclonal IgG3 antibody to the tumor-associated globo H structure produced by immunization with a synthetic glycoconjugate. *Glycoconjugate J.* **15**, 243-249 (1998).

4. In Figure 6, the author should conduct more experiments about the specific binding process of Globo H hexasaccharide to proteins and the strategy lacks generality. Except for MCF-7 cells, the author should utilize the proposed strategy to test more cancer cell lines, including high expression, medium expression, low expression, non-expression cell lines to verify the feasibility of the experiment.

Author response:

We thank the reviewer for the comment. The Globo H hexasaccharide-binding proteins expressed on the surface of cancer cells have not been completely identified yet. In addition, cancer cell lines with different expression levels of Globo H hexasaccharide-binding proteins have not been characterized. Therefore, it is very difficult to quantitatively analyze the binding degree of cancer cells on the glycan chip using cell lines with different expression levels of Globo H hexasaccharide-binding proteins. Instead, we analyzed the quantitative binding of cancer cells on the glycan chip according to the number of surface-expressed Globo H hexasaccharide-binding proteins. The number of surface-expressed Globo H hexasaccharide-binding proteins was adjusted by mixing with MCF-7 cancer and MCF-10A normal cells while keeping the total number of cells constant. As results, the glycan chip exhibited stronger fluorescence intensity

according to increasing the number ratio of MCF-7 cancer to MCF-10A normal cells. These demonstrated the switchable DNA-based glycan chip can be used for quantitative analysis of interactions between cancer cells and glycans. We included the data and related sentences in the revised manuscript (pages 13 and 21) and Supporting Information (newly added Figure S23).

[Page 13]

These results indicated that Globo H hexasaccharide-binding proteins can be present on MCF-7 breast cancer cells, which could support a previous study that identified the major binding protein for Globo H hexasaccharide on the cancer cell membrane.⁵⁴ In addition, quantitative binding analysis was performed using cell mixture with the different ratios of MCF-7 cancer cells to MCF-10A normal cells. The glycan chip exhibited stronger fluorescence intensity according to increasing the number ratio of MCF-7 cells to MCF-10A cells (Figure S23).

[Page 21]

To quantitatively analyze the binding of MCF-7 cancer cells to Globo H hexasaccharide on the glycan chip, the number ratio of MCF-7 cancer cells was adjusted by mixing with MCF-10A normal cells. Dye-treated cells resuspended in cell culture medium were mixed in three ratios (MCF-7 cells accounted for 100%, 50%, and 10%) and applied onto the glycan chip.

[Newly added Figure S23]

Figure S23. (a-c) Scanned raw images and (d) fluorescence intensity plot for the direct binding of mixture of MCF-7 and MCF-10A cells with on-chip biosynthesized Globo H hexasaccharide. The number ratio (%) of MCF-7 cells to MCF-10A cells is (a) 10:90, (b) 50:50, and (c) 100:0. Each value is the mean of forty-nine independent spots, and the error bars represent the standard deviation.

5. The authors mentioned, “the conventional strategies are difficult to control and optimize stepwise glycosylation reactions using the current platforms, resulting in structurally undefined glycans on chip, which leads to unequable and unreliable results.” Therefore, I suggest the authors should make a table to compare the synthesizing efficiency of current strategies and this work.

Author response:

We thank the reviewer for the comment. Shin and co-workers reported the on-chip enzymatic synthesis of the sialyl Lewis X tetrasaccharide from immobilized GlcNAc monosaccharide using three commercially available glycosyltransferases (reference 18). Reichardt and co-workers

reported on-chip fucosylation, galactosylation, and sialylation of N-Glycans (reference 20). Previously, we also reported on-chip biosynthesis of GM1 pentasaccharide and related structures from immobilized lactose disaccharide (reference 19). However, these studies confirmed on-chip enzymatic glycosylations by only interacting with fluorescence dye-labelled lectins. Reichardt and co-workers optimized the conditions of the glycosylation reaction through the change in fluorescence intensity according to the binding of fluorescence dye-labelled lectins (reference 20). But, the saturation in fluorescence intensity could not mean 100% conversion efficiency. It is impossible to calculate the enzymatic glycosylation efficiency from the value of fluorescence intensity of lectin bound on the chip. Therefore, we think that it is difficult to directly compare the enzymatic glycosylation efficiency of the previously reported platforms and our platform. Instead, we included the related sentences in the revised manuscript (page 4).

[Page 4]

Although the feasibility of on-chip complex glycan biosynthesis was confirmed, the current platforms have a key technical barrier. The quantitative information of biosynthesized complex glycans on the surface is unavailable because it is impossible to isolate and analyze immobilized glycans in the current platforms. In addition, these studies analyzed on-chip enzymatic glycosylation reactions by only interacting with fluorescence dye-labelled lectins. It is impossible to calculate enzymatic glycosylation efficiency from fluorescence intensity value of lectin bound on the chip. These drawbacks make it difficult to control and optimize stepwise glycosylation reactions using the current platforms, resulting in structurally undefined glycans on chip, which leads to unequable and unreliable results. Therefore, an advanced strategy is required for on-chip enzymatic glycosylation-based glycan chips.

Other points:

1) Please provide some data to prove that the catalytic activity of the enzymatic glycosylation is independent of pH change.

Author response:

We thank the reviewer for the good comment. The i-motif DNA has been reported to have a transition region of pH 5.7–6.4 for pH-responsive structure change and a linear structure in solution above pH 6.4 (Figure 2b in reference 32). As following the reviewer's comment, we additionally analyzed the stability of the hybridized form under neutral pH conditions in which on-chip glycan biosynthesis were performed. After treating with complementary ssDNA on the i-motif DNA-immobilized chip, the chip was incubated with a pH 7 solution for 24, 48, and 72 h, and then doxorubicin (DOX), which has intrinsic fluorescence, was added to the chip. As a result, there was no change in fluorescence intensity until an incubation time of 72 h (newly added Figure S11). Thus, we confirmed that the hybridization is structurally stable under on-chip glycan biosynthesis reaction conditions. We included the related sentences in the revised manuscript (page 8).

[Figure 2b in reference 32]

Figure 2. Fluorescence spectra and transition curve. (b) Changes in the ratio of the fluorescence intensities of acceptor to total emission ($I_A/(I_A + I_D)$) (O) and ICD (b) of D-I-A probed at 285 nm, respectively, as a function of pH. Theoretical fits obtained from the fitting analysis are shown in red and pink for ICD and ($I_A/(I_A + I_D)$), respectively.

[Newly added Figure S11]

Figure S11. (a) Schematic illustration of an experimental protocol to test the structural stability of the hybridized form under neutral pH conditions. Scanned raw images for incubation times of (b) 0 h, (c) 24 h, (d) 48 h, and (e) 72 h and (f) their quantitative fluorescence intensity plot. Each value is the mean of twenty independent spots, and the error bars represent the standard deviation.

[Page 8]

On-chip enzymatic synthesis of cancer-associated complex glycans. The hybridized form should be structurally stable under neutral pH conditions in which on-chip enzymatic glycosylation reactions are performed. Thus, prior to on-chip glycan biosynthesis, we checked the stability of the hybridized form in pH 7.0 solution according to incubation time. After treating with complementary ssDNA on i-motif DNA-immobilized chip, the chip was incubated with a pH 7.0 solution for 24–72 h, and then doxorubicin, which has intrinsic fluorescence, was added to the chip. As a result, there was no change in fluorescence intensity until an incubation time of 72 h (Figure S11), indicating that the hybridization is structurally stable under on-chip enzymatic glycosylation conditions, which can be supported by a previous study that i-motif DNA has a linear structure in solution above pH 6.4.³²

To substantiate the feasibility of the DNA-based glycan chip platform for on-chip biosynthesis of complex glycans, enzymatic glycosylations were performed on a chip surface using several glycosyltransferases.

2) More data about the comparison between cancer cells and normal cells should be provided to test the generality of the authors' methodology.

Author response:

We thank the reviewer for the good comment. We analyzed the cell-glycan interactions by referring previous reported method (newly added reference 61). In addition, we performed flow cytometry analyses of MCF-7 breast cancer and MCF-10A breast normal cells for Globo H hexasaccharide binding to support the interaction of cancer cells with the glycan chip. For flow cytometry analyses, Globo H hexasaccharide-Alexa Fluor® 488 conjugates were prepared as previously described method (newly added reference 60). The conjugates were analyzed by high-resolution LC-MS (newly added Figure S22). Flow cytometry analyses clearly demonstrated that MCF-7 cancer cells strongly bound to Globo H hexasaccharide, whereas MCF-10A normal cells did not (newly added Figures 4e-g). This can support the methodology for the interaction analysis of MCF-7 breast cancer and MCF-10A breast normal cells on the glycan chip (Figures 4c & 4d). We included the related sentences in the revised manuscript (pages 13 & 19-20) with newly added references (60, 61).

[Page 13]

MCF-7 breast cancer cells strongly recognized Globo H hexasaccharide on the chip, while MCF-10A breast normal cells did not (Figures 4c & 4d). This could be also supported by flow cytometry analyses of MCF-7 cancer and MCF-10A normal cells for Globo H hexasaccharide binding (Figures 4e-g). Except for that of Globo H hexasaccharide, the binding affinities of both cells for other biosynthesized complex glycans were similar. These results indicated that Globo H hexasaccharide-binding proteins can be present on MCF-7 breast cancer cells, which could support a previous study that identified the major binding protein for Globo H hexasaccharide on the cancer cell membrane.⁵⁴

[Pages 19-20]

Fluorescence-activated cell sorting (FACS) analysis. MCF-7 breast cancer cells were cultured in DMEM (high glucose) supplemented with 10% (v/v) heat-inactivated FBS, 100 U/mL penicillin, and 100 µg/mL streptomycin. MCF-10A breast normal cells were cultured with Mammary Epithelial Basal Medium, which contains bovine pituitary extract, hydrocortisone, human epidermal growth factor, insulin, gentamicin, and amphotericin-B. Both cells were incubated at 37 °C in a humidified atmosphere of 5% CO₂ and 95% air, and they were subcultured every 3 days. After incubation, the cells were detached and centrifuged.

To prepare dye-conjugated Globo H hexasaccharide, Alexa Fluor® 488 hydrazide (1.75 µmol, 1 equiv.) and Globo H hexasaccharide (0.88 µmol, 0.5 equiv.) were mixed in 1 mL of 100 mM PBS buffer (pH 7.0).⁶⁰ The mixture was incubated at 37 °C for 6 h in a humidity chamber. After the reaction, the mixture was analyzed by liquid chromatograph-mass spectrometry (LC-MS; Waters, Milford, MA, USA) (Figure S22).

To analyze cell-Globo H hexasaccharide interactions on cell surface, both cells were treated with Alexa Fluor® 488 conjugated-Globo H hexasaccharide in culture medium at 37 °C for 1 h in a humidified atmosphere of 5% CO₂ and 95% air, respectively. Each solution was centrifuged to remove remaining dye-conjugated Globo H hexasaccharide. After washing with culture medium and DPBS, glycan-treated and non-treated cells resuspended in DPBS were placed into the wells of non-coated 96-well plate. These cells were sorted out by FACS (Beckman Coulter, Brea, CA, USA). Data were acquired and analyzed by using CytExpert software (Beckman Coulter).

[Page 20]

Interaction analysis of MCF-7 breast cancer and MCF-10A breast normal cells on the chip. MCF-7 breast cancer and MCF-10A breast normal cells were cultured as described above. To analyze cell-glycan interactions on the chip, cells were stained by referring previously reported method.⁶¹ Both cells (4×10^5 cell/mL, the number of cells was counted by using a C-chip™) were treated with 4 nM calcein-AM in DPBS for 15 min. The solution was centrifuged to remove remaining dye. After washing with DPBS and culture medium, dye-treated cells resuspended in cell culture medium were applied onto the glycan chip at 37 °C for 1 h in a humidified atmosphere of 5% CO₂ and 95% air. To remove unbound cells, the chip was washed once with cell culture medium and twice with DPBS.

[Newly added references]

60. Lee, B.-S. et al. Biotinylation of peptides/proteins using biocytin hydrazide. *J. Chin. Chem. Soc.* **54**, 541-548 (2007).

61. Pai, J. et al. Carbohydrate microarrays for screening functional glycans. *Chem. Sci.* **7**, 2084-2093 (2016).

[Newly added Figures 4e-g]

Figure 4. (e) Schematic presentation for analyzing the binding of MCF-7 breast cancer and MCF-10A breast normal cells to Globo H hexasaccharide using flow cytometry. FACS analyses for the binding (f) MCF-7 breast cancer and (g) MCF-10A breast normal cells with Globo H hexasaccharide.

[Newly added Figure S22]

Figure S22. High-resolution LC-MS analysis of Globo H hexasaccharide-Alexa Fluor 488 conjugate. LC-MS: calculated for [Globo H hexasaccharide-Alexa Fluor 488 conjugate] 1545.38, found 1643.0857 [M+HSO₄]⁻.

3) How about the cytotoxicity of the test?

Author response:

We thank the reviewer for the comment. We analyzed the cell-glycan interactions by referring previous reported method (newly added reference 61). MCF-7 and MCF-10A cells cultured under

adherent condition were incubated with 4 nM calcein-AM in DPBS for 15 min and then centrifuged to remove remaining dye. After washing with DPBS and culture media, dye-treated cells resuspended in culture media were applied to the glycan chip at 37 °C for 1 h in a humidified atmosphere of 5% CO₂ and 95% air. To remove unbound cells, the chip was washed once with culture media and twice with DPBS. A laser scanner was used for analyzing the interactions. Because the interaction with the cells was analyzed based on the already proven method and it did not provide a hazardous environment for the cells, we think that we do not need to test the cytotoxicity. To make reader to understand the method clearly, we included additional information in the revised manuscript (page 21) with newly added reference (61).

[Page 21]

MCF-7 breast cancer and MCF-10A breast normal cells were cultured as described above. To analyze cell-glycan interactions on the chip, cells were stained by referring previously reported method.⁶¹ Both cells (4×10^5 cell/mL, the number of cells was counted by using a C-chipTM) were treated with 4 nM calcein-AM in DPBS for 15 min. The solution was centrifuged to remove remaining dye. After washing with DPBS and culture media, dye-treated cells resuspended in cell culture media were applied onto the glycan chip at 37 °C for 1 h in a humidified atmosphere of 5% CO₂ and 95% air. To remove unbound cells, the chip was washed once with cell culture media and twice with DPBS.

[Newly added reference]

61. Pai, J. et al. Carbohydrate microarrays for screening functional glycans. *Chem. Sci.* **7**, 2084-2093 (2016).

Reviewers' Comments:

Reviewer #1:

Remarks to the Author:

The revised version has addressed my questions regarding whether the release of reaction product is complete and whether the approach is applicable to the construction of large numbers of glycans. Both answers are satisfactory with additional experimental support or proposed solution using the microfluidic system. The paper is acceptable for publication.

Reviewer #2:

Remarks to the Author:

The authors provide extensive texts in reply to comments of the three referees on the original manuscript describing an interesting microfluidic approach to enzymatic synthesis of glycans.

The authors present the DNA-based glycan chip platform as a means of on-chip enzymatic biosynthesis of complex glycans and that it can be coupled with microfluidic systems. The authors intend to develop the platform to render it large scale. Indeed, the study can be regarded as preliminary and awaiting adaptations to a large-scale preparative work for having desired glycans. The products could then be fully characterised structurally and proven to lack impurities prior to use in investigative microarrays. Even traces of impurities are liable to give false results leading to incorrect assignments of glycans recognized.

The reviewer is unconvinced that the method described in its present form constitutes an advanced practical application that would be a worthwhile addition to existing glycan synthesis and microarray analysis platforms. Each glycan batch would need to be repeatedly synthesized and characterised.

MAJOR CRITICISMS

The existence of international glycan array resources with expanding glycan libraries:

The authors seem unfamiliar with the current status of glycan microarray field, including the coverage and diversity of the large glycan libraries of the international microarray resources of CFG/NCFG <https://ncfg.hms.harvard.edu/microarrays> and Imperial College London <https://glycosciences.med.ic.ac.uk/glycanLibraryIndex.html>, and the diverse resources available for complex oligosaccharides, from synthetic laboratories and also commercial sources.

Currently, globoside-related oligosaccharides can be readily obtained from commercial sources, e.g. from Elicityl <https://www.elicityl-oligotech.com/255-globoside-oligosaccharides> and the arrays containing globoside oligosaccharides described in this paper are commercially available <https://www.zbiotech.com/glycosphingolipid-glycan-array.html>.

It is desirable that the authors explain to the glyco and wider community the key advantages of the new array platform. Unfortunately the authors' response falls short and is off target and shows a lack of awareness of the resources available in the glycan microarray field.

Regarding the irregular spots

The irregularities in the sizes of the arrayed spots illustrated indicate that the glycosylations have not proceeded to completion. Averaging the values of the signal intensities of the numerous spots illustrated does not alleviate the concerns of reviewer

Moreover, it appears that some of the images were recorded out of the linear signal range. For a quantitative record of microarray signals saturation of signal intensities should be avoided. It is very likely there will be bigger error-bars if images were recorded without saturation.

ENGLISH

The manuscript suffers, in places, from poor English grammar which makes the intended meaning difficult to follow

Reviewer #3:

Remarks to the Author:

The authors have addressed carefully the reviewers' comments. I think the present version is suitable for publication.

Responses to Reviewers' Comments

We thank the reviewers for their efforts. We feel we've significantly improved the content of the manuscript based on the reviewers' comments.

Reviewer #1

The revised version has addressed my questions regarding whether the release of reaction product is complete and whether the approach is applicable to the construction of large numbers of glycans. Both answers are satisfactory with additional experimental support or proposed solution using the microfluidic system. The paper is acceptable for publication.

Author response:

We thank the reviewer for his/her good comment in reviewing our manuscript.

Reviewer #2

The authors provide extensive texts in reply to comments of the three referees on the original manuscript describing an interesting microfluidic approach to enzymatic synthesis of glycans.

The authors present the DNA-based glycan chip platform as a means of on-chip enzymatic biosynthesis of complex glycans and that it can be coupled with microfluidic systems. The authors intend to develop the platform to render it large scale. Indeed, the study can be regarded as preliminary and awaiting adaptations to a large-scale preparative work for having desired glycans. The products could then be fully characterised structurally and proven to lack impurities prior to use in investigative microarrays. Even traces of impurities are liable to give false results leading to incorrect assignments of glycans recognized.

The reviewer is unconvinced that the method described in its present form constitutes an advanced practical application that would be a worthwhile addition to existing glycan synthesis and microarray analysis platforms. Each glycan batch would need to be repeatedly synthesized and characterised.

MAJOR CRITICISMS

The existence of international glycan array resources with expanding glycan libraries: The authors seem unfamiliar with the current status of glycan microarray field, including the coverage and diversity of the large glycan libraries of the international microarray resources of CFG/NCFG <https://ncfg.hms.harvard.edu/microarrays> and Imperial College London <https://glycosciences.med.ic.ac.uk/glycanLibraryIndex.html>, and the diverse resources available for complex oligosaccharides, from synthetic laboratories and also commercial sources.

Currently, globoside-related oligosaccharides can be readily obtained from commercial sources,

e.g. from Elicityl <https://www.elicityl-oligotech.com/255-globoside-oligosaccharides> and the arrays containing globoside oligosaccharides described in this paper are commercially available <https://www.zbiotech.com/glycosphingolipid-glycan-array.html>.

It is desirable that the authors explain to the glyco and wider community the key advantages of the new array platform. Unfortunately the authors' response falls short and is off target and shows a the lack of awareness of the resources available in the glycan microarray field.

Author response:

We thank the reviewer for the comment. We already know that globoside-related oligosaccharides, which are our target glycans for on-chip biosynthesis in this work, are commercially available. The enzymes required for their syntheses are also commercially available. Globo H series are aberrantly overexpressed on human tumor cells and are involved in tumor progression (refs. 36 and 37). Therefore, we selected Globo H-related oligosaccharides containing sialic acid as target glycans for on-chip glycan biosynthesis. The purpose of this study is not to provide an array of new glycans but to provide a glycan chip platform that simultaneously enables the biosynthesis of complex glycans and the analysis of glycan-involved applications. The proposed platform enables the synthesis of a variety of complex glycans from simple glycans immobilized on the chip by adjusting a combination of glycan-processing enzymes according to microfluidic channels. As the reviewer commented, it is very important for this platform to reproducibly biosynthesize complex glycans on a chip. As shown in Figure 3b, complex glycans were biosynthesized with high reproducibility (relative standard deviation of 0.9–3.4%) under optimized enzymatic reaction conditions. We also determined that several glycans were reproducibly biosynthesized using the proposed platform. We are working on further studies to establish the optimized conditions for various glycan-processing enzymes using the proposed platform. Given the optimized conditions for various enzymes, users can easily fabricate glycan chips (microarrays) composed of diverse glycans without further purification and conjugation with a specific linker. Therefore, our proposed platform would be advantageous in that it allows users to fabricate customized glycan chips (microarrays) by directly biosynthesizing various complex glycans on the chip through a combination of glycan-processing enzymes. We included the related sentences in the revised manuscript (pages 11 & 14).

[Page 11]

To check whether there are minor unreacted starting glycans on the chip after enzymatic reactions under optimized conditions, we analyzed antibody binding to spots where immobilized glycans reacted with glycosyltransferases (Figure 3a). Due to the absence of commercially available anti-lactose and anti-Gb4 antibodies, it was impossible to confirm that all glycosylation processes were completed. However, using anti-Gb5, anti-SSEA-4, and anti-Globo H antibodies, we confirmed that there was no fluorescence when the antibodies against starting glycans were treated on the chip after on-chip enzymatic syntheses of Gb4 tetrasaccharide, SSEA-4 hexasaccharide, and Globo H hexasaccharide (Figures 3c-d & S17). These results clearly indicated that the on-chip enzymatic glycosylations completely proceeded under the conditions used, along with high reproducibility (relative standard deviation of 0.9–3.4%).

[Pages 13-14]

The introduction of a microfluidics system has the advantages of providing a large number of glycans on the chip by biosynthesizing different glycans for each channel, efficiently reusing glycan-processing enzymes, and minimally using expensive reagents (e.g., nucleotide sugars). The structure-switchable DNA-based glycan chip combined with a microfluidics system makes it possible to synthesize a variety

of complex glycans from simple glycans immobilized on the chip by adjusting the combination of glycan-processing enzymes according to channels. Given the optimized reaction conditions for various enzymes, users can easily fabricate glycan chips composed of diverse complex glycans without further purification and conjugation with a specific linker. Our proposed platform would be advantageous in that it allows users to fabricate customized glycan chips (microarrays) by directly synthesizing various glycans on the chip through a combination of glycan-processing enzymes. We anticipate that this platform would enable practical application of glycan chips with on-chip biosynthesized complex glycans.

Regarding the irregular spots

The irregularities in the sizes of the arrayed spots illustrated indicate that the glycosylations have not proceeded to completion. Averaging the values of the signal intensities of the numerous spots illustrated does not alleviate the concerns of reviewer

Author response:

We thank the reviewer for the comment. Because uneven drying of the printed drops causes varying spot sizes and shapes, we analyzed the effect of additives in the printing buffer on drop drying. The additives used were glycerol (10% (v/v)), DMSO (1, 5, 10, and 20% (v/v)), and DMF (1, 5, 10, and 20% (v/v)). There was a difference in the size and morphology of the spots according to the presence or absence of additives, but there was no significant difference in the type and concentration of the additives used. Therefore, we used a printing buffer (pH 4.5) composed of 100 mM sodium phosphate, 10% (v/v) DMF, and 1 M NaCl. However, as shown in the original Figure S10a, samples of the same concentration have somewhat different sizes and morphologies of spots. Thus, we analyzed the surface roughness of gold-coated glass slides to check whether there were any problems in immobilizing the sample on a gold-coated glass slide. As a control, a commercially available NHS-functionalized glass slide (GmbH, Jena, Germany) was used. The gold-coated glass slide was prepared by coating cleaned glass slides with ~100 nm thick gold and an ~10 nm titanium adhesive layer as described previously (refs. 56 and 57). As shown in below supporting Figure 1, the commercially available NHS-functionalized glass slide had a smooth surface, but the gold-coated glass slide had some nanoparticles on the surface. It seems that this surface roughness of gold-coated glass slides might prevent uniform sample immobilization and create spots that are irregular in size and morphology. Therefore, we need to optimize the gold coating method to address the irregularities in the sizes of the arrayed spots. In addition, in the original Figure 2, the intensities of spots are somewhat irregular. This experiment was performed to confirm whether enzymatic glycosylation is possible on the chip, and the enzymatic reaction conditions were not optimized at that time. After on-chip enzymatic glycosylation conditions were optimized, we confirmed the relatively regular intensities of spots in Figures 3, 4, S17, and S23. Therefore, we believe that the on-chip enzymatic glycosylations proceeded to mostly completion under the optimized conditions.

[Original Figure S10a]

Figure S10. (a) Hybridization of Alexa Fluor[®] 647-conjugated oligonucleotides with surface-immobilized i-motif DNAs.

[Supporting Figure 1_NHS-functionalized glass slide]

[Supporting Figure 1_Gold-coated glass slide]

[Figure 2]

[Figure 3c & 3d]

[Figure 4a & 4c]

[Figure S17]

Moreover, it appears that some of the images were recorded out of the linear signal range. For a quantitative record of microarray signals saturation of signal intensities should be avoided. It is very likely there will be bigger error-bars if images were recorded without saturation.

Author response:

We thank the reviewer for the comment. To exclude this signal intensity saturation, we calculated the mean and standard deviation in bar graphs after excluding the highest and lowest signals of forty-nine independent spots. To clarify this calculation, we included the related information in the revised legends of Figures 2, 4, S11, S21 & S23.

[Figure 2]

Figure 2. Scanned raw images and quantitative intensity plots for on-chip enzymatic glycosylation of (a) Gb3 trisaccharide, (b) Gb4 tetrasaccharide, (c) Gb5 pentasaccharide, (d) SSEA-4 hexasaccharide, and (e) Globo H hexasaccharide. Synthesized complex glycans were detected by using biotinylated lectins and Alexa Fluor® 647-conjugated streptavidin. Each value is the mean of forty-nine independent spots (excluding the highest and lowest signals), and the error bars represent the standard deviation. Abbreviations: GS-IB₄, *Griffonia simplicifolia* isolectin B4; SBA, soybean agglutinin lectin; RCA₁₂₀, *Ricinus communis* agglutinin I lectin; MAL II, *Maackia amurensis* lectin II; LTL, *Lotus tetragonolobus* lectin; LgtC, α -1,4-galactosyltransferase; LgtD, β -1,3-N-acetylgalactosaminyltransferase/ β -1,3-galactosyltransferase; α 2,3-SialT, α -2,3-sialyltransferase; α 1,2-FucT, α -1,2-fucosyltransferase. Symbols: blue circle, Glc; yellow circle, Gal; yellow square, GalNAc; red triangle, Fuc; purple square, Neu5Ac.

[Figure 4]

Figure 4. Applications of structure-switchable DNA-based glycan chips. (a, b) Analysis of the glycan binding specificity of the VK9 antibody on the glycan chip. (a) Scanned raw images and (b) quantitative fluorescence intensity plot for the binding of VK9 with biosynthesized Gb3 trisaccharide, Gb4 tetrasaccharide, Gb5 pentasaccharide, SSEA-4 hexasaccharide, and Globo H hexasaccharide. (c, d)

Analysis of glycan-binding specificity of MCF-7 breast cancer cells on the glycan chip. (c) Scanned raw images and (d) quantitative fluorescence intensity plot for the binding of MCF-7 breast cancer and MCF-10A normal breast cells with on-chip biosynthesized Gb3 trisaccharide, Gb4 tetrasaccharide, Gb5 pentasaccharide, SSEA-4 hexasaccharide, and Globo H hexasaccharide. Each value is the mean of forty-nine independent spots (excluding the highest and lowest signals), and the error bars represent the standard deviation. (e) Schematic presentation for analyzing the binding of MCF-7 breast cancer and MCF-10A normal breast cells to Globo H hexasaccharide using flow cytometry. FACS analyses for the binding of (f) MCF-7 breast cancer and (g) MCF-10A normal breast cells with Globo H hexasaccharide.

[Figure S11]

Figure S11. (a) Schematic illustration of an experimental protocol to test the structural stability of the hybridized form under neutral pH conditions. Scanned raw images for incubation times of (b) 0 h, (c) 24 h, (d) 48 h, and (e) 72 h and (f) their quantitative fluorescence intensity plot. Each value is the mean of twenty independent spots (excluding the highest and lowest signals), and the error bars represent the standard deviation.

[Figure S21]

Figure S21. (a) Schematic presentation of the slide format for on-surface biosynthesis of GM3 trisaccharide, GM2 tetrasaccharide, and GM1 pentasaccharide on a lactose disaccharide-immobilized chip surface. (b) Scanned raw images and (c) fluorescence intensity plot for the interactions of cholera toxin B subunit with GM3 trisaccharide, GM2 tetrasaccharide, and GM1 pentasaccharide biosynthesized on the surface. Each value is the mean of twenty-five independent spots (excluding the highest and lowest signals), and the error bars represent the standard deviation.

[Figure S23]

Figure S23. (a-c) Scanned raw images and (d) fluorescence intensity plot for the direct binding of mixture of MCF-7 and MCF-10A cells with on-chip biosynthesized Globo H hexasaccharide. The number ratio (%) of MCF-7 cells to MCF-10A cells is (a) 10:90, (b) 50:50, and (c) 100:0. Each value is the mean of forty-nine independent spots (excluding the highest and lowest signals), and the error bars represent the standard deviation.

ENGLISH

The manuscript suffers, in places, from poor English grammar which makes the intended meaning difficult to follow

Author response:

We corrected the original and revised manuscripts using the English proofreading service.

Reviewer #3

The authors have addressed carefully the reviewers' comments. I think the present version is suitable for publication.

Author response:

We thank the reviewer for his/her good comment in reviewing our manuscript.

Reviewers' Comments:

Reviewer #2:

Remarks to the Author:

The authors provide a considerable amount of data in support of their microscale technology.

I maintain reservations as to whether the technology in its microscale form has a place in unambiguous structural assignments of carbohydrate antigens. I regard it as a prototype that will require rendering (semi)reparative scale such that full structural characterization can be carried out of the final products.

However, in the light of the comments of reviewers 1 and 3 the work can be made available to the community. I would require that my reservations are mentioned in the discussion section of a published form of the manuscript.

Responses to Reviewer's Comment

We thank the reviewer for his/her effort. We feel we've significantly improved the content of the manuscript based on the reviewer's comment.

Reviewer #2

The authors provide a considerable amount of data in support of their microscale technology.

I maintain reservations as to whether the technology in its microscale form has a place in unambiguous structural assignments of carbohydrate antigens. I regard it as a prototype that will require rendering (semi)preparative scale such that full structural characterization can be carried out of the final products.

However, in the light of the comments of reviewers 1 and 3 the work can be made available to the community. I would require that my reservations are mentioned in the discussion section of a published form of the manuscript.

Author response:

We thank the reviewer for the comment. As the reviewer's comment, our proposed platform is still in a relatively early stage of development. We also think that there may be questions about whether the final products are in structurally defined forms. As the reviewer's comment, further work will be performed to make our proposed platform a (semi)preparative scale capable of carrying out full structural characterization of the final glycan products. We believe that this additional work will address the reviewer's reservation and demonstrate the advantages of our platform. As following the reviewer's comment, we included the related sentences in the revised Results & Discussion section (page 14).

[Page 14, Revised Discussion]

The structure-switchable DNA-based glycan chip combined with a microfluidics system makes it possible to synthesize a variety of complex glycans from simple glycans immobilized on the chip by adjusting the combination of glycan-processing enzymes according to channels. Given the optimized reaction conditions for various enzymes, users can easily fabricate glycan chips composed of diverse complex glycans without further purification and conjugation with a specific linker. Our proposed platform would be advantageous in that it allows users to fabricate customized glycan chips (microarrays) by directly synthesizing various glycans on the chip through a combination of glycan-processing enzymes. However, this technology is still in a relatively early stage of development, and there may be questions about whether the final products are in structurally defined forms. Thus, further work will be performed to make our proposed platform a (semi)preparative scale capable of carrying out full structural characterization of the final glycan products. We anticipate that the improved platform would enable practical application of glycan chips with on-chip biosynthesized complex glycans.